# Longitudinal microstructural changes in 18 amygdala nuclei resonate with cortical circuits and phenomics
Karam Ghanem [1,2] ✉, Karin Saltoun [1,2], Aparna Suvrathan[3,4,5], Bogdan Draganski [6,7] & Danilo Bzdok [1,2] ✉

The amygdala nuclei modulate distributed neural circuits that most likely evolved to respond to environmental threats and opportunities. So far, the specific role of unique amygdala nuclei in the context processing of salient environmental cues lacks adequate characterization across neural systems and over time. Here, we present amygdala nuclei morphometry and behavioral findings from longitudinal population data (>1400 subjects, age range 40-69 years, sampled 2-3 years apart): the UK Biobank offers exceptionally rich phenotyping along with brain morphology scans. This allows us to quantify how 18 microanatomical amygdala subregions undergo plastic changes in tandem with coupled neural systems and delineating their associated phenome-wide profiles. In the context of population change, the basal, lateral, accessory basal, and paralaminar nuclei change in lockstep with the prefrontal cortex, a region that subserves planning and decision-making. The central, medial and cortical nuclei are structurally coupled with the insular and anterior-cingulate nodes of the salience network, in addition to the MT/V5, basal ganglia, and putamen, areas proposed to represent internal bodily states and mediate attention to environmental cues. The central nucleus and anterior amygdaloid area are longitudinally tied with the inferior parietal lobule, known for a role in bodily awareness and social attention. These population-level amygdala-brain plasticity regimes in turn are linked with unique collections of phenotypes, ranging from social status and employment to sleep habits and risk taking. The obtained structural plasticity findings motivate hypotheses about the specific functions of distinct amygdala nuclei in humans.

Despite advances in understanding the brain amygdala's anatomy and function, our knowledge about its temporal dynamics in conjunction with connected cortical and subcortical areas remains limited. A detailed understanding of the interplay between amygdala subregions and cerebral cortex units from invasive experiments is limited by between-species differences. However, interpretations of amygdala subspecialization in humans rely heavily on data from past animal experiments—creating an important epistemological gap. Previous investigations have traditionally focused on the amygdala's role in detecting emotional responses exemplified

by correlations between amygdala tissue damage and processing of fearful facial expressions[1]. This, in turn, has ushered a torrent of amygdala research toward emotionally negative stimuli such as fearful and threatening faces[2]. Here we applied an alternative approach by using and mapping the distributed dependencies in structural changes that show covariation ties between amygdala subregions and cortical partners. This analytical approach traced out which exact amygdala subregions show structural plasticity (i.e., longitudinal change over the years) that occurred together with structural plasticity in brain networks. We subsequently delineated

[1]The Neuro - Montreal Neurological Institute (MNI), McConnell Brain Imaging Centre, Department of Biomedical Engineering, Faculty of Medicine, School of Computer Science, McGill University, Montreal, Canada. [2]Mila - Quebec Artificial Intelligence Institute, Montreal, QC, Canada. [3]Department of Neurology and Neurosurgery, Department of Pediatrics, McGill University, Montreal, QC, Canada. [4]Brain Repair and Integrative Neuroscience (BRaIN) Research Program, Montreal, QC, Canada. [5]Research Institute of the McGill University Health Centre, Montreal, QC, Canada. [6]LREN, Department of Clinical Neurosciences, Lausanne University Hospital (CHUV) and University of Lausanne, Lausanne, Switzerland. [7]Neurology Department, Max Planck Institute for Human Cognitive and Brain Sciences, Leipzig, Germany. ✉e-mail: ghanemkaram12@gmail.com; danilo.bzdok@mcgill.ca

how these identified coupled networks are, in turn, associated with phenotypic lifestyle measures indexing behavior and cognition.

Depending on the degree of implication of a brain region in supporting a given behavioral function, it is susceptible to undergo remodeling alterations that are reflected in measures of gray matter volume. The change in gray matter volume in turn, signifies not only an ability to learn but also an enhancement of existing cognitive capabilities and probably also strengthening areas where neural processes are attenuated in other parts of the brain for the sake of compensation. Being able to examine the brain through the lens of structural plasticity provides a window into the effects of various exposures and prompted behaviors in everyday life. Structural plasticity refers to the brain's ability to undergo physical and functional reorganization in response to experiences, learning, and various environmental stimuli. This encompasses modifications in the neuronal connections, synaptic vesicle formation and uptake, neuronal remodeling, myelination, and even the observable changes in gray matter volume[3]. As shown by past imaging and behavioral studies, such adaptability is not limited by age or a particular cognitive domain[4]. Central to understanding this notion of change is the comparison between cross-sectional and longitudinal studies. Past longitudinal analyses have shown their pivotality in comprehending lifelong neuroplasticity[5]. Evidence from past brain-imaging/behavioral studies underscores the sensitivity and specificity of individual-level brain modifications: for instance, training targeted at enhancing distinct empathy systems leads to unique structural gray matter modifications[6]. Similarly, language acquisition in teenagers has demonstrated distinct correlations with changes in gray matter density, emphasizing the malleability of the brain's structural makeup in relation to evolving mental capacities[7]. Furthermore, lateralization plays a role, with certain cognitive functions such as language likely favoring one hemisphere over the other, leading to discernible structural alterations in specific brain regions[3,6–8].

Past animal studies conducted on rats, mice, and monkeys have started to bridge the understanding between MRI-observable changes and the cellular architectures underlying them[4,9]. Notably, investigations spanning humans and rodents have identified commonalities in structural brain changes following task-based training, reinforcing the universality of structural plasticity mechanisms[10]. However, this study adopts a methodology focusing on longitudinal observation rather than direct experimental intervention, offering a nuanced perspective on structural plasticity. Previous research has often established a clear link between administering specific experimental conditions to the study participants (e.g., learning a new motor skill, etc.) and subsequent structural changes[9,10]. In contrast, our approach centers on observing within-subject structural changes over time without a predefined, narrow stimulus in a cohort close to the UK general population. In essence, structural plasticity epitomizes the brain's evolutionary advantage to adapt and evolve in response to a wide array of stimuli and experiences. This malleability, observable through advanced brain-imaging techniques and affirmed through various experimental designs, underpins the rationale for structural plasticity being the main driver for longitudinal studies that track and map the continuum of brain changes over different timescales.

In fact, one of the most agreed upon forms of neuroscience insight, with causal implications, has perhaps been based on invasive axonal tracing studies[11]. Yet, even those laborious experiments lack information on which physical axonal tracts are actually more used or less used to subserve brain adaptations, with manifestations in brain physicality. To start filling this gap of knowledge, longitudinal studies—coming into reach due to recently emerged data resources—now allow the probing of aspects that underpin structural plasticity and enable certain statements with causal implications by examining the changes occurring within the brain of the same individual across time, while recording the variety of life circumstances of the participant's sample in the middle and at the end of their lifespan. If we were able to access all the untapped benefits of longitudinal studies, we would be able to further comprehend brain development for contextualization with evidence from anatomical and histological studies. Richer datasets allow us to employ new analytical approaches to quantitatively revisit classical

questions in neuroscience[12]. Thus, having a large number of subjects in a longitudinal study brought into reach robust statements about plastic adaptations of brain architecture that can be coherently observed at a population scale. Past longitudinal studies have suffered from low numbers of participants. The UKBiobank initiative was able to obtain longitudinal brain scanning on >1400 healthy participants of ages 40–69 years from two different timepoints. Additionally, previous longitudinal studies on structural covariation were also limited by narrow time windows—typically days to several weeks—between the data acquisition time points. Furthermore, in our study, the delay between these two timepoints of 2–3 years propels more authentic insights into brain-behavior changes. Past human brain-imaging studies that examined the amygdala often did so as a single region or under three larger umbrella groups of nuclei: the laterobasal, centromedial, and superficial subdivisions[13,14]. These course subdivisions lumped together a heterogeneous ensemble of microanatomically distinct nuclei[15] which resulted in a loss of contextual information about cortical and subcortical partnerships with amygdala subregions in relationship to external stimuli. Moreover, past amygdala studies are well-defined within laboratory settings but could rarely illuminate the effects of everyday life on plasticity changes. Instead, in this study, we have examined the effects of various lifestyle factors on structural plasticity at the population level, querying ~1000 indicators of behavior, everyday habits, and mental health.

In our study, we believed that we could increase the analysis of subregional anatomical specificity within the amygdala subdivisions by using a tailored set of analyses with high-resolution and high-quality brain-imaging measurements of the amygdala. The analyses conducted within our study are expected to reveal the plasticity effects from subregional amygdala interplay with (sub)cortical regions in a way that shows relationships to behavioral traits at an unprecedented subregional resolution in the amygdala. The amygdala subregional–(sub)cortical associations that we expect to reveal in our analyses can help us trace a relationship between various brain networks and specific amygdala subregions that cooperate to regulate various tasks within the body. In addressing the complex interplay between amygdala subregions and the broader neural circuitry, our study is grounded in specific, hypothesis-driven inquiries into the structural nuances of these relationships. Recognizing the amygdala's role beyond its traditionally discussed links with fear and emotion, we delve into a more detailed picture of its nuclei subdivisions—laterobasal, centromedial, and superficial groups—and their unique contributions to neural processes. This renewed focus is motivated by the premise that distinct amygdala subregions engage in specialized interactions with cortical and subcortical areas, shaping a range of neurocognitive functions from social cognition to decision-making[16,17].

First, we hypothesized that specific subregions within the laterobasal nuclei group exhibit patterns of longitudinal change in conjunction with the prefrontal cortex, reflecting high-level integration in cognitive processes such as in assisting decision-making and sensory information processing. This hypothesis is informed by the known connectivity between the laterobasal amygdala, in particular and prefrontal regions, suggesting a dynamic interplay that supports especially complex cognitive functions[18,19]. Second, we anticipated that the centromedial and superficial amygdala subdivisions exhibit specific patterns of longitudinal change in conjunction with subcortical regions implicated in autonomic response initiation and social cognition. This contention is based on their established roles in emotional processing and social behavior regulation, suggesting a network adaptation to environmental and internal stimuli that is critical for emotionally based attention allocation and social function[20,21]. Third, we expected lateralization effects to occur with longitudinal change, specifically in the brain regions preferentially responsible for social cognition and brain regions responsible for receiving and processing external sensory stimuli, in relation to left-right deviation changes in the superficial and centromedial larger subdivisions. Lateralization effects in the longitudinal change in the centromedial and the superficial amygdala subregions were also expected to be prominent with the longitudinal change in brain regions related to conscious awareness. We expected that this is the case given that specific subdivisions in the amygdala have been found in past brain-imaging studies to have unique lateralization

patterns[16]. In anticipation of lateralization patterns, we propose that particular (sub)cortical brain regions will exhibit distinct longitudinal changes in conjunction with specific amygdala subregions, reflecting hemispheric specialization. This hypothesis is reinforced by evidence suggesting that lateralization in the amygdala is not a uniform feature but varies across its subregions, with implications for specialized functions[22,23]. We anticipate that the laterobasal and centromedial subregions, due to their differential connectivity and functional roles, will demonstrate distinct lateralization patterns in their structural changes over time. This is consistent with past findings, which highlighted hemisphere-specific amygdala engagement in processing emotion-laden stimuli[24]. Accordingly, we hypothesized that specific amygdala subregions will exhibit lateralized structural changes, which will be mirrored by lateralization in the associated (sub)cortical neural systems involved in cognitive and affective functions.

We finally hypothesized at the outset that indicators related to socioeconomic status and related to contributors to mental health, will be most prominent in relationship to longitudinal changes in the amygdala subregion-(sub)cortical region patterns. Past studies have shown that different stress and health implications are associated with stable and unstable social hierarchies, and the study investigates how neural responses differ between these two contexts[17]. Unstable social hierarchies elicited unique neural responses, such as increased activity in areas linked with social-emotional processing and social cognition, particularly when viewing a superior player. The amygdala, which is known for processing socially emotional stimuli and social anxiety related to hierarchical challenges, showed increased activity in unstable social hierarchies[17]. The thus disclosed amygdala subregion-brain network correspondence is expected to show robust links to a variety of broader phenotypes, such as those related to regulating bodily affective states.

## Methods
### Population data resource
The UKBiobank is an epidemiology resource that contains extensive behavioral and demographic assessments, medical and cognitive measures, as well as biological samples for ~500,000 participants recruited from across Great Britain (https://www.ukbiobank.ac.uk/). This openly accessible population dataset aims to provide high-quality brain-imaging measurements for ~100,000 participants. The present study was based on the recent release from February 2020 that provides data from ~40,000 participants with brain-imaging measures and expert-curated image-derived phenotypes of gray matter morphology (T1-weighted MRI) from 48% men and 52% women aged 40–69 years when recruited (mean age 55 years, standard deviation (SD) 7.5 years). A few years after recruitment, a relevant fraction of the original baseline subject cohort was invited for the imaging data collection arm of the study, 1414 of them being the dataset that provided the basis for this study. 2–3 years later, the 1414 participants came back for another imaging visit. At that point, the age range of the 1414 participants ranged between 48 and 81. We attempted to improve comparability and reproducibility in our study by building on the uniform data preprocessing pipelines designed and carried out by FMRIB, Oxford University, UK[25]. All ethical regulations relevant to human research participants were followed, and all participants provided informed consent with information on the participant consent process being openly disclosed (http://biobank.ctsu.ox. ac. uk/crystal/field.cgi?id=200).

### Brain imaging and preprocessing procedures
Matching magnetic resonance imaging scanners (3-T Siemens Skyra) were offered by Siemens (32-channel radiofrequency receiver head coils) in several dedicated brain-imaging data collection sites with homogenized acquisition protocols. The anonymity of the study participants was protected by defacing the brain scans and removing any sensitive meta information while employing automated processing and quality control pipelines[25]. The homogeneity of the brain-imaging data was improved by filtering out noise by means of 190 sensitivity features which allowed for a more reliable identification[25]. It also allowed for a more

reliable exclusion of brain scans with error-inducing features such as excessive head motion.

A three-dimensional (3-D) magnetization-prepared rapid gradient echo (MPRAGE) sequence at 1-mm isotropic resolution was used to obtain structural MRI brain-imaging data as high-resolution T1-weighted images of brain anatomy. Preprocessing included gradient distortion correction (GDC), field of view reduction using the Brain Extraction Tool[26] and FLIRT[27,28], and nonlinear registration to MNI152 standard space at 1-mm resolution using FNIRT[29].

All image transformations were estimated, combined, and applied by a single interpolation step to reduce unnecessary computation. Tissue type segmentation into cerebrospinal fluid (CSF), gray matter (GM), and white matter (WM) was applied by using FAST [FMRIB's Automated Segmentation Tool[30]] to generate full bias-field-corrected images. SIENAX[31], in turn, was used to derive volumetric measures normalized for head size.

### Signal extraction using anatomical reference atlas
The used 109 cortical and subcortical regions were based on the Harvard-Oxford reference atlas as a part of UKBiobank Imaging[32]. Volume measures from 18 amygdala subregions (9 per hemisphere) were extracted, taking into account subject-specific brain anatomy based on FreeSurfer subsegmentation[33]. This FreeSurfer 7.0 suite tool pays special attention to surrounding anatomical structures to refine the amygdala subregion segmentation in each individual participant. To that end, the limbic volumetric segmentation draws on a probabilistic amygdala atlas with ultrahigh-resolution at ~0.1 mm isotropic. The automatic volumetric segmentation of the amygdala using Freesurfer has been successfully evaluated to yield an accurate parcellation of the 18 amygdala subregions[33].

As a preparatory data-cleaning step[27,34], variations in each of our brain region volumes that could be explained by variables outside of scientific interest were regressed out. To account for variation that can be explained by potential confounding influences, we regressed out the effects of possible confounding variables from body mass index, head size, head motion during task-related brain scans, head motion during task-unrelated brain scans, head position and receiver coil in the scanner ($x$, $y$, and $z$), position of scanner table, as well as the data acquisition site, in addition to age, $age^2$, sex, $sex \times age$, and $sex \times age^2$ [25]. The nuisance-cleaned volumetric measures from the 109 (sub)cortical regions and the 18 amygdala subregions served as the basis for all subsequent analysis steps.

Our core analysis was performed on brain-imaging data that encompassed information on the gray matter volume of the (sub)cortical regions in 1414 UKBiobank participants at two different points in time. The gray matter volume data recorded on the second visit occurred ~2.3 years after each participant's first visit (min 24 months, max 36 months, STD 5 months). This brought into reach the creation of volume change variables associated with the amygdala subregions in both hemispheres and the cortical and subcortical regions. The volume change variables were created by subtracting the gray matter volume at the first time point from the gray matter volume at the second time point in each (sub)cortical region and each amygdala (AM) subregion.

### Plasticity covariation between amygdala subregions and brain regions
As our primary analysis, we aimed to probe for patterns of population covariation that provide insights into how structural covariation of structural plasticity events among the segregated amygdala subregions can explain the structural plasticity events among 109 (sub)cortical regions. Partial least squares canonical analysis (PLSC) was a natural choice of method to evaluate a relationship between two rich variable sets. Compared to CCA and PLS-R, PLSC combines the best of both worlds: it provides a useful balance between dimensionality reduction and correlation maximization. PLSC is preferred when the goal is to understand and explore the relationship between two multidimensional datasets. This model class was ideally fitted to our data analysis scenario on the grounds of (i) feature-to-samples ratio, (ii) native auto-correlation in our variable sets with

brain-derived measurements, and iii) the latent-factor decomposition capability. A first variable set $X$ was constructed from the amygdala subregions change in gray matter volumes over time (number of participants $\times$ 18 amygdala parcels matrix). A parallel variable set $Y$ was constructed from the (sub)cortical volume change over time (number of participants $\times$ 109 brain region parcel matrix):

$$X \in \mathbb{R}^{n \times p}$$

$$Y \in \mathbb{R}^{n \times q}$$

where $n$ denotes the number of observations or participants, $p$ is the number of amygdala (AM) subregions, and $q$ is the number of whole-brain regions. Each column of the two data matrices was normalized by $z$-scoring to zero mean (i.e., centering) and unit-variance scaling (i.e., rescaling) across participants $n$. The PLSC algorithm then addressed the problem of maximizing the covariance between low-rank projections, each of the two variable sets or data matrices. The two sets of linear combinations of the original variables are obtained by PLSC as follows:

$$L_X = XV \quad L_Y = YU$$

$$l_{X,l} = Xv_1 \quad l_{Y,l} = Yu_1$$

$$cov(l_{X,l}, l_{Y,l}) \propto l_{X,l}^T, l_{Y,l}^T = max$$

where $V$ and $U$ carry the respective contributions of $X$ and $Y$, $L_X$ and $L_Y$ denote the subject-specific expressions in the derived embedding space derived from $X$ and those derived from $Y$, $l_{X,l}$ is the $l$th column of $L_X$, and $l_{Y,l}$ is the $l$th column of $L_Y$. We define patterns as general principles of population covariation in our target anatomical regions that can be robustly extracted in brain structure at the population level. The goal of our PLSC approach was to find pairs of pattern expression vectors $l_{X,l}$ and $l_{Y,l}$ that yield maximal covariation in the derived latent space that parsimoniously embeds the participant brain data. The data matrices $X$ and $Y$, holding per-parcel volume change, were decomposed into $L$ components iteratively, where $L$ denotes the number of patterns to be estimated by the model. PLSC finds the canonical vectors $u$ and $v$ that maximize the (symmetric) relationship between a linear combination of AM volume changes ($X$) and a linear combination of brain volume changes ($Y$). PLSC identifies the two concomitant projections, $Xv_l$ and $Yu_l$. These resulted in the optimized co-occurrence between patterns of subregion covariation of volume change over time inside the segregated AM and patterns of brain region covariation of volume change across participants over several years.

In other words, each identified principle cross-association was indicative of a two-part latent representation: a constellation of within-subject AM changes and a constellation of within-subject brain changes that go hand in hand with each other at the population level. The set of $k$ orthogonal patterns of covariation is mutually uncorrelated by construction[35]. The patterns of covariation are also naturally ordered from the most important to the least important AM-brain covariation pattern based on the amount of covariance explained between the amygdala and (sub)cortical variable sets. The first and strongest pattern therefore explained the largest fraction of joint plasticity effects between combinations of AM subregion effects and combinations of brain region effects. Each ensuing cross-covariation pattern captured a fraction of structural co-adaptations that are not already explained by one of the $k-1$ preceding patterns. The variable sets were entered into PLSC after a confound-removal procedure based on previous UKBiobank research (cf. above).

Although our analysis pushed PLSC to its extreme functional limit, given that the number of participants was just above 1400 subjects, we were able to achieve explanatory results with high degrees of the significance of the explained variance for the derived unique patterns. We

further extended our analyses to examine magnitudes and directions of change in gray matter volume in the amygdala subregions at different stages of life by examining the median of the expression of the covariation pattern. Furthermore, we examined the cortical and subcortical regions across age and sex to classify the differences in sex and age pattern strengths by extracting the sex- and age-related differences from the median of the subject-specific expressions.

## Hemispheric difference analysis

We next examined possible lateralization effects in the plasticity effects of amygdala subregion changes and brain region changes. For this purpose, we performed a hemisphere contrast analysis to identify hemispheric left-right divergence of the plasticity of how the 9 amygdala subregions in the left versus right hemisphere are differentially tied to that of 109 target brain regions. We supplemented our derived population patterns of AM-brain structural plasticity co-adaptation (cf. last paragraph) by zooming in on amygdala lateralization effects with their systematic whole-brain coupling with longitudinal plasticity changes.

To find the systematic hemisphere differences in the limbic system, we devised a bootstrap difference test[36–38] by using a PLSC solution but based exclusively on the left hemispheric amygdala subregions with the entire brain versus a PLSC solution that is exclusively based on the right hemispheric amygdala subregions with the entire brain. To that end, we started out by implementing 100 bootstrap iterations in which we randomly pulled participant samples with replacements to build alternative datasets of volume change (i.e., same sample size $n$ as the original dataset). We then carried out parallel PLSC co-decompositions of the set of amygdala subregions in the left hemisphere and the brain, as well as one between the set of amygdala subregions in the right hemisphere and the brain in every bootstrap iteration. The brain-wise effect size differences were recorded across the 100 bootstrap datasets to obtain a nonparametric distribution of left-right hemisphere contrast estimates. The two distinct PLSC solutions from each iteration were then matched signature by signature regarding two sources of non-identifiability[39]: sign invariance and signature order. Canonical vectors from each signature that carried opposite signs were aligned by multiplying one with −1. The order of the PLSC signatures was aligned using pairwise Pearson's correlation coefficient between the canonical vectors from every PLSC model.

We then subtracted the amygdala subregion canonical variates in the right hemisphere PLSC solution from the ones in the left hemisphere PLSC solution to get a left–right hemisphere contrast estimate of the lateralization in the amygdala. A similar element-wise subtraction was carried out between the brain region canonical variates in the PLSC solution where the left hemisphere amygdala subregions were used and the PLSC solution where right hemisphere amygdala subregions were used to get a left–right contrast estimate of the lateralization in the brain. We recorded these subtraction-retrieved difference estimates for each vector entry (each corresponding to the degree of deviation in one particular anatomical subregion), and the subregion-wise and brain-wise hemispheric differences were aggregated across the 100 bootstrap iterations to obtain a nonparametric distribution of left–right contrast estimates. This process was performed in every bootstrap iteration, which in turn yielded a quantification of the lateralization strength in the doubly multivariate covariance between the volume change in AM subregions and the volume change in the brain subregions in each hemisphere.

In a final step, based on the obtained bootstrapped confidence distributions, we created uncertainty estimates of the AM-brain covariation in the UKBiobank population cohort. Statistically relevant alterations of anatomical lateralization in the amygdala subregions and their twin effects in whole-brain regions were determined by whether the two-sided confidence interval included zero or not according to the 10/90% bootstrap-derived distribution of (PLSC parameter) difference estimates[40]—an approach that is patron to our doubly multivariate analytical strategy and research question on amygdala-cortex correspondence over time. This non-parametric approach directly quantified the statistical uncertainty of lateralization in the

AM-brain covariation to single out the brain parcels of driving plasticity effects.

## Phenome-wide profiling

As an extension to our primary analysis, we performed a rich annotation of the derived AM-brain covariation patterns by means of a variety of almost 1000 lifestyle factors, demographic indicators, and mental health assessments. To carry out this phenome-wide association analysis, two utilities were designed to obtain, clean, and normalize UKBiobank phenotype data according to predefined rules that were used for feature extraction. We started out with a raw collection of ~15,000 phenotypes that were fed into the FMRIB UKBiobank Normalization, Parsing And Cleaning Kit (FUN-PACK version 2.5.0; https://zenodo.org/record/4762700#.YQrpui2caJ8). Data harmonization was conducted using FUNPACK by carefully curating a collection of phenotypes associated with the 11 categories of interest. The output of FUNPACK, consisting of ~3300 phenotypes, was then inputted into PHEnome Scan ANalysis Tool[41] (PHESANT; 42, https://github.com/MRCIEU/PHESANT) for further refinement, cleaning, and data categorization. The ensuing collection of 977 target phenotypes was then compared to the discovered covariation patterns to probe for relations between coupled AM-brain volume changes over time and behavior.

The first step involves extracting phenotype information that is spread among 11 major categories, ranging from lifestyle to mental health and from cognitive phenotypes to blood pressure measurements, by using FUNPACK on the UKBiobank sample to extract the phenotype information. These categories of phenotypes were predefined in the FUNPACK utility using the -cfg fmrib arguments. They include only lifestyle phenotypes and exclude any brain-imaging-derived information. We discarded the diet category since there were a total of four candidate phenotypes in that category. The FUNPACK setting used to define the phenotype categories contained built-in rules tailored to the UKBiobank. We curated the phenotype data using FUNPACK's built-in toolset. An example of such refinements is removing 'do not know' responses and replacing unmasked dependent data. As such, a participant who answered that they do not use mobile phones was not asked how long per week they spent using a mobile phone. FUNPACK, in this case, would fill in a value of zero hours per week as a response. FUNPACK's built-in rules pipeline yielded 3330 high-quality phenotype columns.

The FUNPACK output was then fed into PHESANT, which is a toolkit used specifically for curating UKBiobank phenotypes[41] (https://github.com/MRCIEU/PHESANT). The toolkit combined phenotypes across visits, normalized and cleaned the data. Additionally, PHESANT categorized the data as belonging to one of four datatypes: categorical ordered, categorical unordered, binary, and numerical. All categorical unordered columns were converted into binary columns to encode a single response. For example, the employment status phenotype was originally encoded as a set of values representing different conditions (e.g., retired, employed, on disability). These conditions were converted into a binary column (e.g., retired true or false). We then combined the output of categorical one-hot encoding of unordered phenotypes with all measures classified by PHESANT as binary, numerical, or categorical ordered. The final set comprised 977 phenotypes.

We deployed both FUNPACK and PHESANT with their default parameter choices for missing data. All columns with fewer than 500 participants were automatically discarded from further analysis as per PHESANT's default procedure. On the other hand, FUNPACK, by default, assessed pairwise correlation between phenotypes and discarded all but one phenotype of a set of highly correlated phenotypes (>0.99 Pearson's correlation rho). The choices of which phenotypes to discard were also automatically streamlined and conducted by FUNPACK.

We were thus able to explore relationships between the subject-wise expression of a given covariation pattern and the 977 phenotypes, with appropriate correction for multiple comparisons. For each extracted phenotype, we computed Pearson's correlation between the given phenotype and the inter-individual variation in a covariation pattern to reveal both the association strength and accompanying statistical significance of the given phenotype-covariation pattern association. Two standard corrections were used to adjust for the multitude of association tests being assessed for each uncovered covariation pattern. Bonferroni's correction for multiple comparisons was used by adjusting for the number of tested phenotypes (0.05/977 = 5.11e−5). The significance of our correlation strength was further analyzed using the false discovery rate (FDR), which is another popular method of multiple comparison correction. The false discovery rate[42] was set as 5%[32,43,44] and computed for each covariation pattern in accordance with standard practice[45]. For visualization purposes, phenotypes in Manhattan plots were colored and grouped according to the category membership defined by FUNPACK.

## Statistics and reproducibility

Our study employed PLSC to explore the relationship between changes in amygdala subregion volumes and gray matter volume changes across two-time points in 1414 UK Biobank participants. The choice of PLSC was based on its effectiveness in handling high-dimensional data and its capability to reveal patterns of covariation between two sets of variables. Given the nature of our data and analysis objectives, PLSC was deemed to be the most appropriate for its ability to handle the longitudinal brain imaging data and for its robustness in high-dimensional contexts. The statistical significance of the PLSC results was assessed using a bootstrapping permutation test ($n = 100$ iterations), providing confidence intervals for the PLSC parameters.

Our study's reproducibility is ensured through the use of data from the UKBiobank and the employment of widely recognized preprocessing pipelines and statistical methods. The brain imaging data were processed using publicly available pipelines, which are accessible for validation and replication purposes[25]. The analytical strategies, including PLSC and bootstrap testing for hemispheric differences, are documented in detail, allowing for replication by other researchers. Our dataset consisted of 1414 participants, with measurements taken at two time points ~2–3 years apart. The detailed documentation of our methods and the public availability of the UKBiobank data support the reproducibility of our findings.

For phenome-wide profiling involving nearly 977 lifestyle, demographic, and mental health assessments, we corrected for multiple comparisons using Bonferroni correction and the false discovery rate (FDR), setting the significance threshold at 5.11e−5 and an FDR of 5%, respectively. This approach rigorously controls for errors arising from multiple comparisons. Descriptive statistics are provided for all datasets, including means, standard deviations, and ranges, as appropriate.

## Reporting summary

Further information on research design is available in the Nature Portfolio Reporting Summary linked to this article.

## Results
### Rationale

At the resolution of single subregions, the amygdala has mostly been investigated by means of axonal tracer injection studies in invasive studies in animals. In the human brain imaging literature, however, three larger umbrella groups of nuclei have repeatedly been investigated: the laterobasal, centromedial, and superficial subdivisions[13,14]. These coarser subdivisions are known to collapse heterogeneous, microanatomically distinct nuclei that we know to exist in the amygdala[15]. Ignoring known subnuclei boundaries muddies the neural processes subserved by specific nuclei and their coupled neural systems in the cortex. To overcome these major shortcomings, we designed an analysis that goes several steps further: (1) we analyzed the amygdala at the granularity level of 18 distinct amygdala subregions and (2) we examined their twin effect changes in gray matter volume over several years in the 18 amygdala subregions and those in the rest of the brain. We explored the principled signatures of conjoint structural variation of a set of 18 amygdala subregions with 109 cortical and subcortical target regions. Several dissociable population regimes of longitudinal change adaptations, hand-in-hand, in the amygdala subregions and (sub)cortical regions were computed using two-pronged multivariate pattern-learning analysis.

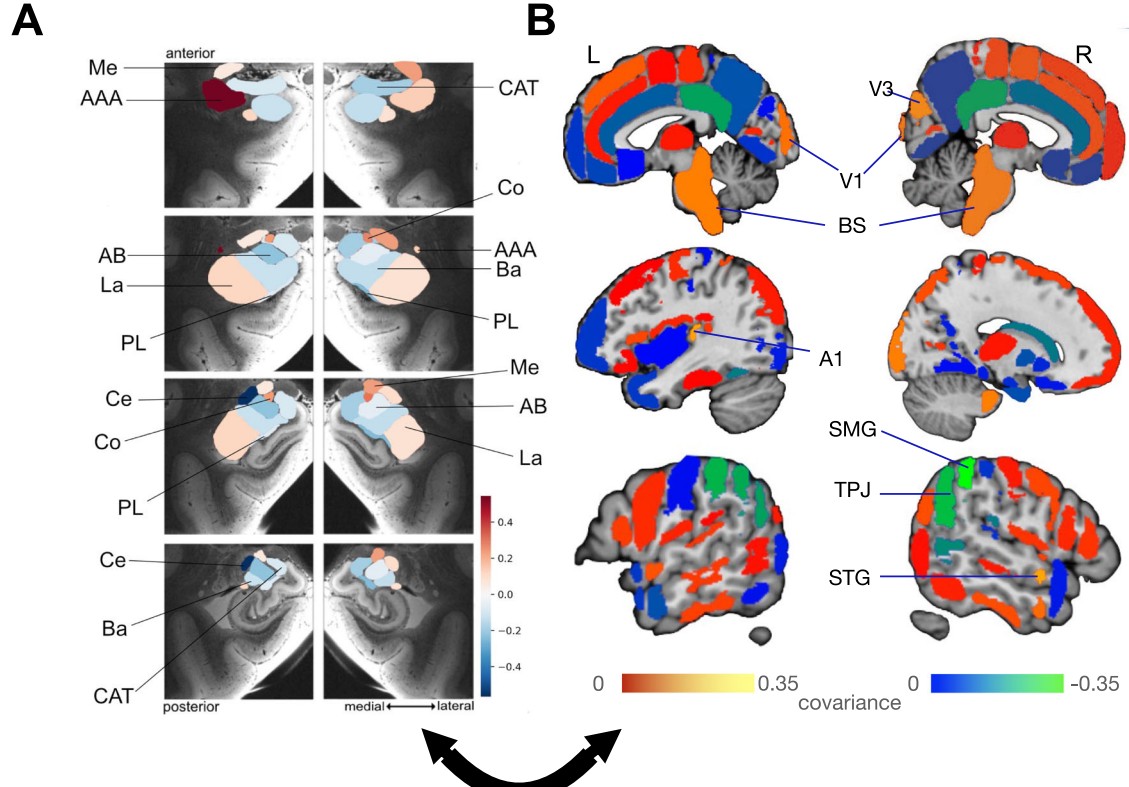

**Fig. 1 | Within-subject structural plasticity effects in the central nucleus and anterior amygdaloid area covary, especially with the inferior parietal lobule.**
Principle patterns of structural covariation due to longitudinal plasticity between gray matter volume change over time in 18 microanatomically distinct amygdala subregions (9 per hemisphere)[33] and gray matter volume change over time in 109 (sub)cortical brain regions (Harvard-Oxford atlas) in 1414 UK biobank participants. We thus derived unique signatures of amygdala-brain co-variation modes. The ensuing parameter weights corresponding to the 18 specific amygdala subregions (hot and cold colors) in **A** outline the structural associations with the resultant brain regions. The subregions with the strongest volume shift in mode 1 are the left central nucleus and the left anterior amygdaloid area. **B** Shows the regions with the highest covariation among the 109 brain regions. The parameter weights indicate the

strength of covariation of volume change in the amygdala subregions with the volume change of the cortical and subcortical brain regions (hot/cold colors = positive/negative volume association). The results show that the structural plasticity of the supramarginal gyrus (SMG), extrastriate visual area (V3), primary visual area (V1), brainstem (BS), primary auditory cortex (A1), superior temporal gyrus (STG), and temporoparietal junction (TPJ) covary with the changes in gray matter volume over time of the left central nucleus and the left anterior amygdaloid area which is complementary to the salience network found in the second pattern, as the left central nucleus is associating with the set of brain regions in the first pattern, while the right central nucleus is co-forming the salience network in the second pattern (cf. Fig. 2). In addition, the subregions covary with the brain regions from the inferior parietal lobule (SMG and TPJ) which primarily contribute to social cognition[47].

The purpose-tailored framework helped achieve a clean co-decomposition of the longitudinal changes of the amygdala subregions and those of the regions in the rest of the brain. To finally chart their potential functional implications, we performed a phenome-wide analysis of the deconvolved AM-brain covariation patterns by invoking 977 lifestyle variables across 11 domains.

### Intrinsic plasticity coupling
Our central analysis yielded six robust modes of change-dependent covariation, with each population mode capturing within-subject volume shifts that co-occurred between the set of amygdala subregions and the set of (sub) cortical regions (Supplementary Fig. 7). This approach explained the degree of joint structural covariation between plasticity events in the 18 amygdala subregions and plasticity events in the 109 cortical and subcortical regions. We evaluated the robustness of the six modes of change-dependent covariation with three separate arguments: (a) highly explained variances of distributed longitudinal changes over several years, (b) numbers of phenotypic hits, and (c) highest statistical significance based on permutation testing framework. More specifically, to quantify model performance, we evaluated the explained variance for the derived unique patterns and found six modes of covariation to yield Pearson's rho values achieving 0.25, 0.24, 0.20, 0.22, 0.22, and 0.23, respectively. Given the significance of all six modes, we brought to bear the phenome-wide analysis as a device to find real-world relevance of the uncovered principled signatures. The results of the

phenome-wide analysis showed no significant phenotype associations above the FDR threshold in the fourth, fifth, and sixth patterns. Thus, we reduced the six significant patterns to three modes of covariation with convincing real-world relevance as evidenced by their ~1000 diverse phenotype profiles. Indeed, these leading three plasticity patterns also showed the strongest joint adaptation effects between the amygdala and the cortex, respectively, with $p$-values < 0.005 according to our permutation test. Given that only the first three patterns showed significant phenotype hits above the FDR threshold, we will only focus on the leading three patterns in the UKBiobank population.

### First pattern uncovers plasticity coupling of the central nucleus and the anterior amygdaloid area with left parietal cortex
The leading plasticity pattern highlighted the dominant covariation of specific gray matter volume change over time in the left central and anterior amygdaloid area subregions with the volume change in the temporoparietal junction (TPJ), extrastriate visual area (V3), primary visual area (V1), brainstem (BS), primary auditory cortex (A1), superior temporal gyrus (STG), and supramarginal gyrus (SMG) (Fig. 1). We found longitudinal change effects to systematically covary at various degrees of effect magnitude and in distinct directions in the first pattern; with the left central nucleus and the left anterior amygdaloid area subregions showing the strongest effects of covariance with other brain regions. We found the left central nucleus,

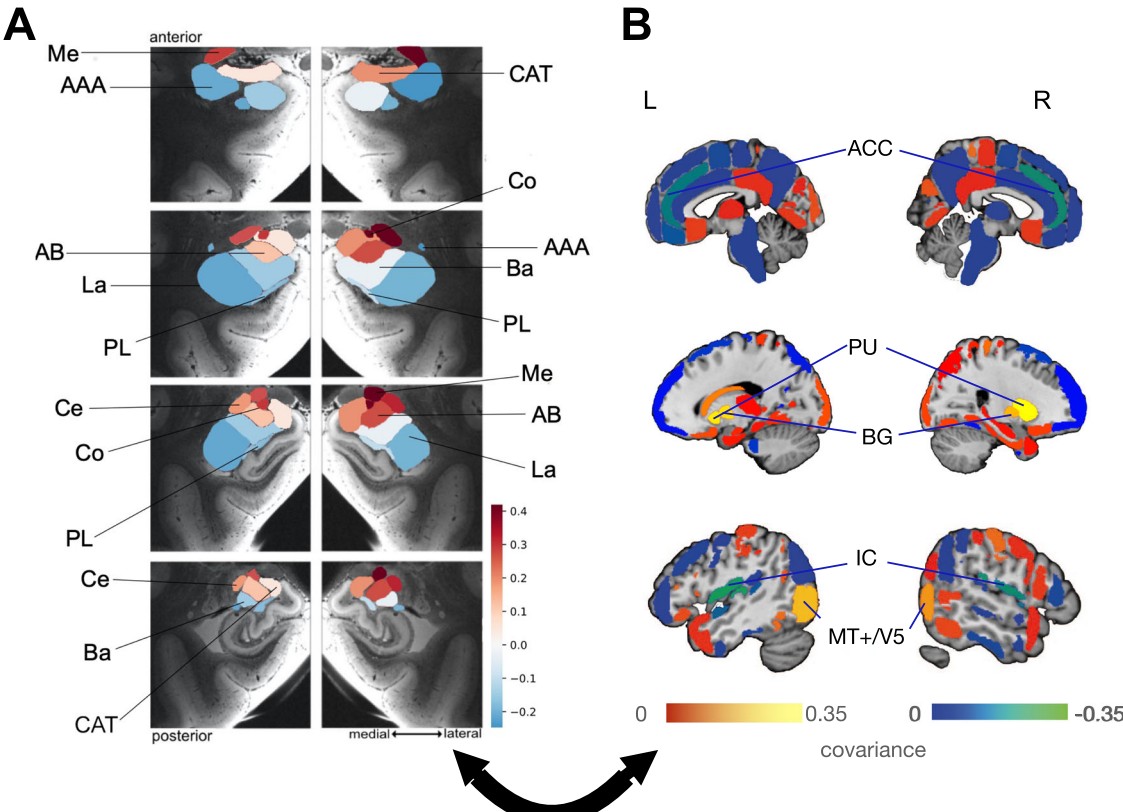

**Fig. 2 | Within-subject structural plasticity ties the medial, cortical, lateral, and central amygdala nuclei to brain regions related to alertness and visual conscious awareness. A** Shows the parameter weights tracking the volume effects of the 18 specific amygdala subregions[33] with their co-occurring structural changes in (sub) cortical partner regions across the brain. The parameter weights indicate the strength of covariation between volume change in the amygdala subregions with the volume change of the cortical and subcortical brain regions (hot/cold colors = positive/negative volume association). The subregions with the strongest volume shifts in mode 2 are the medial, cortical, and central subregions, slightly stronger in the right amygdala. The positive effects in this pattern (hot color), located in the subregions with the strongest volume shift, are larger than the negative effects in terms of magnitude. The collective findings suggest that the right cortical nucleus, the right medial nucleus, and the right central nucleus resonate in structural changes in a number of (sub)cortical brain regions. **B** Shows the brain regions with the highest covariation effects. The results show that the structural plasticity of the basal ganglia (BG), including the putamen (PU), middle temporal visual area (MT+/V5), and salience network (insular cortex (IC) and the anterior cingulate cortex (ACC)) covary with the changes in gray matter volume over time of the right medial, right cortical, and right central amygdala subregions through the coupled interplay of the brain regions and the amygdala subregions to regulate internal conscious awareness through the salience network and external conscious awareness served by the MT +/V5[59,60,68–70].

the left anterior amygdaloid area, and the V1 and BS regions to share a coupled relationship that provides the backbone of what was previously interpreted as regulating internal and external conscious awareness[46–54] with the covarying brain regions and amygdala subregions in the second pattern, as the central nucleus emerged as a recurrent covarying subregion. Further, the central nucleus and the left anterior amygdaloid area covaried with the right TPJ and right SMG, which are part of the right inferior parietal lobule —core nodes in neural systems known to be responsible for such processes as social cognition[47].

**Second pattern discloses plasticity coupling of the medial, cortical and central amygdala with cortical saliency circuits**
In the second most important covariance pattern, we isolated longitudinal change effects in (sub)cortical regions that systematically covaried depending on gray matter changes over time in distinct amygdala subregions (Fig. 2). In the amygdala, the right medial nucleus, right cortical nucleus and right central nucleus were responsible for driving roles, with weaker contributions in their left counterparts. The most pronounced covariations in the brain from largest to smallest were found in the basal ganglia (BG) including putamen (PU), middle temporal visual area (MT +/V5), as well as the insular cortex (IC) and anterior cingulate cortex (ACC) (Fig. 2), both commonly referred to as 'salience network'[55,56]. Thus, these findings indicated that specific subregions in the amygdala undergo a collective change in their gray matter volume over time, with cross-associations with the coherent (sub)cortical system.

**Third pattern emphasizes plasticity coupling of the laterobasal amygdala with prefrontal partners**
In lockstep with the distributed effects identified across the entire brain, all the amygdala subregions in the third pattern of the analysis underwent flanking longitudinal changes in the basal nucleus, lateral nucleus, accessory basal nucleus, and paralaminar nucleus in the right hemisphere and, more so in the left hemisphere (Fig. 3). Our results showed the salient brain regions to have the strongest longitudinal change covariation effects from the largest to smallest magnitudes are the orbitofrontal cortex (OFC), ventromedial prefrontal cortex (vmPFC), dorsomedial prefrontal cortex (dmPFC), dorsolateral prefrontal cortex (dlPFC), ventrolateral prefrontal cortex (vlPFC), parahippocampal gyrus (PHG), lingual gyrus (LING), and the precuneus (PCUN). The cortical region effects underlying this pattern reflected a mostly symmetric volume change in both hemispheres, with a focus on the prefrontal cortex. The covariation of the longitudinal change in the basal, lateral, accessory basal, and paralaminar subregions and the prefrontal cortex were found to strongly covary in the same direction in this pattern which indicates that the laterobasal subregions and the prefrontal cortex share coupled relationships.

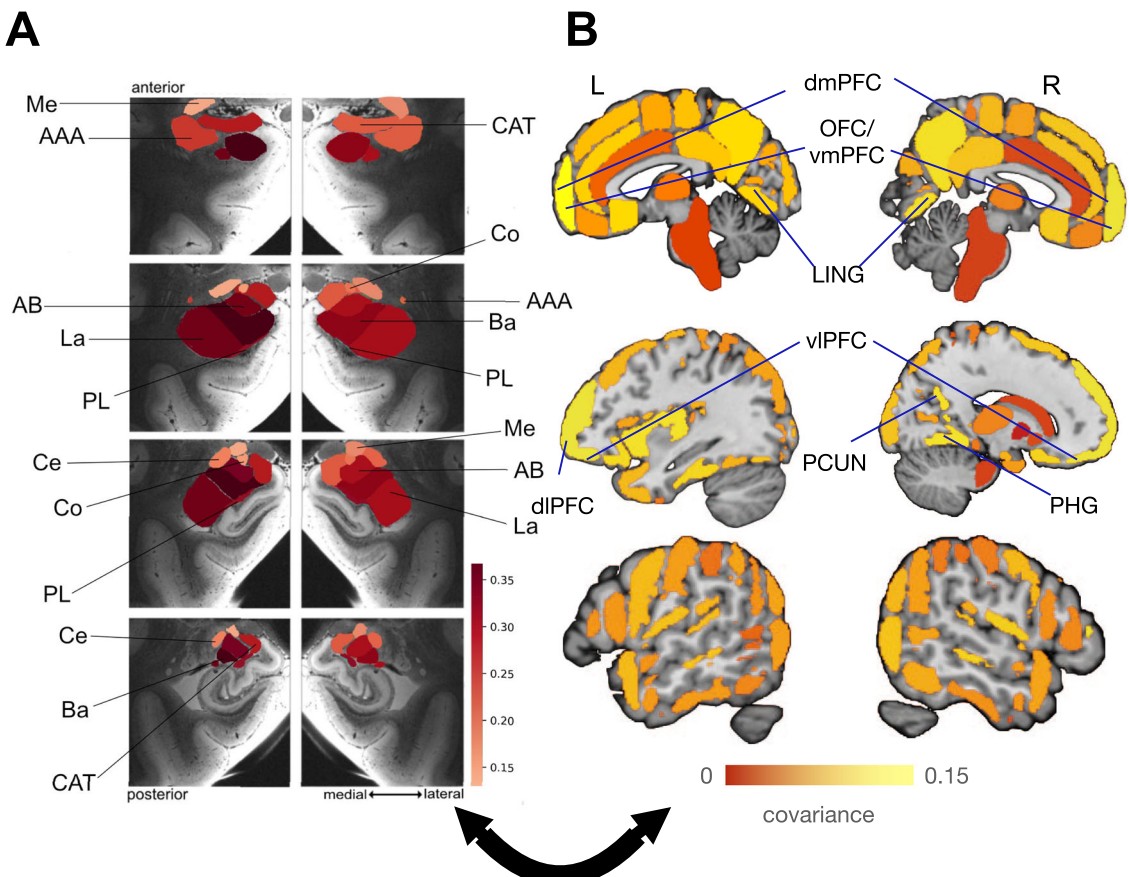

**Fig. 3 | Within-subject structural plasticity links the basal, accessory basal, lateral, and paralaminar nuclei, especially with the prefrontal cortex. A** The basal, accessory basal, lateral, and paralaminar subregions undergo the strongest structural covariation in population mode 3, among all amygdala subregions[33], in the context of the distributed cortical volume changes. **B** Reverberating effects of amygdala covariation located in the prefrontal cortex: the orbitofrontal cortex (OFC), ventromedial prefrontal cortex (vmPFC), dorsomedial prefrontal cortex (dmPFC), dorsolateral prefrontal cortex (dlPFC), and ventrolateral prefrontal cortex (vlPFC),

in addition to the parahippocampal gyrus (PHG), lingual gyrus (LING), and the precuneus (PCUN). The structural plasticity of these regions and, in particular, the prefrontal cortex, which serves several of the most advanced functional integration processes in the human brain, covary in lockstep with the left basal, left accessory basal, and left lateral subregions. The prefrontal atlas regions have direct projections to and from the prefrontal cortex from the amygdala. The highlighted prefrontal cortex regions are known to act as a hub for some of the most abstract forms of cognitive domain-independent classes of neural processes[20,91].

## Hemispheric difference analysis

Next, to more directly examine possible lateralization effects, we performed a dedicated left-right hemisphere contrast analysis to identify hemispheric left-right divergence of the microstructural plasticity of 9 amygdala subregions, with their relation to the same 109 target brain regions (cf. "Methods"). We estimated PLSC models in two separate instances to pit against each other left-right deviation measures in the amygdala subregion set. In one model instance, we estimated the covariation of the amygdala subregions in the left hemisphere with the entire brain, and we then, in a second model instance, estimated the covariation of the amygdala subregions in the right hemisphere with the entire brain in the second model instance. We aimed to identify which anatomical subregions show statistically defensible deviation between (i) how the left hemisphere amygdala covaries with the brain and (ii) how the right hemisphere amygdala and the brain. Our examination of the number of significant patterns in our primary analysis also determined which of the signatures in the hemispheric difference analysis were significant by using the phenome-wide analysis to explore the biological pertinence of each of the candidate patterns through the elimination of the patterns with no phenotypic correlation magnitudes above the FDR threshold when compared to the gray matter volume in the amygdala subregions at the first time point, analogous to our primary analysis (cf. above). As a result, our hemispheric difference analysis produced the same number of significant modes (i.e., 3) as our primary analysis.

## Lateralization effects in the cortical nucleus, anterior amygdaloid area, central nucleus, and lateral nucleus with hemispherically biased brain regions in the first signature

We found driving lateralization effects in the central nucleus and the lateral nucleus, as well as even more strongly in the cortical nucleus and the anterior amygdaloid area (Fig. 4). These amygdala subregions underwent lateralized volume change over time hand-in-hand with the left IC, right hippocampus (HPC), left parahippocampal cortex (PHC), left STG, bilateral ACC and the bilateral vmPFC in the cortex, respectively, found to undergo a salient hemispherically-biased covariation in distinct directions with the hemispherically-biased covarying amygdala subregions. Consider revising for clarity, perhaps by rephrasing like: "The groups formed by the anterior amygdaloid area and central nuclei, as well as the groups formed by the lateral and cortical nuclei, exhibit lateralization in opposite directions. These subregions are key drivers of lateralization, closely linked to both the salience and the ventromedial neural systems.

## Lateralization plasticity effects in the anterior amygdaloid area, lateral nucleus and the cortical nucleus with hemispherically-biased brain regions in the second signature

We observed salient unidirectional lateralization occurring in the covariation of the change in gray matter volume over time in the cortical nucleus, lateral nucleus, and more strongly in the anterior amygdaloid area in the second lateralization plasticity signature (Fig. 5). The left IC, right TPJ, left

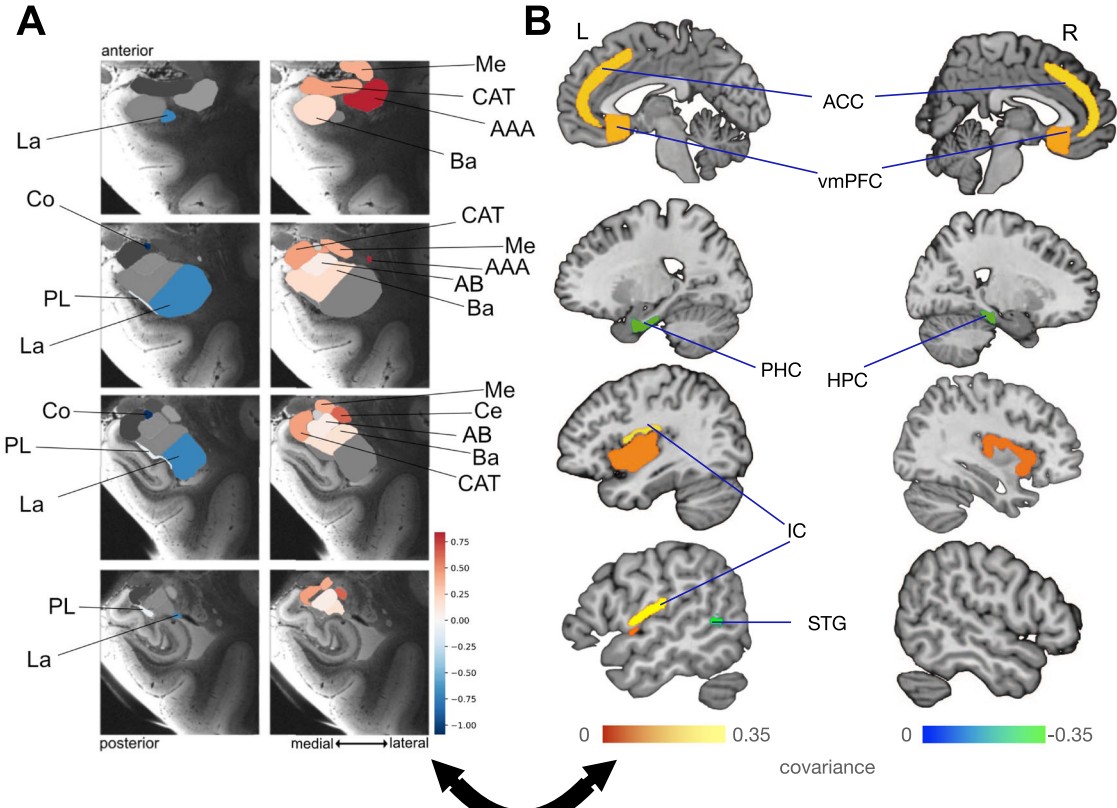

**Fig. 4 | Lateralization plasticity effects driven by the cortical nucleus, anterior amygdaloid area, central nucleus and lateral nucleus co-vary with awareness/alertness-related brain regions.** We performed a hemispheric difference analysis in the context of the left–right divergence of the structural plasticity changes in 9 amygdala subregions with the structural plasticity of 109 brain regions by means of co-decomposition based on partial least-squares canonical (PLSC). We determined how the ensuing subregion patterns lateralized in the 9 amygdala subregions and which cortical/subcortical regions are experiencing lateralization in the covariance of their longitudinal changes with the lateralized amygdala subregions. No color is shown for the brain regions that do not undergo robust lateralization effects. Shown here are the results of the bootstrap difference test as a form of variable selection in signature 1 which in turn conveys the lateralization in the amygdala subregion—brain region covariation. **A** Conveys the direction of lateralization of each of the 9 amygdala subregions in mode 1[33]. The parameter weights of the subregions that diverge between both hemispheres are depicted on 2 columns of 4 coronal slices of the amygdala parcellated into 9 subregions, with each column portraying a different direction of lateralization occurring in each hemisphere. The subregions labeled with cold colors that are depicted in the left column exhibit the same direction of lateralization with varying magnitudes. Simultaneously, the subregions labeled with hot colors in the right column of coronal slices exhibit the opposite direction of lateralization to the subregions in the left column, with each subregion having a distinct effect magnitude. While structural divergences were found to exist in all the amygdala subregions, they are most pronounced in the cortical nucleus, anterior

amygdaloid area, central nucleus, and lateral nucleus. The cortical nucleus and the lateral nucleus lateralize in the opposite direction relative to the anterior amygdaloid area and the central nucleus and lateral nucleus, with the cortical nucleus going through the largest magnitude of hemispherically biased covariation while cross-associating with various brain regions that play significant roles in various functions. **B** Shows the salient brain regions among the 109 brain regions that were found to covary with the changes of the gray matter volume over time in the amygdala subregions. The brain region patterns observed show a robust and systematic divergence in the structural covariation patterns between the right hemisphere and the left hemisphere. The most pronounced structural divergences in the cortex and subcortex occur, from the largest to smallest magnitude, in the left insular cortex (IC/SI), right hippocampus (HPC), left parahippocampal cortex (PHC), left superior temporal gyrus (STG), anterior cingulate cortex (ACC) and the ventromedial pre-frontal cortex (vmPFC). The left IC/SI, ACC, and vmPFC structurally diverge in the same direction as the anterior amygdaloid area and the central nucleus, while the right HPC, left PHC, and left STG do so in the same direction as the cortical nucleus and the lateral nucleus. The lateralization effects in the anterior amygdaloid area/central nuclei groups and lateral/cortical nuclei groups are driven by the lateralization of the salience system and the vmPFC due to a coupling effect in structure, which reveals more particulars on the nature of the associations found in the second pattern (cf. Fig. 2) between the salient amygdala subregions and the salience network from a lateralization perspective.

HPC, left PHC, left inferior temporal gyrus (ITG) and ACC were found to undertake relevant distinct lateralization patterns with the amygdala subregions. The unidirectionally hemisphere-biased longitudinal changes in the amygdala subregions form a coupled and cross-hemispheric relationship with the regions in the inferior parietal lobule (SMG and TPJ) while collectively undergoing a driven lateralization effect with the brain regions, which have been found to co-form a constellation reminiscent of the salience network (ACC and IC) in the second primary pattern. This, therefore, offers more in-depth information about the hemispherically biased tendencies of the salience network uncovered in the primary analysis (cf. Fig. 2) and reveals a coupled association between the covarying amygdala subregions and the inferior parietal lobule.

### Lateralization effects in the cortical, central and medial nuclei in the third signature

In yet another separable lateralization plasticity signature, our hemispheric difference analysis showed the amygdala subregions to be hemispherically biased between both hemispheres but with parameter weights that are relatively less strong in this third signature. We observed hemispherically biased longitudinal change configurations occurring in the medial nucleus and yet more strongly in the central nucleus and in the cortical nucleus. The centromedial amygdala subregions (central and medial nuclei) were found to be driven by collective lateralization in the same hemisphere as the accessory basal nucleus and the cortical nucleus, while all the other subregions collectively lateralized to the other hemisphere.

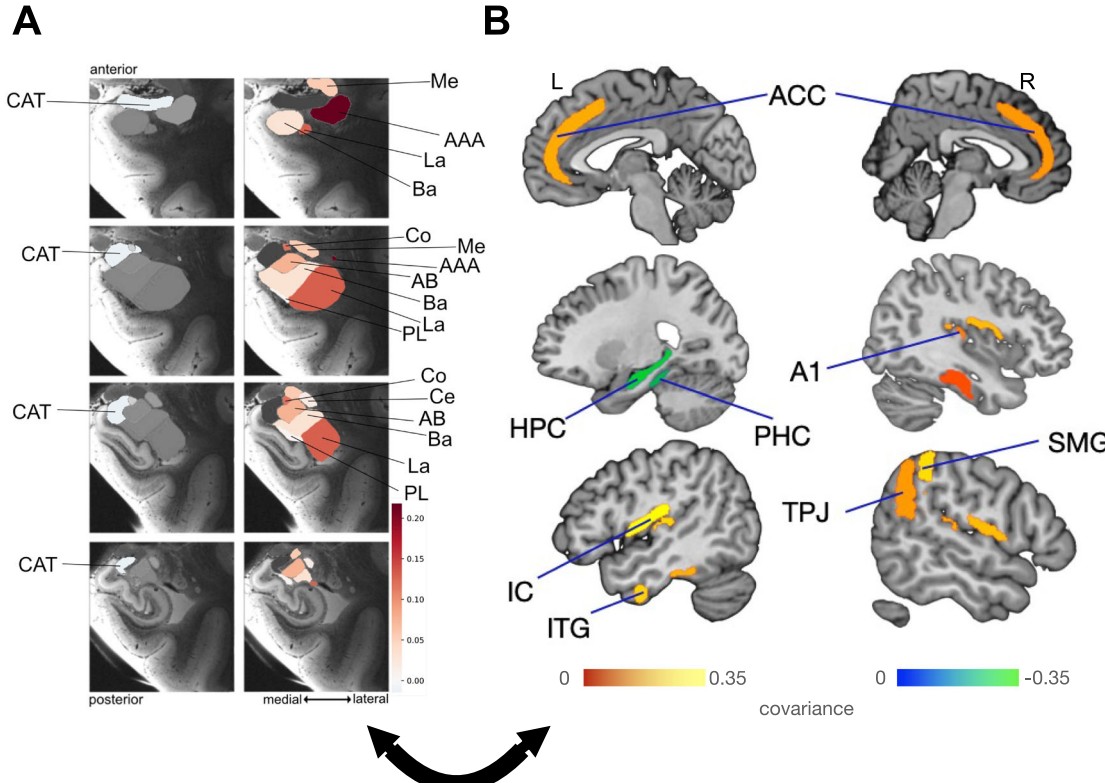

**Fig. 5 | Lateralization plasticity effects driven by the anterior amygdaloid area, lateral nucleus, and cortical nucleus with a lateralization effect in awareness/alertness-related brain regions together with the inferior parietal lobule.** We determined how the ensuing subregion patterns lateralized in the 9 amygdala subregions and which cortical/subcortical regions are experiencing lateralization in the covariance of their longitudinal changes with the lateralized amygdala subregions. The results also show the magnitude of the lateralized covariance in the regions experiencing lateralization among the 109 cortical/subcortical brain regions and among the 18 amygdala subregions. No color is shown for the brain regions that do not undergo robust lateralization effects. Shown here are the results of the bootstrap difference test as a form of variable selection in signature 2 which in turn conveys the lateralization in the amygdala subregion—brain region covariation. **A** Conveys the direction of the lateralization plasticity effects of each of the 9 amygdala subregions in signature 2[33]. The parameter weights of the subregions that robustly diverge between both hemispheres are depicted on 2 columns of four coronal slices of the amygdala parcellated into nine subregions, with each column portraying a different direction of lateralization occurring in each hemisphere. The subregions labeled with cold colors that are depicted in the left column exhibit the same direction of lateralization with varying magnitudes. Simultaneously, the subregions labeled with hot colors in the right column of coronal slices exhibit the opposite direction of lateralization to the subregions in the left column, with each subregion having a distinct effect magnitude. The subregion patterns observed show a robust and systematic divergence in the structural covariation pattern between the right hemisphere and the left hemisphere. While structural divergences were found to exist in all the amygdala subregions, they are greatly pronounced in the anterior amygdaloid area. **B** Shows the salient brain regions among the 109 brain regions that were found to covary with the changes of the gray matter volume over time in the amygdala subregions. The left insular cortex (IC/SI), right temporoparietal junction (TPJ), left inferior temporal gyrus (ITG) and anterior cingulate cortex (ACC) structurally diverge in the same direction as all the amygdala subregions except for the cortico-coamygdaloid transition area which structurally diverges in the same direction as the left hippocampus (HPC) and left parahippocampal cortex (PHC) with the anterior amygdaloid area being subjected to the largest magnitude of hemispherically biased covariation. The findings of the second signature show that the brain regions that were found to co-form a salience network (ACC and IC) in the primary analysis (cf. Fig. 2) and the regions that form the inferior parietal lobule (SMG and TPJ) undergo lateralization that is driven with the lateralization effect in the anterior amygdaloid area, lateral nucleus, and cortical nucleus.

## Phenome-wide analysis

We systematically explored associations between individual expressions of a covariation pattern and 977 phenotypes, applying multiple comparison corrections. Utilizing Pearson's correlation, we analyzed associations and statistical significance between phenotypes and amygdala subregion-cortical region longitudinal change coupling covariation patterns. Two corrections were applied to accommodate association tests for each covariation pattern: Bonferroni's correction, adjusted for the number of tested phenotypes (0.05/977 = 5.11e−5), and the false discovery rate (FDR)[42], set at 5%[32,43,44], according to standard protocols[45]. For visualization, phenotypes in Manhattan plots were color-coded and categorized per FUNPACK-defined membership. Building upon our primary analysis, the thorough annotation of the derived AM-brain covariation patterns in the context of our phenome-wide analysis revealed distinct relationships between the AM-brain covariation patterns and 977 lifestyle factors, demographic indicators, and mental health assessments.

## Phenome-wide analysis of the first pattern associates with body constitution, liver health markers, and blood work indicators phenotypes

In our phenome-wide assays, the first pattern of the primary analysis showed 34 (Bonferroni's correction for multiple comparisons) and 65 (above the FDR threshold) significant associations with target phenotypes (Supplementary Fig. 2; Supplementary Table 1). The strongest phenotypic hits implicated phenotype indicators across three categories: physical general, physical cardiac, and blood assays. Early life factors, lifestyle-exercise and work, lifestyle-alcohol, and lifestyle-tobacco variables were found to be unrelated to the expression of the first pattern. In addition to the phenome-wide analysis, we extracted the sex- and age-related differences from the median of the subject-specific (sub)cortical expressions of the first pattern. The extracted sex- and age-related differences from the subject-specific expressions of the first pattern showed similar trajectories in male and female participants. The male participants formed a relatively horizontal

line of best fit, which indicated a consistent transition of gray matter volume through time, while the female line of best fit started with a lower rate of change of gray matter volume that slowly increased to converge with the male line. Body constitution measures such as body fat percentage, blood work indicators such as red blood cell count, and liver health markers such as glutamyl transferase contributed to the longitudinal change associations between the amygdala and the rest of the brain in the first pattern of the primary analysis. The cross-association between the change in gray matter volume over time in the amygdala subregions and the change in gray matter volume over time in the brain regions showed converging trends in the female and male sexes in the first pattern of the primary analysis.

### Phenotype-wide association studies analysis of the second pattern associates with phenotypes related to socioeconomic status and household members' well-being

Our phenome-wide assays linked the second pattern of the intrinsic plasticity coupling analysis showed 31 (Bonferroni's correction for multiple comparisons) and 70 (above the FDR threshold) significant associations (Fig. 6; Supplementary Figs. 2–6; Supplementary Table 2). The strongest

phenotype hits above the Bonferroni threshold implicated phenotype indicators across five categories in the second pattern: lifestyle general, physical cardiac, cognitive phenotypes, physical general, and blood assays. Early life factors and lifestyle alcohol variables were found to be unrelated to the expression of the second pattern. The phenotype analysis of the second (sub)cortical covariation pattern showed significant associations with phenotypes related to financial well-being, the number of people living in the household, sons/daughters in the household, and the mortality of the mother. The lines of best fit plotted to show that the differences between the sexes are minimal, with a similar trend direction with the progress of age. The cross-association between the change in gray matter volume over time in the amygdala subregions and the change in gray matter volume over time in the brain regions followed a similar trend in both sexes within the age range of 50 and 80 throughout the second and third achieved patterns of the primary analyses. The socioeconomic status-related phenotypes and the community-based phenotypes related to social well-being and family relationships contributed the most to the coupling relationship between the amygdala subregions and the brain regions in the second pattern of the primary analysis.

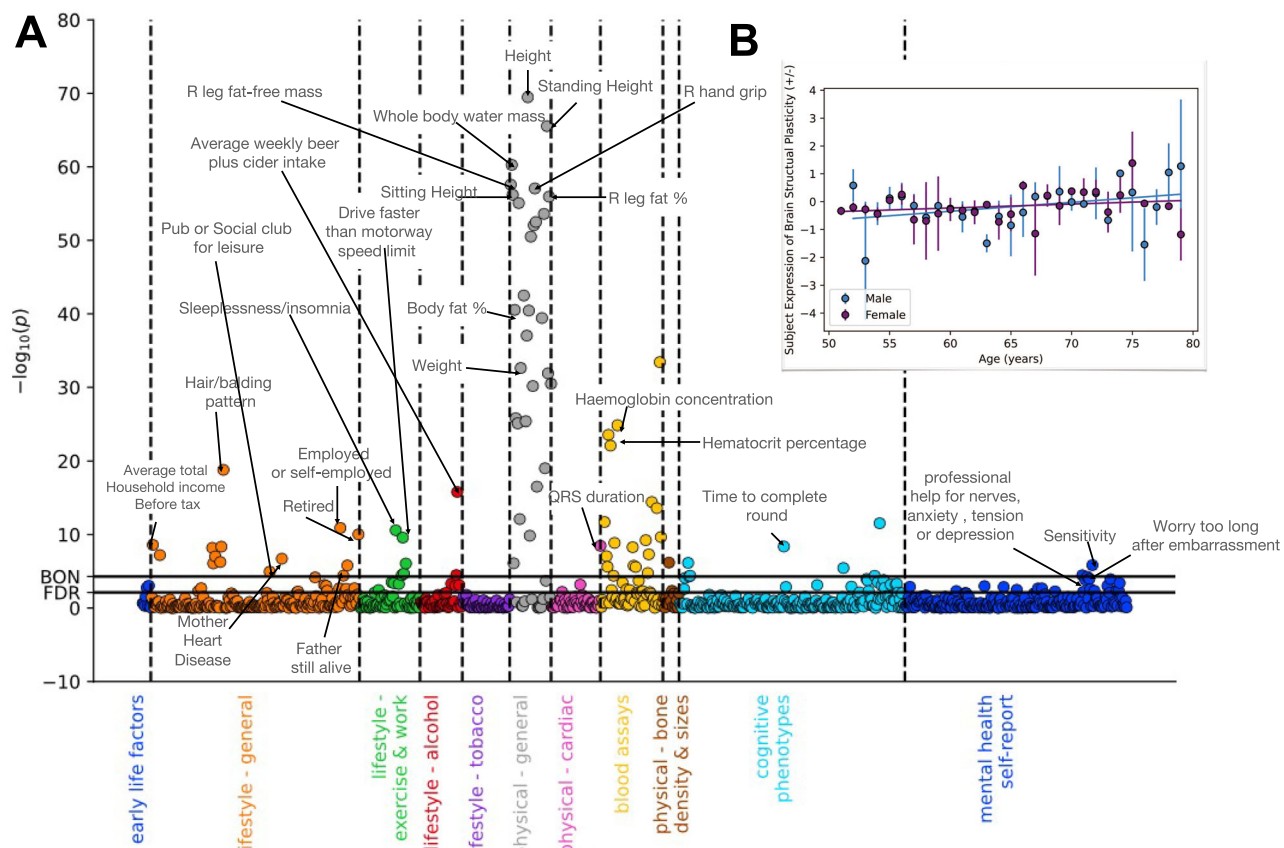

**Fig. 6 | Phenome-wide assay spotlights phenotypes related to socioeconomic status, work status, sleep, risk-taking, and leisure regular activities. A** Manhattan plot shows phenotype associations with individual expressions in the third plasticity pattern (cf. Fig. 3) in the UKBiobank population which charts 977 lifestyle indicators related variables divided across 11 domains. For each phenotype, the plasticity–behavior links are shown in units as p-values (−log. scale). Horizontal lines indicate the significance thresholds at Bonferroni correction and at FDR correction for phenotypes (0.05/977). 146 phenotypes exceeded the FDR threshold, and 79 exceeded the Bonferroni threshold in the third pattern. These significant phenotypes do not endorse or imply causality but rather afford a valuable lens through which the amygdala-brain covariations can be contextualized. **B** Shows the median expression of the covariation pattern in the cortical and subcortical regions across

age and sex, which classify the differences in sex and age pattern strength. Error bars illustrate the lower 5th percentile and upper 95th percentile thresholds obtained by bootstrapping the median of the population, and two lines of best fit to the data are shown: the purple line corresponds to the female data, while the blue line corresponds to the male data. The phenotype analysis showed the most significant associations with physical characteristics such as body fat percentage, phenotypes related to social activities, physical health of parents, household income, employment status, and alcohol consumption-related phenotypes, while other significant phenotypes found in this analysis are hemoglobin concentration, ventricular depolarization (QRS Duration), professional help for nerves, anxiety tension or depression, sleeplessness and insomnia, and balding pattern.

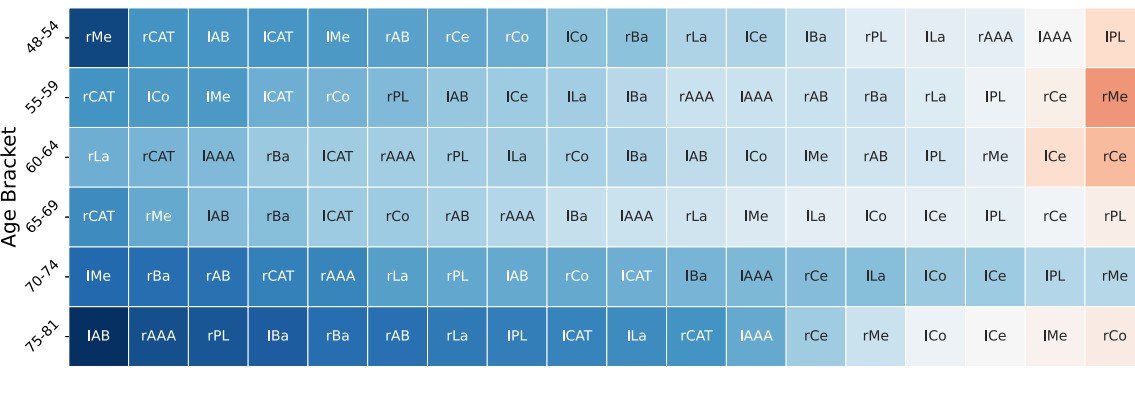
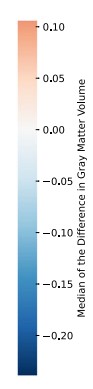

**Fig. 7 | Centromedial and laterobasal nuclei groups change the most over time in UKBiobank participants.** The analysis conveys the specific nuclei change in gray matter volume over time between the second and the first timepoints for the 18 amygdala subregions, arranged in ascending order, from the highest atrophy (cold = atrophy; most negative change) to the largest growth (hot = growth; most positive change) in six different age groups (rows). It shows that the gray matter volume in the amygdala subregions is subject to different degrees of change in distinct directions at various stages of an individual's life. The medial nucleus (Me) was found at both extremities of atrophy and growth of gray matter volume over time in different age groups, and the central nucleus (Ce) and Me were found to undergo the largest growth in gray matter volume over time across all amygdala subregions, while the accessory basal nucleus (AB) atrophied the most across the subregions.

## Phenotype-wide association studies analysis of the third pattern associates with phenotypes related to socioeconomic status, work status, sleep, risk-taking, and leisure regular activities

In our phenome-wide assay analysis, the third pattern of the intrinsic plasticity coupling analysis showed 79 (Bonferroni's correction for multiple comparisons) and 146 (above the FDR threshold) significant associations with target phenotypes (Fig. 6; Supplementary Figs. 2–6; Supplementary Table 3). The most significant phenotypic hits above the Bonferroni threshold implicated phenotype indicators across seven categories in the third pattern: physical general, blood assays, lifestyle general, exercise and work, alcohol, cognitive phenotypes, and mental health-self report. Lifestyle tobacco variables were not related to the expression of the third pattern. The extracted sex- and age-related differences from the median of the subject-specific expressions of the third pattern showed similar gradients and proximity of the male and the female lines of best fit just as the second pattern. Body fat percentage, phenotypes related to social activities, physical health of parents, household income, employment status, as well as alcohol consumption-related phenotypes promoted the coupled behavior in the amygdala subregions and the brain regions in the third pattern of the primary analysis. The cross-association between the change in gray matter volume over time in the amygdala subregions and the change in gray matter volume over time in the brain regions followed nearly indistinguishable trends in both sexes within the age range of 50 and 80 as in the second pattern.

### Ranking of amygdala subregion changes

We finally conducted a 'ranking' analysis to provide a meticulous quantification of how AM subregions undergo substantial modifications in gray matter volume across distinct age categories, thereby resonating with the earlier affirmations regarding the brain's dynamic reconfiguration. The centrality of our amygdala subregion changes ranking analysis lies in the differential trajectory of longitudinal changes in the amygdala subregions across various life stages, underpinning the phenomenological variations intrinsic to individual cognitive and emotional experiences. Our motivation stems from comprehensively characterizing the most significant shifts in the amygdala (AM) subregions in a longitudinal framework, providing granular insights into the temporal dynamics of gray matter volume changes amidst diverse age groups. Given the substantiated ties between gray matter volume changes and various cognitive domains, our careful quantification of the "biggest movers" in AM subregions will potentially unveil the covert neural substrates that underpin the variegated cognitive and emotional landscapes experienced by individuals as they traverse through different life stages. Moreover, it provokes additional queries regarding how diverse environmental, genetic, and lifestyle factors might further modulate these structural transitions.

To examine the magnitudes and directions of change in gray matter volume in the amygdala subregions at different stages of life, we performed a final analysis. The analysis showed the 18 amygdala subregions ranked in terms of the median of the difference in gray matter volume between the second and the first time points (Fig. 7). The cases of largest growth occurred between in individuals with ages 48 and 64 at the first time point, while we found the atrophy to be most prominent in individuals with ages 70–81 at the first time point. Atrophy mainly occurred between the ages 75–81 and was especially dominant in the left and right accessory basal nuclei, right anterior amygdaloid area, left and right paralaminar nuclei, left and right basal nuclei, left and right lateral nuclei. Contrary to the widespread belief that only atrophy occurs with increasing age, the analysis showed that both growth and atrophy occur in the amygdala throughout mid-age. The amygdala subregions undergo different degrees of growth and atrophy, potentially due to life-stage related daily tasks that employ relevant amygdala subregions and due to genetic/lifestyle factors. The medial nucleus experienced the second-largest magnitude of atrophy; after the accessory basal nucleus. The hemispheric difference analysis showed that the medial nucleus and the accessory basal nucleus lateralized to the same hemisphere, perhaps because of amygdala interplay (cf. Supplementary Fig. 1). The last two oldest age groups showed that the medial and accessory basal nucleus underwent the largest atrophy in the same hemisphere which speaks to the findings of the hemispheric difference analysis. The present analysis revealed further information about the ages at which this lateralization is occurring. Moreover, the central and medial nuclei underwent the largest relative growth throughout the entire analysis and both subregions tended to be grouped up together while undergoing growth according to our analysis.

## Discussion

While the here observed structural changes in the brain indeed correlate with learning and environmental interactions, attributing these changes directly to specific functional stimuli can be challenging without experimental manipulation[6,8].

Thus, the findings from our present study contribute valuable insights into coordinated longitudinal trajectories of spatially distributed features of brain structure, simultaneously highlighting the intricacies of inferring

functional stimuli from structural observations alone. Future research could benefit from integrating direct experimental interventions with longitudinal observations to more precisely delineate the relationship between specific narrow features of experiences and structural brain adaptations[4,9].

In essence, structural plasticity can reflect the brain's integrated response to a wide array of environmental stimuli, whether in a carefully parameterized laboratory setting or "in the wild", which are believed to encapsulate both the physical reorganization of neuronal dendrite connections and the functional outcomes of these adaptations[3]. This interdependence underscores the holistic nature of brain plasticity, inviting a more nuanced picture of how experiences and environmental factors sculpt the brain's structure and function over time.

Invasive research into animal brains has shown that the deciphering and gating of responses to self-relevant external information sources depend on several specific nuclei within the amygdala. This heart of the limbic system has long been investigated for its role in channeling overt and covert action steered by behavioral salience[57,58]. In a series of quantitative analyses at unprecedented statistical precision and scale, here we delineated how 18 amygdala subregions undergo volume change in tandem with 109 (sub) cortical brain regions. By tailoring a dedicated analytical framework, our study brought to the surface "cliques" of coordinated amygdala–brain changes that were systematically coupled in their within-participant plasticity effects over several years. We then profiled the derived population-level plasticity patterns using a rich palette of ~1000 phenotypical indicators. The characterization has tied the structural amygdala–brain couplings to various phenotypes such as social status, employment, sleep habits, risk-taking, and leisure activities. Our analyses were performed on participants in the middle and at the end of their lifespan.

Foreshadowing our core findings, the disclosed plasticity associations extend beyond past brain-imaging and lesion studies: salience network nodes were shown to be responsible for regulating what we will call 'internal conscious awareness', whereas MT+/V5 is a key node in a circuit believed to modulate visual stimulus appraisal as a form of 'external conscious awareness'[59,60]. Here, plasticity events specific to the medial, cortical, and central amygdala nuclei showed population co-variation with those of the IC and ACC of the salience network as well as the MT+/V5—a recurring theme across our quantitative analyses. Moreover, we observed asymmetric co-variation in the salience network (i.e., hemispheric difference analysis) concurrently with inferior parietal lobule parts (SMG and TPJ) on the one hand and with gray matter volume changes in the dopamine-receptor-rich anterior amygdaloid area and cortical amygdala nuclei on the other hand. The structural changes in prefrontal cortex regions (OFC, vmPFC, dmPFC, dlPFC, and vlPFC) were covarying with volume changes of the basal, accessory basal, lateral, and paralaminar amygdala nuclei (i.e., primary analysis on brain-amygdala co-variation patterns), which are thought to be responsible for influencing conscious awareness through somatic marker processing[61]. In our phenome-wide analysis, socioeconomic status indicators, such as household income and employment status, were found to explain the coupling relationships between amygdala nuclei and whole-brain regions. The phenotypic indicators of high correlation happen to be more oriented toward the middle and the end of a lifespan, given the age range of the participant's sample. These plasticity–phenotype associations have additionally highlighted inter-personal relationship phenotypes related to social well-being and family relationships, contributing heavily to the longitudinal links of the medial, cortical, and central amygdala nuclei with coordinated changes in the salience network and the MT+/V5 that we observed to occur over the years.

More specifically, we found sub-amygdala co-dependencies between the salience network and the medial, cortical, and central amygdala nuclei to play a dominant role since this constellation reappeared in several population patterns derived by our study. The potential commonality between the cohesive unit of three amygdala subregions and the salience network may lie in their assumed roles in mediating cognitive processes serving interoceptive awareness[14,62–66]. To support such involvements, there exist direct projections from the IC to the medial, cortical and central nuclei as

found, for example, by a rabbit axonal tracing study[67]. The covarying set of amygdala nuclei in humans was previously reported to mediate key aspects of interoceptive awareness based on past brain-imaging studies in humans, psychological studies, and invasive studies in monkeys that investigate the roles of amygdala nuclei in relationship to stimuli[14,62–66].

Interoceptive conscious awareness was found in past studies to mediate autonomic responses and acute pain via a circuit that passes through the medial nucleus of the amygdala[62–65]. In addition, the roles of the central and medial nuclei in driving autonomic responses to salient environmental stimuli were found to contribute to the regulation of conscious awareness[14]. An example of this is the implication of these amygdala nuclei in anxiety responses[14]. More broadly, the functions of the IC and ACC are believed to contribute to self-monitoring of one's inner milieu, both physical state and mental state. Functional interactions within the insula believed to relay between its primarily viscero-somatic posterior and the higher-cognitive anterior parts[68] create a potential interface between the homeostatic conditions of the body and the motivational, hedonic, and social conditions[69]. The salience network was proposed in past studies to realize cognitive control and emotional processing as its core nodes are believed to contribute to the maintenance of a focused action frame[69]. This integration establishes a foundation for relaying between external stimuli and internal states. In line with this, individuals with higher capacities of emotional regulation were reported to feature stronger functional connectivity between IC and the amygdala's central and medial nuclei[70]. Activity changes in the IC were also reported to respond to subjective feelings rather than objective external stimuli, which suggests IC activity binds information on homeostatic, bodily, motivational, and hedonic contingencies[68]. As a consequence of present and previous findings, our analysis took a step forward in terms of granularity by disentangling the medial, cortical, and central nuclei, rather than the amygdala as a monolithic unit, to be flanked by integrative salience network contributions in the context of adaptive structural brain changes over the years.

The cortical nucleus part of the amygdala, which we have revealed to be a partner of structural change to the salience network (i.e., primary analysis), additionally revealed targets of amygdala asymmetry co-variation (i.e., hemispheric difference analysis). The asymmetry co-variation of the cortical nucleus adds weight to a potential adaptive plasticity change of the salience network and internal conscious awareness through its local interplay with the central and medial nuclei. Indeed, in addition to the cortical nucleus, we pinpointed the medial nucleus, central nucleus, and anterior amygdaloid area as robust asymmetry co-variation partners of salience network regions (in the leading signatures of the hemispheric difference analysis). In past axonal tracing studies in rhesus monkeys, fibers were found to be sent from the IC to several specific amygdala nuclei[71,72]. As such, our longitudinal in-vivo data-led analyses were able to recapitulate known biological pathways from earlier wet-science findings in hand-selected monkey samples, thus extending the neuroscience toolkit. Our results add to current knowledge in that the plasticity changes in the salience network happen to not only covary with medial, cortical, central, and anterior amygdaloid area nuclei but also do so with a coherent left–right signature, with possible implications for long-term adaptation in conscious awareness supporting neural systems.

As another fruit from our present effort, select amygdala subregions showing yearlong adaptive remodeling in liaison with the salience network also revealed covariation partners in the inferior parietal lobule (SMG and TPJ). A brain-imaging study has shown the inferior parietal lobule to be closely related to social cognition[47], and its activity responses relate to interoceptive awareness as well[46]. Our observation encourages interpretations based on a number of past psychological and brain-imaging studies speaking to interoception as a necessary component of social cognition[48,51]. Furthermore, past work has highlighted the importance of the inferior parietal lobule in external conscious awareness as it states that damage to the TPJ reliably entails disruptions of awareness[54]. These authors have further highlighted the key role of the inferior parietal lobule in 'constructing' one's own self-awareness[54]. Social cognition is supported by the dopaminergic system, which is reinforced by the release of dopamine by the brainstem in

successful social interactions[49]. Past axonal tracer studies in the monkey amygdala revealed that its central nucleus projects to the substantia nigra, which contains dopaminergic neurons of the nigrotectal pathway, in line with earlier neuroanatomical work[50,73]. Similarly, the anterior amygdaloid area has been established in a past study to have the largest amount of dopamine within the amygdala and one of the relatively highest dopamine amounts within the limbic system globally[74]. Dopamine levels in dissected rat limbic nuclei were measured immediately after decapitation which comes after the feeding process. This insight was gained by examining dopamine levels in the limbic nuclei of rats that were decapitated immediately after the feeding process[74]. According to several present longitudinal analyses, structural adaptations in the brainstem, inferior parietal lobule, and salience network over several years were closely related to two amygdala subregions relevant to reward processing: the central nucleus and the anterior amygdaloid area[74].

The longitudinal coupling between inferior parietal lobule plasticity and salience network plasticity found in our analysis is also of note. This is because several of these regions, notably the insular cortex, are believed to serve integrative somatosensory signal processing[75]. The so-called somatic marker hypothesis states that introspective-somatosensory representations maintained by the vmPFC/OFC help guide human reasoning and decision-making[61]. According to this idea, somatic marker signals, which express themselves as "gut" feelings, are thought to color and thus guide responses to stimuli[61]. The amygdala, the vmPFC/OFC, and the somatosensory cortices are stated to be necessary structures involved in self-awareness of sensations in one's body associated with emotions or 'somatic markers'[61]. Such visceral cue-processing circuits may undergo systematic change over time, as suggested by our results (primary analysis). This assessment highlighted the prefrontal cortex regions (OFC, vmPFC, and dmPFC) as tightly covarying, especially with the amygdala's laterobasal subdivision: the basal, accessory basal, lateral, and paralaminar amygdala nuclei. In particular, volume change happening in the vmPFC/OFC was one of the strongest prefrontal coupling partners with the laterobasal amygdala subregions. This is in line with past invasive animal studies that have found pathways/projections from the OFC terminating in the laterobasal subregions[76–78]. Hence, based on previous research, our longitudinal analysis can be taken to show that shifts in the moderation of arousal to stimuli might be specifically oriented by laterobasal amygdala circuits.

Our findings also favor a plasticity partnership between the prefrontal cortex and especially the laterobasal amygdala subregion—in line with the idea of a division of labor in somatic marker processing influenced by external stimulus input. The amygdala's laterobasal subregion is commonly believed to serve as the receiving hub of stimuli information from the external environment through different sensory cortices[18,19]. This may provide a driver for synaptic plasticity in this particular amygdala segment[20,21]) and potentially the downstream processing partners of this amygdala nucleus. Adding weight to this possibility, the central and medial amygdala nuclei have been shown to be receptive to information issued from the lateral subregion[20]. Our results would be compatible with this interpretation, given that the vmPFC here emerged as one of the strongest covarying partners in the entire prefrontal cortex with the amygdala subregions. Taken together, plasticity events coupled between the amygdala volume and prefrontal volume may occur due to the role of the larger laterobasal subdivision; with potential relevance for adaptations in somatic markers processing, and thus perhaps internal conscious awareness, over time.

Furthermore, our phenome-wide assays linked laterobasal-prefrontal plasticity with various factors: socioeconomic status measures (e.g., income, employment), phenotypes related to social activities, physical health indicators (e.g., body fat percentage), and alcohol consumption.

This phenotype constellation hence corroborates our AM-brain plasticity covariation, pointing to circuits that may serve conscious awareness by means of somatic marker processing. Socioeconomic and social standing being among the strongest phenotype hits in our analysis, links into the roles of the salient amygdala subregions and brain regions that subserve self-

conceptualization and key aspects of one's relation to the rest of society. An individual's standing in the social order is known to lock into mental and physical health[79]. Indeed, a past psychological study reported that social class attribution resonates with individual differences in interoceptive capabilities[80]. Consistently, our phenotype analysis found social circles and alcohol consumption-related phenotypes to be highly significant in the lifestyle domain of the brain changes tying the laterobasal amygdala subregions to the prefrontal cortex. The PFC is established in a past review, based on previous brain imaging studies, to be integral in exerting executive control over alcohol consumption behaviors, with its impairment often leading to disinhibition and increased alcohol intake[81]. Furthermore, the reported role of the basal nucleus in processing somatic stimuli appears to get extended by our phenotype hits disclosed in the physical-general and the lifestyle-alcohol domains. Phenotype associations, where diet and alcohol consumption habits play an important role, can be attributed to the stimulus-value association role supported by the laterobasal subdivision in the amygdala[82]. The tight plasticity coupling of the OFC with the laterobasal amygdala and the reward-seeking themed phenotype hits may be attributed to their roles in gleaning valuable information from external stimuli, including those related to the consumption of alcohol[82]. Hence, the combination of our plasticity patterns and their underlying phenome-wide picture suggests a relationship between key aspects of one's social standing and conjoint laterobasal-prefrontal change, perhaps related to longer-term adaptations in binding somatic markers to subserve processes revolving around conscious awareness.

The putative contributions to conscious awareness by the AM-prefrontal axis through somatic marker processes may further relate to the BG—here, a strong covariation partner. This idea would indeed fall in line with the earlier claim that damage to the BG impairs consciousness in 'exacting tasks' that require focused concentration[83]. These brain correlates were also highlighted by a recent study on neurological patients who suffer from lost consciousness, exhibiting different levels of dissolution of consciousness, including coma and vegetative state[84]. Another brain-imaging study, which investigated connectivity changes during anesthesia, linked decreased levels of consciousness to activity changes in the BG[29]. Functional connectivity between the BG and prefrontal cortex present in unconscious states may relate to our observed covariation between long-term volume change of the BG with that of the medial nucleus, cortical nucleus, and central nucleus in our primary analysis. Such involvement becomes more plausible given that we found these amygdala subregions to also change structurally in conjunction with the salience network and the MT+/V5 (cf. above). Our described covariation of the BG with the medial nucleus, cortical nucleus, and central nucleus and the findings from previous studies help nominate the basal ganglia as one relevant neural node subserving conscious awareness.

We now discuss more in detail the obtained plasticity findings consistent with the notion of 'external conscious awareness'. We observed the MT+/V5 region as a robust covarying region in the company of the salience network with the medial, cortical, and central amygdala subregions. A past brain-imaging longitudinal study that investigated the relationship between the training of cognitive and social skills and changes in brain morphology established the MT+/V5 to be a part of a plasticity network of brain regions that undergo structural plasticity in response to regular training interoceptive awareness over several weeks[6]. A human brain lesion study that examined brain activity using PET in relationship to conscious awareness of visual stimuli has found that there is a direct relationship between MT+/V5 activity and differences in visual conscious awareness[59]. Faster moving visual stimuli were found to cause stronger activity changes of the MT+/V5 and incurred a more accurate conscious discrimination in the human brain lesion study[59]. The 'conscious' state was operationalized in this study through a visual discrimination task involving various visual stimuli[59]. A lesion study took it a step further by examining the behavior of MT+/V5 when the individual is blinded by a V1 lesion[59]. Although this caused the MT+/V5 to lose many of its cortical inputs and reduced motion-related activity in MT+/V5, conscious awareness features were still observed. hus, the intact

**Article**

MT+/V5 may play a crucial role in conscious awareness, particularly in identifying visual objects and encoding the value of visual stimuli based on memory and past experiences. We propose that the MT+/V5 is primarily responsible for 'external conscious awareness' as inputs that do not pass through other regions, such as V1, still lead to conscious awareness of motion.

Our study revealed plastic amygdalar coupling with the MT+/V5 and the salience network that we, in turn, show to be highly related to socio-economic status phenotypes and community-based phenotypes that are related to social well-being and family relationships by our phenome-wide associations study. This probably relates the regulation of internal and external conscious awareness to socioeconomic status, the health of social environments, and the well-being of family relationships. As shown by the past brain-imaging study, structural plasticity in the MT+/V5 occurs when subjects undergo mental training in interoception[6]. Contingent upon the findings in a previous study, the MT+/V5 and salience network are subject to covarying structural plasticity with the right medial nucleus, right cortical nucleus, and right central nucleus due to their roles in interoceptive and exteroceptive awareness. Thus, by examining the correlation with various socioeconomic status indicators, we can determine the extent of volume changes in amygdala subregions and their associated brain regions.

Compatible with facilitation to initiate a fight or flight mode of execution, our amygdala's structural coupling partners were previously found to instigate noradrenergic cells[85]. According to a human brain-imaging study, the central and medial nuclei of the amygdala control emotional and physical responses that entail freezing and action suppression[85]. Past animal studies involving amygdala lesions and axonal tracing have emphasized the role of the central nucleus in regulating defensive states. The central nucleus integrates relevant information, creating memories that assist in formulating strategies against various threats. Additionally, the central nucleus initiates risk assessments for threat detection and regulates cardiovascular activity during dangerous situations[86]. The central nucleus is believed to initiate switches between the fight or flight modes of execution depending on the appropriate reaction the central nucleus prompts[87]. The increase in noradrenaline levels is a direct physiological reaction to stress, according to rat lesion studies and psychological studies on humans and rats[88–90]. Parts of the salience network and the MT+/V5 were also found to be involved in risk prediction with the noradrenergic system in a past brain-imaging study linking internal conscious awareness to risk assessment during fight or flight mode[60].

In line with our findings on the amygdala, the idea of conscious awareness and interoception is coherently extended from being only an internal process to regulating conscious awareness both internally and externally. External conscious awareness may seem more abstract than its internal counterpart because it encompasses one's subjective sense of relevance and presence within their environment. However, insights into the roles of the central and medial amygdala nuclei in awareness and interoception have shed light on this concept. The roles of the central and medial nuclei have been verified in conscious awareness in various forms, internal and external, as they have shown their roles in controlling fight or flight through noradrenaline regulation.

## Data availability
All used data is available to other investigators online (ukbiobank.ac.uk, https://fsl.fmrib.ox.ac.uk/fsl/fslwiki/Atlases). More information on the FreeSurfer 7.0 suite tool used to obtain measures from the 18 amygdala subregions is available to other investigators online (https://surfer.nmr.mgh.harvard.edu/fswiki/HippocampalSubfieldsAndNucleiOfAmygdala). The source data behind the graphs in the paper can be found in Supplementary Data 1.

## Code availability
The analysis scripts that reproduce the results of the present study are available online: https://github.com/dblabs-mcgill-mila/amygdala_plasticity.

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

## Acknowledgements

A.S. was supported by an FRQS (Fonds de Recherche Santé Québec) Chercheurs-boursiers Junior Award #266531.

## Author contributions

K.G. and D.B. conceived, executed, and wrote the paper. K.S., A.S. and B.D. analyzed the data and edited the manuscript.

## Competing interests

The authors declare no competing interests.
