## [Peer Review File · Communications Biology]

Reviewers' comments:

Reviewer #1 (Remarks to the Author):

The paper "Longitudinal microstructural changes in 18 amygdala nuclei resonate with cortical circuits and phenomics" investigated morphological changes of 18 bilateral amygdala subregions (9 per hemisphere) in relation to changes in 109 cortical and subcortical regions across the whole brain in a longitudinal design using MRI data from ~1400 subjects from the UK biobank. These pattern of volumetric changes between the amygdala subregion and the rest of the brain were then submitted to phenome-wide associations linking brain plasticity changes to a broad set of phenotypes.

Characterizing the functional role of amygdala nuclei in humans is of great interest to the neuroscience community and using a large richly characterized sample across multiple timepoints seems like a potentially rich way to address the issue.

I have several severe major concerns regarding the motivation, methodological approach and consequently the conclusions the authors draw from their reported findings.

1. The specific hypotheses of the study are not clearly stated. The introduction described broadly what will be done, but fails to motivate what results may be expected given prior literature. If the study design is purely exploratory, it should be explicitly stated. The lack of hypotheses is in great contrast to the in depth interpretation of their findings in the discussion.

2. Relatedly, it should be clearly stated that the age range of this dataset and thus brain morphological changes are not in the young adult range but more towards the changes in the brain related to aging. The introduction and most of the discussion is not emphasizing that sufficiently; the abstract does not report age range of the sample. The presented conclusions of their findings would be limited towards the middle and end of the lifespan. Along the same lines, the hemispheric difference analysis is not well (??) motivated. Why did the authors consider to investigate a "systematic hemisphere difference in the limbic system" with respect to amygdala subregion functional change across age?

3. The description of methodological procedures lack clarity, contains errors and is not adequately accounting for bias in template choice.

- My most severe concern is in the design choice to address the longitudinal data set. I assume the authors used FreeSurfer (at least in part, see comment below) to assess volumetric changes. FreeSurfer has the option of using a longitudinal processing stream which addresses the problem of biasing the analyses with respect to the first timepoint. Regardless of choice of processing tools, there is always some level of random variation in the processing procedure (e.g. algorithm initialization) that affects robustness and sensitivity of the overall longitudinal analysis. Using the output of a cross-sectional approach for both timepoints is thus unnecessarily increasing variability of results across timepoints and may be severely impacting the robustness of the results.

- In the top of page 5 the authors state that "SIENAX was used to derive volumetric measures normalized for head size". In the following paragraph they state to use the FreeSurfer tool but cite FSL (Jenkinson et al., 2012). The reference provided for the segmentation (and validation thereof) of amygdala nuclei is also incorrect (again stating an FSL tool).

- Here, using template space to delineate cortical/subcortical whole brain ROIs and then using "subject-specific brain anatomy" in native space for the amygdala ROIs seems not ideal. Given the size of the regions and inherent problem of accurate registration across spaces, how did the authors verify that ROIs were not indeed overlapping (e.g., hippocampus vs amygdala subregions)?

- The authors do not provide sufficient explanation for including some of the nuisance regressors (e.g. task related head motion, position of scanner table).

- Demographic information reported on page 3 is misleading. It appears that the age of the sample is between 40-69 when in fact Figure 7 indicates that the age across timepoints is between 48-81.

- The authors state that their choice of assessing "patterns of covariation" on page 5 "was a natural choice of method to evaluate a relationship between two rich variable sets". There is no reference nor an explanation of benefits over using different methods.

- The Harvard Oxford atlas covers 48 cortical and 21 subcortical structural areas. What were the 109 regions chosen by the authors. Which of those are bilateral?
- The lateralization analyses description was not clear to me. The authors state on page 7 that “how the 9 amygdala subregions in the left versus right hemisphere are differentially tied to that of 109 target brain regions” and “based exclusively on the left hemispheric amygdala subregions with the entire brain versus a PLSC solution that is exclusively based on the right hemispheric amygdala subregions with the entire brain”. That suggest to me that all bilateral/unilateral “whole brain” regions were used? The results description of the analyses however suggest that non-amygdala brain regions were also considered per hemisphere?
- Why did the authors chose anatomically defined amygdala ROIs instead of otherwise functionally defined ROIs by previous studies?

4. The manuscript fails to report how the authors addressed data quality confounds relevant to their analyses. Data quality assessments are especially relevant considering the size and location of the amygdala and its nuclei. For instance, how was head motion in T1 image quality or segmentation quality of amygdala subregion as well as cortical and subcortical ROIs assessed? The authors point to previous publications for some related aspects (e.g. page 4 second paragraph), but it is necessary to report exclusions due to data quality or related strategy in the main manuscript. There are many different tools for quality assessments of T1 images nowadays to address some of these aspects (e.g. MRIQC) that would have been especially relevant in assessing signal quality in the small (and thus prone to susceptibility effects) amygdala nuclei. Without these assessments the resulting findings could be severely influenced by these confounding factors.

5. I am not an expert in phenome wide-profiling so my insight into methodological soundness are limited. The approach seems similar to previous work of the group (Saltoun et al., 2022; Nat Hum Behaviour). I do regardless have some conceptual suggestions for improvements and some major concerns:

- The UKBiobank naïve reader would benefit from a more in depths descriptions of these phenotypes in the text (at last the 11 major categories).
- The authors use two strategies of multiple comparison correction (Bonferroni and FDR) but do not explain or describe differences between the results nor specify which methods’ outcome is then reported.
- Is there a cross-validation procedure that could show that these phenome wide associations are robust results that are replicable across independent samples (e.g. a- cross validation analyses, comparing to other larger samples (such as HPC)? Given the larger n (yet not in the range of 10k, see BWAS discussion –not addressed by the authors) it seems likely that even after multiple comparison corrections false positives may be present. This seems especially relevant given my methodological concerns regarding the noise in the brain measure.

6. Conceptually I do not follow the logic on page 11 about the robustness of the 6 modes. What is the standard to consider a result a “high explained variance”? Why are low/high numbers of phenotypic hits indicative of robustness? Relatedly, the authors say that they use the “phenome-wide analysis as a device to find real-world relevance of the uncovered principled signature” and argue “convincing real-world relevance” (notably, the variables with the highest significance threshold are “height” and “mother still alive” in the two modes with above threshold correction). Is the argument that if there is no significant (beyond multiple comparison correction) phenotype association, it is not considered relevant? How is noise in each of the measures (MRI, phenotypes) taken into account into a null finding?

7. The interpretation and conclusions drawn in the discussion about the functional relevance of anatomical co-variations and respective phenotypes appear post-hoc, selective and not well derived from a breadth of previous literature – instead a few single studies seem picked to fit a narrative. This concern relates back to my comment on the lack of hypotheses describing expected relationships within the context of prior literature.

Further Comments that I hope will be helpful to the authors to improve the manuscript:

a. Figures 1-5 show amygdala subregions but captions are not clearly stating the source of the anatomical images. It appears that these are taken from the Saygin et al paper, yet the authors do not state the original source and related copyright aspects. The B section of these figures also lack relevant details (left/right) and appear like a screenshot from a GUI (e.g. the colorbar selection that also misses the description of the measure).

b. It seems that the study is conceptually similar to at least one other study from the same group using similar methodology (e.g. relating a brain measure with phenome-wide association analyses). To limit the degrees of freedom in analytic choices, it would have been ideal for scientific rigor and replicability to preregister the analyses plan prior to conducting the analyses. The authors may want to comment on their choice to not preregister this and may want to outline which analytic decisions were made prior to observing results thereof.

c. It seems that the data and code for this study would not be made available to the scientific community. I understand that the input data is available. Open Sciences practices that foster reproducibility and replicability would suggest to also provide the scientific community with the extracted measures of interest (i.e. the volume measures per ROI), confound regressors and or at least the code to rerun the analyses (if in line with the data policies of the UK biobank) to reproduce said outputs.

d. The structuring of the information provided in the manuscript needs to be improved and cleaned from errors.

- Figure captions are redundant with information from the text and unusually detailed.

- The middle paragraph on page 26 contains the same section twice ("Our findings also...")

- The rationale for some analyses is stated in the results and not provided in detail for other analyses.

- It seems that the final conclusions of the paper are pointing to "past studies" (e.g., last sentence "Past studies further verify the roles of the central and medial nuclei in conscious awareness in all its forms, internal and external, as they have shown their roles in controlling fight or flight through noradrenaline regulation.")

e. Figure 7: The content of the figures is very hard to decipher – have the authors considered to re-order the data according to nuclei? Their main claim seems to be that specific nuclei change most over time and not necessarily which age group shows the most changes.

f. Comments on the supplementary data plan:

- It states that diffusion imaging was used. Respective results were not reported in the manuscript.

- Normalization template is labeled as "MNI152" but does not specify which one (e.g., linear, non-linear...).

- Merely relating to previous publications (e.g. Miller & Alfaro-Almagro) for relevant details on major methodological aspects is not sufficient.

Minor Comments (not an exclusive list):

PHEASANT vs PHEASANT (page 9)

Page 18: "(Figure?)"

What is a "plasticity twin"? (Page 7)

Providing a table for the subregion abbreviations seems unnecessary.

Reviewer #2 (Remarks to the Author):

Ghanem et al have presented an interesting paper examining structural MRI data from the UK Biobank to identify cortical/subcortical regions that have longitudinal changes (2 timepoints, adult data) that are associated with amygdala changes in grey matter. Using a partial least squares approach they identify three circuits between the amygdala and subcortical/cortical regions that have associated longitudinal changes. They examine the laterality specificity and also the association with a number of phenotypes including socioeconomic factors, and lifestyle factors. Overall the premise of the paper is interesting and creative, with what appears to be a data-driven approach to finding the relationship between two datasets (amygdala grey matter volume and cortical/subcortical grey matter volume). However, there are a number of issues, particularly with the interpretation, that I suggest the authors review below:

1. The introduction makes large generalizations and, at times assumptions, that are not referenced or well-supported. As the greatest example, lines 64-71 discuss changes in gray matter volume as reflecting changes in structural plasticity. This is a large assumption, as measurements from MRI are coarse, and cannot comment on cellular-level processes, and changes in structural MRI can be due to a number of influences, making it hard to pinpoint the exact influence that led to gray matter changes, as opposed to fMRI or EEG, in which a stimulusresponse measurement can more directly be measured. Changes in gray matter can be due to a number of influences that may include, but may also not be solely due to, plasticity. Furthermore, given that the ages of the participants are middle-aged or older, there are likely to be additional influences that likely cause the observed changes. Another example is line 54, stating that the amygdala has a role in carrying out risk assessment in initiating self-preserving modes of execution. The amygdala does have a role in fear learning, as is discussed by the authors, but it is not clear (nor referenced) what the role of the amygdala is in risk assessment—i.e, making judgements about the surroundings and decisions on how to behave. These are only a few examples. I would suggest that the authors provide a clear definition of plasticity, and thoroughly review the introduction for more thorough referencing, and specific descriptions throughout.
2. Similarly, in the introduction there are a number of confusing and oddly constructed sentences, ex: line 86, Richer datasets enable us to utilize the techniques and analytical methods that quantitatively revisit questions that we have been trying to answer since the one of the first recorded brain studies from the early periods of history (Mills and Tamnes, 2014). Or on lines 97, 98: Adding to the data wealth of the UKBiobank, we have exercised a granularity of 18 amygdala subregions (9 per hemisphere). Or lines 106-108, Instead, we here brought to bear phenome effects of various life factors on structural plasticity probed at the population level and across ~1,000 behaviour, lifestyle, and mental health indicators. These are a few examples, but numerous exist in the introduction as well as throughout the rest of the text, including the discussion. I would recommend that the authors review the full text for grammar.
3. Can the authors clarify the description of partial least squares canonical analysis? How is this different from canonical correlation analysis (CCA) or partial least squares correlation, for instance? Furthermore, can the authors provide a rationale for why this method was chosen over others?
4. Can the authors address whether they are powered to conduct the partial least squares canonical analysis?
5. The bootstrapping described in the methods seems to be the bare minimum to achieve a non-parametric distribution (100 iterations). More iterations (at least 1000) would lead to a distribution that is more likely to reflect the underlying distribution.
6. The references chosen are quite odd—for example, none of the foundational scientists examining primate amygdala neuroanatomy over the past few decades have been cited (David Amaral, Helen Barbás, Julie Fudge (one paper cited), Joe Price, Lynn Selemon and others).
7. The number of phenotypes created is very large—I am glad that the authors used a Bonferroni

correction.

8. Were there only 6 modes found?

9. The descriptions of the results are somewhat confusing, as it seems as though the authors have jumped to interpreting, rather than simply reporting, the results. Perhaps a better way to describe the findings are to describe which amygdala nuclei had the greatest parameter weights (to draw parallels to the figures), and then the associated covariance in the cortical areas, stating that this suggests a relationship between these regions. As the results are written currently, they jump to stating there are structural plasticity effects, which is really an interpretation, rather than a description of the results of their PLSC. It also makes the methods↯results relationship somewhat confusing. I would recommend that the authors stick to describing the results in the results, and bring in more interpretation in the discussion.

10. For Model 2, why is the Accessory Basal not discussed? It seems to have a relatively higher parameter weight.

11. Model 3 appears to be somewhat more generalized to the entire amygdala—the authors point out the subregions with relatively greater change, but really, the map here is strikingly different from the prior two maps, with all amygdala nuclei having changes in the same direction. Perhaps relatedly, the strength of association with the cortical areas is somewhat weaker compared to the prior models. It would be helpful if the authors discussed why this may be the case in this model.

12. In Figures 1-5, in the cortex, the heat scales that are set have a color for zero (blue or red, for instance), yet not every region of the cortex is colored. Can the authors provide an explanation for why not every region of the cortex is colored?

13. Can the authors provide an explanation for why they did two separate analyses for combined R/L amygdala and R and L amygdala separately? What is the advantage of doing these both? Why wouldn't the combined R/L amygdala analysis be the most comprehensive analysis? If the separate R/L amygdala analyses do not add much, the authors could consider whether to place these analyses in a supplement.

14. Please fix the figure callout on line 615.

15. As the authors have defined the circuits that have similar GM changes, it might be helpful to have a summary figure that depicts the three types of circuits/relationships from their analysis

16. I am not sure what to make of the phenotypic findings—could the authors provide some interpretation of the findings? What is the specificity of these findings to the different amygdala circuits? These are all associative findings, and it is hard to describe these as causative.

17. I found the discussion to be a bit unfocused. I would suggest tightening the sections on the salience network, as these are more established findings within the literature, and focusing the discussion on the circuits identified in the results.

18. The discussion continues to focus on plasticity, without clear evidence that the findings represent plastic changes. I would suggest that the authors focus on the circuits that they identify as having similar longitudinal changes, as these findings may not necessarily represent 'plastic' changes.

Reviewer #3 (Remarks to the Author):

The authors study the amygdala's temporal dynamics and its connections with cortical and subcortical areas. The authors contend that much of the research has been focused on the amygdala's role in fear recognition, often overlooking its contributions to memory, attention, decision-making, and other processes that are connected to cortico-subcortical circuits.

The study aims to map the changes that show co-variation between amygdala subregions and cortical partners, shedding light on the structural plasticity (longitudinal changes over years) of these interconnected brain areas. They perform an additional analysis linking these amygdala-cortical changes with lifestyle measures that reflect behavior and cognition, providing a broader understanding of the amygdala's function.

The study utilizes the UKBiobank initiative which was able to scan the brains of over 1,400 healthy

participants at two different timepoints, with a delay of 2-3 years between each, providing more authentic insights into amygdala-brain-behavior changes.

The study uses a detailed approach, analyzing 9 amygdala subregions and various behavioral, lifestyle, and mental health indicators. This represents a departure from past research, which often examined the amygdala as a single region or under three broader subdivisions. It is hoped that this more granular approach will allow for a better understanding of the amygdala's function and plasticity in response to various external stimuli and life factors.

Advantages:

Longitudinal Analysis: Longitudinal studies provide a significant advantage over cross-sectional studies in studying brain development, as they can track changes within the same individuals over time. This approach provides more reliable data about individual changes and could have greater potential for understanding causal relationships.

Large Sample Size: The study uses a large sample size of over 1,400 participants from the UKBiobank, which increases the reliability of the results and allows for the detection of smaller effects that may be missed in studies with fewer participants.

Detailed Analysis: The detailed analysis of 9 amygdala subregions is a significant step forward from the common approach of studying the amygdala as a whole or in a few broad subdivisions. This could provide new insights into the specific functions and connections of these subregions.

Connection to Lifestyle Factors: By connecting the analysis to around 1,000 behavioral, lifestyle, and mental health indicators, the study aims to provide a more holistic understanding of the brain's operations and adaptations in real-world contexts.

Potential Limitations:

Generalizability: While the large sample size is a strength, the participants are all from the UKBiobank. This could limit the generalizability of the findings to other populations, particularly those with different demographic characteristics or lifestyle factors.

Time Delay: The 2-3 year delay between brain scans could be both a strength and a limitation. While it allows for the detection of longitudinal changes, it may also miss more rapid changes that occur on a shorter timescale.

Structural Plasticity Measures: The use of gray matter volume as a measure of structural plasticity may not reflect the complexity of neural changes. It can capture large-scale alterations, but smaller scale, yet significant, changes might not be adequately represented.

Lack of Functional Data: While the focus on structural changes is important, it could be complemented by functional data to provide a more comprehensive picture of how these changes influence brain function and behavior.

Reliance on Self-Reported Measures: This study likely relies on self-reported measures for lifestyle and mental health indicators. These measures can be subject to bias and inaccuracies, which could impact the results.

Specifically, the individual figures need to be updated to reflect the statistical measures directly on the brain images to include the strength of associations across each brain region.

There are grammatical issues throughout the manuscript that need to be corrected. As a specific

example: "This hence provides more in-depth more information about the hemispherically-biased tendencies of the salience network uncovered in the primary analysis (cf. Figure 2) to interact with the salient amygdala subregions to regulate alertness and visual consciousness while also revealing a coupled association between the covarying amygdala subregions and the inferior parietal lobule."

The strength of associations to the behavioral/social measures is not indicated. This is a major problem and needs to be corrected/included.

Overall, while this study provides a detailed structural analysis of the amygdala as it correlates with (sub)cortical structures, I found the behavioral/social measures somewhat arbitrarily chosen. However, this is my personal bias and cannot be used against the authors.

More concerning, I believe the authors overextend their analysis with little evidence from their data. Given that these are associations, there can be no statement regarding causation. The discussion is far too long and speculative. I don't believe this type of gray matter analysis is enough to come to the conclusions regarding the role of the amygdala in intero-exteroception that the authors claim. I believe the authors should be more conservative in their discussion and conclusions and not venture too far from what the data are indicating.

Reviewer #1

The paper “Longitudinal microstructural changes in 18 amygdala nuclei resonate with cortical circuits and phenomics” investigated morphological changes of 18 bilateral amygdala subregions (9 per hemisphere) in relation to changes in 109 - and subcortical regions across the whole brain in a longitudinal design using MRI data from ~1400 subjects from the UK biobank. These patterns of volumetric changes between the amygdala subregion and the rest of the brain were then submitted to phenome-wide associations linking brain plasticity changes to a broad set of phenotypes. Characterizing the functional role of amygdala nuclei in humans is of great interest to the neuroscience community and using a large richly characterized sample across multiple timepoints seems like a potentially rich way to address the issue.

We are grateful to the reviewer for the positive feedback on our work.

1.The specific hypotheses of the study are not clearly stated. The introduction describes broadly what will be done, but fails to motivate what results may be expected given prior literature. If the study design is purely exploratory, it should be explicitly stated. The lack of hypotheses is in great contrast to the in depth interpretation of their findings in the discussion.

We thank the reviewer for this helpful point regarding the lack of hypotheses to support the interpretation of our findings. In response to the reviewer’s comment, we have amended the introduction section as follows:

Introduction:

Previous literature has associated the three larger umbrella groups, the laterobasal, centromedial and superficial subdivisions, to a variety of neurocognitive processes. These subdivisions have been found through invasive studies to work with other (sub)cortical regions to accomplish the tasks. In our study, we believed that we can increase analysis of subregional anatomical specificity within the amygdala subdivisions by using a tailored set of analyses with high-resolution and high-quality brain-imaging measurements of the amygdala. The analyses conducted within our study are expected to reveal the plasticity effects from subregional amygdala interplay with (sub)cortical regions in a way that shows relationships to behavioral traits at an unprecedented subregional resolution in the amygdala. The amygdala subregional-(sub)cortical associations that we expect to reveal in our analyses can help us trace a relationship between various brain networks and specific amygdala subregions that cooperate to regulate various tasks within the body.

At the outset of our study, we hypothesized to find which subregions in the laterobasal larger subdivision to be experiencing the most plasticity effects that covary with regions in the prefrontal cortex. We also expected to find which subregions in the centromedial and superficial larger subdivisions would undergo the most covarying plasticity with the (sub)cortical regions responsible for initiating autonomic responses and social cognition. In addition, the (sub)cortical regions that are responsible for receiving and processing external sensory stimuli with their ensuing consequences were expected to show the most explanatory structural changes with specific subregions in the centromedial and superficial subdivisions.

We expected lateralization effects to be observed in the brain regions preferentially responsible for social cognition and brain regions responsible for receiving and processing external sensory stimuli, in relation to left-right deviation changes in the superficial and centromedial larger subdivisions. Lateralization of plasticity covariance in the centromedial and the superficial amygdala subregions was also expected to occur with lateralizing plasticity covariance in brain regions related to conscious awareness. We expected that this is the case given that specific subdivisions in the amygdala have been found in past brain-imaging studies to have unique lateralization patterns (Tonio et al., 2007). We hypothesized that the same (sub)cortical brain regions that happen to experience covarying plasticity with amygdala subregions are good candidates to also lateralize in their plasticity effects with these amygdala subregions.

We also hypothesized at the outset that indicators related to socioeconomic status and related to contributors to mental health, will be revealed by our phenome-wide association assays at the population level. Past studies have shown that different stress and health implications are associated with stable and unstable social hierarchies, and the study investigates how neural responses differ between these two contexts (F. Zink et al., 2008). Unstable social hierarchies elicited unique neural responses, such as increased activity in areas linked with social emotional processing and social cognition, particularly when viewing a superior player. The amygdala, which is known for processing socially emotional stimuli and social anxiety related to hierarchical challenges, showed increased activity in unstable social hierarchies (F. Zink et al., 2008). The thus disclosed amygdala subregion-brain network correspondence is expected to show robust links to a variety of broader phenotypes such as those related to regulating bodily affective states.

References:

- Ball T, Rahm B, Eickhoff SB, Schulze-Bonhage A, Speck O, Mutschler I. Response properties of human amygdala subregions: evidence based on functional MRI combined with probabilistic anatomical maps. *PLoS One*. 2007 Mar 21;2(3):e307. doi: 10.1371/journal.pone.0000307. PMID: 17375193; PMCID: PMC1819558.
- Zink CF, Tong Y, Chen Q, Bassett DS, Stein JL, Meyer-Lindenberg A. Know your place: neural processing of social hierarchy in humans. *Neuron*. 2008 Apr 24;58(2):273-83. doi: 10.1016/j.neuron.2008.01.025. PMID: 18439411; PMCID: PMC2430590.

2. Relatedly, it should be clearly stated that the age range of this dataset and thus brain morphological changes are not in the young adult range but more towards the changes in the brain related to aging. The introduction and most of the discussion is not emphasizing that sufficiently; the abstract does not report the age range of the sample. The presented conclusions of their findings would be limited towards the middle and end of the lifespan.

We thank the reviewer for this helpful point with regards to being clear about the age range of the dataset. We have amended the abstract, introduction and the discussion to better render explicit the age range of the different parts of the dataset as follows:

Abstract: Here we present amygdala nuclei morphometry and behavioural findings from longitudinal population data (> 1,400 subjects, **age range 40-69 years**, sampled 2-3 years apart): the UK Biobank offers exceptionally rich phenotyping together with brain morphology scans.

Introduction: To start filling this gap of knowledge, longitudinal studies – coming into reach due to recently emerged data resources - now allow the probing of aspects that underpin structural plasticity and enable certain statements with causal implications by examining the changes occurring within the brain of the same individual across time, while recording the variety of life circumstances of the participants sample **in the middle and at the end of their lifespan.**

Introduction: Past longitudinal studies have suffered from low numbers of participants. The UKBiobank initiative was able to obtain longitudinal brain scanning on >1,400 healthy participants **of ages 40–69 years** from two different timepoints. Additionally, previous longitudinal studies on structural covariation were also limited by narrow time windows – typically days to several weeks – between the data acquisition time points.

Discussion: We then profiled the derived population-level plasticity patterns using a rich palette of ~1,000 phenotypical indicators. The characterization has tied the structural amygdala-brain couplings to various phenotypes such as social status, employment, sleep habits, risk taking and leisure activities. **Our analyses were performed on participants in the middle and at the end of their lifespan.**

Discussion: In our phenome-wide analysis, socioeconomic status indicators, such as household income and employment status, were found to explain the coupling relationships between amygdala nuclei and whole-brain regions. **The phenotypic indicators of high correlation happen to be more oriented towards the middle and the end of a lifespan given the age range of the participants sample.** These plasticity-phenotypic associations have additionally highlighted inter-personal relationship phenotypes related to the social well-being and family relationships to contribute heavily to the longitudinal links of the medial, cortical, and central amygdala nuclei

with coordinated changes in the salience network and the MT+/V5 that we observed to occur over years.

Along the same lines, the hemispheric difference analysis is not well (??) motivated. Why did the authors consider to investigate a “systematic hemisphere difference in the limbic system” with respect to amygdala subregion functional change across age?

We thank the reviewer for this helpful point with regards to being clear about the motivation of our hemispheric difference analysis. We have amended the introduction (cf. above) with our hypothesis about why we think it is important to investigate the systematic hemisphere difference in the amygdala.

3. The description of methodological procedures lack clarity, contains errors and is not adequately accounting for bias in template choice.

- My most severe concern is in the design choice to address the longitudinal data set. I assume the authors used FreeSurfer (at least in part, see comment below) to assess volumetric changes. FreeSurfer has the option of using a longitudinal processing stream which addresses the problem of biasing the analyses with respect to the first timepoint. Regardless of choice of processing tools, there is always some level of random variation in the processing procedure (e.g. algorithm initialization) that affects robustness and sensitivity of the overall longitudinal analysis. Using the output of a cross-sectional approach for both timepoints is thus unnecessarily increasing variability of results across timepoints and may be severely impacting the robustness of the results.

While the reviewer has brought up a valid point, we found that ensuring the comparability of our approach with other UKbiobank studies in the future is crucial. This is why we found an alternative way of achieving robust findings in our results in which comparability to future studies is maintained as well.

Please note, technically, that the hippocampus Freesurfer outputs had already been computed and quality controlled by the Oxford MRIB team at the beginning of our study. As such, our research project did not involve any new Freesurfer outputs computed on our own. By building on the expert-curated Freesurfer information, our findings will be more directly comparable to those of other scientists and clinicians.

Furthermore, neurobiologically, some of the amygdala subregion-(sub)cortical region associations revealed by the analysis in our study have corroborated some well known facts already established in the neuroscience literature. In the third revealed pattern, our analysis revealed that the covariation of the structural change in the basal, lateral, accessory basal, and paralaminar subregions and the prefrontal cortex were found to strongly covary in the same

direction in this pattern which indicated that the laterobasal subregions and the prefrontal cortex share coupled relationships. Indeed, a number of past imaging studies and invasive rat studies have shown direct and indirect projections between the laterobasal larger subdivision and the prefrontal cortex (Aggleton et al., 1979; Singh et al., 2015). Another example of our reproduced findings is how our study further confirmed another relationship that has already been unveiled in a past study. Our analysis showed covariation between the amygdala central nucleus and the brainstem in line with a past animal study has shown that the central nucleus sends efferents to a variety of brainstem regions, as shown in invasive studies in rats (Wallace et al., 1991).

For empirical validation of the stability of our results, our analysis yielded 6 modes that explained the degree of joint structural covariation between change events in the 18 amygdala subregions and change events in the 109 cortical and subcortical regions. We evaluated the robustness of the obtained 6 modes using three different approaches. The first approach was based on explained variance of each mode, by finding the Pearson rho value between the covariation patterns of the amygdala subregions and the covariation patterns of the (sub)cortical regions in every mode. By doing so, we are able to determine how much the structural changes in the amygdala subregions are correlated to those in the (sub)cortical regions in the participant sample. Based on obtaining a high significance of the rho values between the change events in the amygdala subregions and the change events of the (sub)cortical regions, we are able to confirm the robustness of the modes produced by our analysis.

Second, while testing the strength of the modes using the Pearson correlation rho does reveal the strength of the significance of covariation between the longitudinal change in the amygdala subregions and longitudinal change in the (sub)cortical regions, we further wanted to check the robustness by examining the behavior of the association between the given phenotypes and the inter-individual covariance of the coupling over time between the amygdala subregions and the (sub)cortical regions in each of the key modes. By doing so, we are able to discover what behavioral traits are linked to the coupling of the amygdala-cortex covariation regimes. Phenotype associations above the FDR threshold in the modes indicate a more significant interplay between the amygdala subregions and the (sub)cortical regions associated with the phenotype of significance. The results of the phenome-wide analysis showed no significant phenotype associations above the FDR threshold in the fourth, fifth and sixth pattern while 34 phenotypes, 31 phenotypes and 79 phenotypes exceeded the Bonferroni threshold the first, second and third patterns respectively. Thus, we reduced the 6 significant patterns to 3 modes of covariation with convincing relevance as evidenced by their ~1,000 diverse phenotype profiles.

Our third and last acid test for ensuring robustness entails an empirical permutation testing framework. To quantify every mode's statistical significance, we evaluated the explained variance for all the derived unique components and found the significance of the modes using their calculated Pearson rho values. 100 permutation iterations were carried out, where in each permutation iteration, the significance of the correlation coefficients obtained from the modes, are assessed by generating randomized permuted versions of the (sub)cortical regions' plasticity covariation patterns, fitting the model with the resultant permuted versions of the (sub)cortical regions' plasticity covariation patterns and the amygdala subregional plasticity covariation patterns, and then calculating the correlation coefficients and scores of each mode, yielding an

empirical null distribution. The results are then compared to the original observed explained variance performances from the original version of the model fitted with unpermuted data to compute p-values. The p-values are used to present the resultant significance of each mode in the model which helps create an apple-to-apple statistical comparison of the significance between all the modes. The leading 3 plasticity patterns, found to be the most significant through our phenome-wide analysis, also showed the strongest joint adaptation effects between amygdala and the cortex out of the 6 original patterns, with p-values < 0.005 according to our permutation test.

References:

Aggleton JP, Burton MJ, Passingham RE. Cortical and subcortical afferents to the amygdala of the rhesus monkey (*Macaca mulatta*). *Brain Res.* 1980 May 26;190(2):347-68. doi: 10.1016/0006-8993(80)90279-6. PMID: 6768425.

Singh MK, Kelley RG, Chang KD, Gotlib IH. Intrinsic Amygdala Functional Connectivity in Youth With Bipolar I Disorder. *J Am Acad Child Adolesc Psychiatry.* 2015 Sep;54(9):763-70. doi: 10.1016/j.jaac.2015.06.016. Epub 2015 Jul 6. PMID: 26299298; PMCID: PMC4548854.

Wallace DM, Magnuson DJ, Gray TS. Organization of amygdaloid projections to brainstem dopaminergic, noradrenergic, and adrenergic cell groups in the rat. *Brain Res Bull.* 1992 Mar;28(3):447-54. doi: 10.1016/0361-9230(92)90046-z. PMID: 1591601.

- In the top of page 5 the authors state that “SIENAX was used to derive volumetric measures normalized for head size”. In the following paragraph they state to use the FreeSurfer tool but cite FSL (Jenkinson et al., 2012). The reference provided for the segmentation (and validation thereof) of amygdala nuclei is also incorrect (again stating an FSL tool).

We thank the reviewer for this important pointer. We have corrected this recurrent mistake in our revised version of the manuscript as follow:

Volume measures from 18 amygdala subregions (9 per hemisphere) were extracted taking into account subject-specific brain anatomy based on FreeSurfer subsegmentation (Saygin ZM & Kliemann D et al., 2017).

The automatic volumetric segmentation of the amygdala using Freesurfer has been successfully evaluated to yield an accurate parcellation of the 18 amygdala subregions (Saygin ZM & Kliemann D et al., 2017).

Figure 1. Within-subject structural plasticity effects in the central nucleus and anterior amygdaloid area covary especially with the inferior parietal lobule. Principle patterns of structural covariation due to longitudinal plasticity between gray matter volume change over time in 18 microanatomically distinct amygdala subregions (9 per hemisphere) (Saygin ZM & Kliemann D et al., 2017).

References:

Saygin ZM & Kliemann D (joint 1st authors), Iglesias JE, van der Kouwe AJW, Boyd E, Reuter M, Stevens A, Van Leemput K, Mc Kee A, Frosch MP, Fischl B, Augustinack JC., 2017. High-resolution magnetic resonance imaging reveals nuclei of the human amygdala: manual segmentation to automatic atlas. *Neuroimage*, 155, 370-382.

- Here, using template space to delineate cortical/subcortical whole brain ROIs and then using “subject-specific brain anatomy” in native space for the amygdala ROIs seems not ideal. Given the size of the regions and inherent problem of accurate registration across spaces, how did the authors verify that ROIs were not indeed overlapping (e.g., hippocampus vs amygdala subregions)?

We thank the reviewer for bringing up this appreciated point about the nature of the amygdala ROIs. In fact, we did not use a subject specific brain atlas for the amygdala. In contrast, we also used an analysis that departs from a generic template space that delineates amygdala subregion ROIs, analogous for the cortical atlas used in our present investigation. While tools for delineating the amygdala subcompartments in the living human brain are few, the Freesurfer team has created an atlas by scanning postmortem specimens at high resolutions to visualize and label the nine amygdala nuclei per hemisphere. This consensus atlas (Saygin ZM & Kliemann D et al., 2017) was built using an algorithm based on Bayesian inference as a part of Freesurfer to automatically segment nine amygdala nuclei (in each hemisphere) from a given subject’s structural MR image.

More specifically, Freesurfer’s amygdala template atlas was tested by applying it onto two publicly available population datasets (before the emergence of the UK Biobank Imaging population dataset): ADNI and ABIDE with standard resolution T1 data by using individual volumetric data of the amygdala nuclei as the measure and found that the atlas achieves high accuracy when used to classify between individuals with Alzheimer’s disease and older adult controls and to classify between individuals with autism and age-matched controls. In the application of the new approach to delineating the amygdala subregions, the new atlas was able to outperform previously proposed versions of the amygdala atlas. Our present study directly benefited from these recent advancements in probabilistic modeling (Bayesian modeling) in order to parcellate the nuclei of the amygdala. The approach also takes into account the individual underlying anatomy which provides greater spatial sensitivity and hence aims to reduce overlap between the regions (Saygin ZM & Kliemann D et al., 2017).

References:

Saygin ZM & Kliemann D (joint 1st authors), Iglesias JE, van der Kouwe AJW, Boyd E, Reuter M, Stevens A, Van Leemput K, Mc Kee A, Frosch MP, Fischl B, Augustinack JC., 2017. High-resolution magnetic resonance imaging reveals nuclei of the human amygdala: manual segmentation to automatic atlas. *Neuroimage*, 155, 370-382.

- The authors do not provide sufficient explanation for including some of the nuisance regressors (e.g. task related head motion, position of scanner table).

We thank the reviewer, and we will add this more detailed information on nuisance handling in the revised version of the manuscript.

During our preparatory data-cleaning step, the variation in each of our brain region volumes that could be explained by variables outside of scientific interest and by potential confounding influences were regressed out. Some of the factors mentioned (eg. age) can be considered a confound to be removed depending on the questions being asked in a study and some factors (eg. head motion) can be considered a confound to be removed due to their relationship to the type of specific dataset such as the way the data was collected (fMRI etc.) - these head motion indicators have been chosen by the UK Biobank Imaging team itself, even for structural MRI analyses (Alfaro-Almagro et al., 2018; Miller et al., 2016). A past imaging study has shown that taking into account the confounds relevant to the specific dataset can reveal various hidden associations between the variables of interest due to partially shared variance (Miller et al., 2016). Apart from these earlier occurrences, research in our own lab routinely involves deconfounding based on these variables (Kiesow et al., 2020; Spreng et al., 2020; Saltoun et al., 2023).

References:

Alfaro-Almagro F, Jenkinson M, Bangerter NK, Andersson JLR, Griffanti L, Douaud G, Sotiropoulos SN, Jbabdi S, Hernandez-Fernandez M, Vallee E, Vidaurre D, Webster M, McCarthy P, Rorden C, Daducci A, Alexander DC, Zhang H, Dragonu I, Matthews PM, Miller KL, Smith SM. Image processing and Quality Control for the first 10,000 brain imaging datasets from UK Biobank. *Neuroimage*. 2018 Feb 1;166:400-424. doi: 10.1016/j.neuroimage.2017.10.034. Epub 2017 Oct 24. PMID: 29079522; PMCID: PMC5770339.

Miller KL, Alfaro-Almagro F, Bangerter NK, Thomas DL, Yacoub E, Xu J, Bartsch AJ, Jbabdi S, Sotiropoulos SN, Andersson JL, Griffanti L, Douaud G, Okell TW, Weale P, Dragonu I, Garratt S, Hudson S, Collins R, Jenkinson M, Matthews PM, Smith SM. Multimodal population brain imaging in the UK Biobank prospective epidemiological study. *Nat Neurosci*. 2016 Nov;19(11):1523-1536. doi: 10.1038/nn.4393. Epub 2016 Sep 19. PMID: 27643430; PMCID: PMC5086094.

Kiesow H, Dunbar RIM, Kable JW, Kalenscher T, Vogeley K, Schilbach L, Marquand AF, Wiecki TV, Bzdok D. 10,000 social brains: Sex differentiation in human brain anatomy. *Sci Adv.* 2020 Mar 18;6(12):eaaz1170. doi: 10.1126/sciadv.aaz1170. PMID: 32206722; PMCID: PMC7080454.

Spreng RN, Dimas E, Mwilambwe-Tshilobo L, Dagher A, Koellinger P, Nave G, Ong A, Kernbach JM, Wiecki TV, Ge T, Li Y, Holmes AJ, Yeo BTT, Turner GR, Dunbar RIM, Bzdok D. The default network of the human brain is associated with perceived social isolation. *Nat Commun.* 2020 Dec 15;11(1):6393. doi: 10.1038/s41467-020-20039-w. Erratum in: *Nat Commun.* 2021 May 21;12(1):3202. PMID: 33319780; PMCID: PMC7738683.

Saltoun K, Adolphs R, Paul LK, Sharma V, Diedrichsen J, Yeo BTT, Bzdok D. Dissociable brain structural asymmetry patterns reveal unique phenome-wide profiles. *Nat Hum Behav.* 2023 Feb;7(2):251-268. doi: 10.1038/s41562-022-01461-0. Epub 2022 Nov 7. PMID: 36344655.

- Demographic information reported on page 3 is misleading. It appears that the age of the sample is between 40-69 when in fact Figure 7 indicates that the age across timepoints is between 48-81.

We thank the reviewer for pointing out a source of potential confusion for future readers when mentioning the age ranges in the previous version of our manuscript. Here are the key details for clarification: About 500,000 participants, recruited when aged between 40 and 69, went through a baseline assessment involving the collection of blood, urine, and saliva samples (not brain-imaging). Samples for genetic analysis and physical measurements were also taken, and each individual answered an extensive questionnaire focused on aspects of health and lifestyle. A few years later, a relevant fraction of the original baseline subject cohort were invited for the imaging data collection arm of the study, 1,414 of them being the dataset that provided the basis for this study. 2-3 years later the 1,414 participants came back for another imaging visit. This explains why we have time points for participants measured across an age range of 48-81 in Figure 7.

In response to this helpful reviewer feedback, we have now more clearly spelled out this difference in the mentioned age brackets in the specific places in the improved manuscript version.

Methods:

The present study was based on the recent release from February 2020 that provides data from ~40,000 participants with brain-imaging measures and expert-curated image-derived phenotypes of gray matter morphology (T1-weighted MRI) from 48% men and 52% women, aged 40–69 years when recruited (mean age 55 years, standard deviation (SD) 7.5 years). **A few years after recruitment, a relevant fraction of the original baseline subject cohort were invited for the imaging data collection arm of the study, 1,414 of them being the dataset that**

provided the basis for this study. 2-3 years later the 1,414 participants came back for another imaging visit. At that point, the age range of the 1,414 participants ranged between 48-81. We attempted to improve comparability and reproducibility in our study by building on the uniform data preprocessing pipelines designed and carried out by FMRIB, Oxford University, UK (Alfaro-Almagro et al., 2018). All participants provided informed consent with information on the participant consent process being openly disclosed (<http://biobank.ctsu.ox.ac.uk/crystal/field.cgi?id=200>).

References:

Alfaro-Almagro F, Jenkinson M, Bangerter NK, Andersson JLR, Griffanti L, Douaud G, Sotiropoulos SN, Jbabdi S, Hernandez-Fernandez M, Vallee E, Vidaurre D, Webster M, McCarthy P, Rorden C, Daducci A, Alexander DC, Zhang H, Dragonu I, Matthews PM, Miller KL, Smith SM. Image processing and Quality Control for the first 10,000 brain imaging datasets from UK Biobank. *Neuroimage*. 2018 Feb 1;166:400-424. doi: 10.1016/j.neuroimage.2017.10.034. Epub 2017 Oct 24. PMID: 29079522; PMCID: PMC5770339.

- The authors state that their choice of assessing “patterns of covariation” on page 5 “was a natural choice of method to evaluate a relationship between two rich variable sets”. There is no reference nor an explanation of benefits over using different methods.

We thank the reviewer and we have added dedicated language as to this point in the revised version of the manuscript. To elaborate our reasons here, to respond to this reviewer query:

- 1) We have quite a few variables per observation in our modeling scenario (cf. methods section); we can hence probably not afford more than a linear model, as opposed to a model class that would be capable of capturing higher-order non-linearities. It is difficult for most non-linear methods to allow for interpreting weights for understanding the profiles of neural and behavioral features carrying the brain-behavior relationship with such a high number of features and low number of observations. On the other hand, the PLSC has previously been carefully empirically evaluated to yield stable and accurate results with a large number of considered features obtained within the space of UKBiobank-level sample size (Helmer et al., 2020).
- 2) In the case of the brain-imaging data used, autocorrelation tends to be very prominent aspects within these datasets, which should be reflected in what analysis tools are exactly chosen for a modeling task. Our method is not only able to handle, but actually thrives on and explicitly quantifies the autocorrelation pattern within an input data matrix (Wang et al., 2020) as part of solving an input-output modeling problem.

- 3) Our method is effective for reducing a large set of related variables to a typically smaller number of combinations or latent variables that capture most of the variation of the original set. Hence, given a collection of vectors, we are able to produce a derived set of uncorrelated features. Reducing high-dimensional variable sets to its essence by using the dimensionality-reduction capability of our method is an ideal solution in the case of an expected coexisting, spatially overlapping brain structure covariation pattern. The UK Biobank dataset of brain region volumes happens to be a high-dimensional dataset with a relatively low input-features-to-sample size ratio. Additionally, its features are not necessarily most informative when they are analyzed individually and so we expected the covariation in the brain regions to have an underlying topographical overlap. Hence making use of the dimensionality-reduction property of our method is suitable for the dataset we are using (Bzdok et al., 2019).

We have added these explanations in the revised version of the manuscript as follows:

Methods:

Partial least squares canonical analysis (PLSC) was a natural choice of method to evaluate a relationship between two rich variable sets. **This model class was ideally fitted to our data analysis scenario on grounds of i) feature-to-samples ratio, ii) native auto-correlation in our variable sets with brain-derived measurements, and iii) the latent-factor decomposition capability.**

References:

Helmer M, Warrington S, Mohammadi-Nejad AR, Ji JL, Howell A, Rosand B, Anticevic A, Sotiropoulos SN, Murray JD. On stability of Canonical Correlation Analysis and Partial Least Squares with application to brain-behavior associations.. doi: <https://doi.org/10.1101/2020.08.25.265546>

Wang HT, Smallwood J, Mourao-Miranda J, Xia CH, Satterthwaite TD, Bassett DS, Bzdok D. Finding the needle in a high-dimensional haystack: Canonical correlation analysis for neuroscientists, *NeuroImage*, 216, 2020.

Bzdok D, Nichols TE, Smith SM. Towards Algorithmic Analytics for Large-scale Datasets. *Nature Machine Intelligence*, 1:296-306, 2019.

- The Harvard Oxford atlas covers 48 cortical and 21 subcortical structural areas. What were the 109 regions chosen by the authors. Which of those are bilateral?

The calculation of the 109 regions used in our study is derived from both cortical and subcortical areas, specifically: $109 = 96 \text{ cortical regions} + 13 \text{ subcortical regions}$.

For clarity:

Cortical Regions: We utilized all 48 UKBiobank-precomputed cortical regions, implementing them bilaterally, which results in 96 regions (48 regions x 2 for bilateral representation).

Subcortical Regions: The Harvard Oxford atlas denotes 21 subcortical regions. All are typically bilateral except for the brainstem; however, we excluded several and utilized only 13.

Specifically:

Excluded: Left Cerebral White Matter, Right Cerebral White Matter, Left Cerebral Cortex, Right Cerebral Cortex, Left Lateral Ventricle, Right Lateral Ventricle, Left Amygdala, and Right Amygdala (since the amygdalae were under investigation).

Included: The remaining 13 subcortical regions, with all but the brainstem represented bilaterally.

In sum, the study utilized 96 cortical and 13 subcortical regions, totaling 109 regions for the investigation.

- The lateralization analyses description was not clear to me. The authors state on page 7 that “how the 9 amygdala subregions in the left versus right hemisphere are differentially tied to that of 109 target brain regions” and “based exclusively on the left hemispheric amygdala subregions with the entire brain versus a PLSC solution that is exclusively based on the right hemispheric amygdala subregions with the entire brain”. That suggests to me that all bilateral/unilateral “whole brain” regions were used? The results description of the analyses however suggest that non-amygdala brain regions were also considered per hemisphere?

We thank the reviewer, and we will add this note in the revised version of the manuscript. We have conducted a PLSC analysis of only the left 9 amygdala subregions in relationship with all the rest of the brain regions, and then did the same kind of analysis with only the right 9 amygdala subregions with their covariation with the brain. The steps taken helped us find the separate covariation effects between the 9 left amygdala subregions and the 109 (sub)cortical regions separately and then find the separate covariation effects between the 9 right amygdala subregions and the 109 (sub)cortical regions respectively. The left- and right-centric model fits, and their direct comparison to interrogate any hemisphere-sensitive covariation effects, were directly compared with each other, which helped reveal the specific lateralization effects of amygdala subregions brain covariation.

We have refined results description of the analyses for clarification, based on this valuable reviewer feedback:

Results: In one model instance, we estimated the covariation of the amygdala subregions in the left hemisphere with the entire brain, and we then, in a second model instance, estimated the covariation of the amygdala subregions in the right hemisphere with the entire brain in the second model instance. **We aimed to identify which anatomical subregions show statistically defensible deviation between i) how the left hemisphere amygdala covaries with the brain and ii) how the right hemisphere amygdala and the brain.** Our examination of the number of significant patterns in our primary analysis also determined which of the signatures in the hemispheric difference analysis were significant by using the phenome-wide analysis to explore the biological pertinence of each of the candidate patterns through the elimination of the patterns with no phenotypic correlation magnitudes above the FDR threshold when compared to the gray matter volume in the amygdala subregions at the first time point, analogous to our primary analysis (cf. above).

- Why did the authors choose anatomically defined amygdala ROIs instead of otherwise functionally defined ROIs by previous studies?

Freesurfer 18-subregion resolution decomposition of the amygdala has not been implemented by this tool or another software backed on functional MRI data, which is why we had no other choices.

As such, all our atlases - the Freesurfer derived amygdala compartments and the cortical/subcortical derived compartments - were all structurally defined and coherent across our entire analysis pipelines.

4. The manuscript fails to report how the authors addressed data quality confounds relevant to their analyses. Data quality assessments are especially relevant considering the size and location of the amygdala and its nuclei. For instance, how was head motion in T1 image quality or segmentation quality of amygdala subregion as well as cortical and subcortical ROIs assessed? The authors point to previous publications for some related aspects (e.g. page 4 second paragraph), but it is necessary to report exclusions due to data quality or related strategy in the main manuscript. There are many different tools for quality assessments of T1 images nowadays to address some of these aspects (e.g. MRIQC) that would have been especially relevant in assessing signal quality in the small (and thus prone to susceptibility effects) amygdala nuclei. Without these assessments the resulting findings could be severely influenced by these confounding factors.

We thank the reviewer for this helpful point with regards to clarity about the data quality confounds relevant to the analysis. We have clarified how we regressed out the potential

confounding influences, including head-motion-related confounds, and explained how we ensured segmentation quality of the ROIs (cf. above).

Neurobiologically, again, our results show several cases of direct reflections of established knowledge from invasive animal studies, attesting to the quality of our T1-derived input data.

5. I am not an expert in phenome wide-profiling so my insight into methodological soundness are limited. The approach seems similar to previous work of the group (Saltoun et al., 2022; Nat Hum Behaviour). I do regardless have some conceptual suggestions for improvements and some major concerns:

- The UKBiobank naïve reader would benefit from a more in depths description of these phenotypes in the text (at last the 11 major categories).

We thank the reviewer for the helpful suggestion. The Manhattan plots summarize phenome-wide association studies by providing an interpretational framework which facilitates the ability to assess brain-behaviour associations across 11 broad phenotypic domains that classify 977 Biobank phenotype variables (Miller et al., 2016). We chose to home in on specific examples in the text in order to illustrate wider phenomena. As a specific example, the third pattern of the intrinsic plasticity coupling analysis had phenotype associations related to income, work status, sleep, risk taking, and leisure regular activities exist above the Bonferroni correction threshold. Hence, we chose to summarize the revealed associations to be related to phenotypes that are contributors to mental health and to socioeconomic status.

For the interested reader, we will publish our code onto a github repository containing the full set of results for all intrinsic plasticity coupling patterns. Included with each pattern will be the association and significance levels for all 977 phenotypes. Therefore, any reader interested in more carefully investigating the direction of effects is welcomed to do so. We will include a link to the UKBiobank data showcase (<https://biobank.ndph.ox.ac.uk/ukb/>) that has detailed explanations of every phenotype part of our Phewas analysis.

References:

Miller, K.L., Alfaro-Almagro, F., Bangerter, N.K., Thomas, D.L., Yacoub, E., Xu, J., Bartsch, A.J., Jbabdi, S., Sotiropoulos, S.N., Andersson, J.L.R., Griffanti, L., Douaud, G., Okell, T.W., Weale, P., Dragonu, I., Garratt, S., Hudson, S., Collins, R., Jenkinson, M., Matthews, P.M., Smith, S.M., 2016. Multimodal population brain imaging in the UK Biobank prospective epidemiological study. *Nature Neuroscience* 19, 1523-1536.

We have included a table for each of the patterns of our phenome-wide association studies in our Supplementary Material that has the 11 major categories and all phenotype effect

associations for each of the domains that lies above the Bonferroni correction threshold. All these phenotype associations exceeded the Bonferroni correction threshold.

Phenotype-wide association studies analysis table of the first pattern:

Supplementary Table 1. Phenotype-wide association studies analysis table of significant hits (from largest to smallest hit association magnitude) above the Bonferroni correction threshold of the first pattern. The Magnitude - Phenotype column has the magnitude of the association between the phenotypes and the amygdala subregion-(sub)cortical region covariance which are shown in units as $-\log_{10}$ of the p-values (left) and the name of the phenotype (right). The Category column shows under which of the 11 broad domains the phenotype lies.

Pattern 1			
Magnitude - Phenotype	Category	Magnitude - Phenotype	Category
11.0 - Leg fat percentage (right) (0.0)	PHYSICAL MEASURES - GENERAL	5.81 - Maximum carotid IMT (intima-medial thickness) at 240 degrees (2.0)	PHYSICAL MEASURES - CARDIAC & BLOOD VESSELS
8.84 - Leg fat mass (right) (0.0)	PHYSICAL MEASURES - GENERAL	5.8 - Hand grip strength (left) (0.0)	PHYSICAL MEASURES - GENERAL
8.67 - Body fat percentage (0.0)	PHYSICAL MEASURES - GENERAL	5.57 - Red blood cell (erythrocyte) count (0.0)	BLOOD ASSAYS
8.59 - Impedance of whole body (0.0)	PHYSICAL MEASURES - GENERAL	5.52 - Minimum carotid IMT (intima-medial thickness) at 240 degrees (2.0)	PHYSICAL MEASURES - CARDIAC & BLOOD VESSELS
8.18 - Impedance of arm (right) (0.0)	PHYSICAL MEASURES - GENERAL	5.43 - Impedance of leg (right) (0.0)	PHYSICAL MEASURES - GENERAL
8.02 - Arm fat percentage (left) (0.0)	PHYSICAL MEASURES - GENERAL	5.32 - Height (2.0)	PHYSICAL MEASURES - GENERAL
7.91 - Arm fat-free mass (left) (0.0)	PHYSICAL MEASURES - GENERAL	5.29 - Haematocrit percentage (0.0)	BLOOD ASSAYS
7.77 - Arm predicted mass (left) (0.0)	PHYSICAL MEASURES - GENERAL	5.23 - Gamma glutamyltransferase (0.0)	BLOOD ASSAYS
7.66 - Impedance of arm (left) (0.0)	PHYSICAL MEASURES - GENERAL	5.17 - Haemoglobin concentration (0.0)	BLOOD ASSAYS
7.55 - Whole body fat-free mass (0.0)	PHYSICAL MEASURES - GENERAL	5.11 - Sitting height (0.0)	PHYSICAL MEASURES - GENERAL
7.48 - Arm fat percentage (right) (0.0)	PHYSICAL MEASURES - GENERAL	5.0 - Urate (0.0)	BLOOD ASSAYS
7.27 - Mean carotid IMT (intima-medial thickness) at 240 degrees (2.0)	PHYSICAL MEASURES - CARDIAC & BLOOD VESSELS	4.75 - Mean carotid IMT (intima-medial thickness) at 120 degrees (2.0)	PHYSICAL MEASURES - CARDIAC & BLOOD VESSELS
7.2 - Arm fat-free mass (right) (0.0)	PHYSICAL MEASURES - GENERAL	4.69 - Creatinine (0.0)	BLOOD ASSAYS
6.81 - Leg fat-free mass (right) (0.0)	PHYSICAL MEASURES - GENERAL	4.65 - Impedance of leg (left) (0.0)	PHYSICAL MEASURES - GENERAL
6.69 - Hand grip strength (right) (0.0)	PHYSICAL MEASURES - GENERAL	4.57 - Trunk fat percentage (0.0)	PHYSICAL MEASURES - GENERAL
6.12 - Standing height (0.0)	PHYSICAL MEASURES - GENERAL	4.4 - Testosterone (0.0)	BLOOD ASSAYS
5.98 - Whole body water mass (2.0)	PHYSICAL MEASURES - GENERAL	4.31 - Systolic blood pressure	PHYSICAL MEASURES - CARDIAC & BLOOD VESSELS

Phenotype-wide association studies analysis table of the second pattern:

Supplementary Table 2. Phenotype-wide association studies analysis table of significant hits (from largest to smallest hit association magnitude) above the Bonferroni correction threshold of the second pattern. The Magnitude - Phenotype column has the magnitude of the association between the phenotype and the amygdala subregion-(sub)cortical region covariance which are shown in units as $-\log_{10}$ of the p-values (left) and the name of the phenotype (right). The Category column shows under which of the 11 broad domains the phenotype lies.

Pattern 2			
Magnitude - Phenotype	Category	Magnitude - Phenotype	Category
12.24 - Mother still alive (0.0)	LIFESTYLE AND ENVIRONMENT - GENERAL	6.23 - Impedance of whole body (0.0)	PHYSICAL MEASURES - GENERAL
12.12 - Current employment status (0.0)	LIFESTYLE AND ENVIRONMENT - GENERAL	5.9 - Mean carotid IMT (intima-medial thickness) at 210 degrees (2.0)	PHYSICAL MEASURES - CARDIAC & BLOOD VESSELS
11.08 - Own or rent accommodation lived in (0.0)	LIFESTYLE AND ENVIRONMENT - GENERAL	5.87 - Glycated haemoglobin (HbA1c) (0.0)	BLOOD ASSAYS
9.41 - Own or rent accommodation lived in (0.0)	LIFESTYLE AND ENVIRONMENT - GENERAL	5.58 - Interval between previous point and current one in alphanumeric path (trail #2) (2.0)	COGNITIVE PHENOTYPES
9.3 - Mean carotid IMT (intima-medial thickness) at 240 degrees (2.0)	PHYSICAL MEASURES - CARDIAC & BLOOD VESSELS	5.33 - Duration to complete alphanumeric path (trail #2) (2.0)	COGNITIVE PHENOTYPES
8.73 - Current employment status (0.0)	LIFESTYLE AND ENVIRONMENT - GENERAL	5.32 - Maximum carotid IMT (intima-medial thickness) at 210 degrees (2.0)	PHYSICAL MEASURES - CARDIAC & BLOOD VESSELS
8.65 - Maximum carotid IMT (intima-medial thickness) at 240 degrees (2.0)	PHYSICAL MEASURES - CARDIAC & BLOOD VESSELS	5.31 - Maximum carotid IMT (intima-medial thickness) at 120 degrees (2.0)	PHYSICAL MEASURES - CARDIAC & BLOOD VESSELS
7.79 - Mean carotid IMT (intima-medial thickness) at 150 degrees (2.0)	PHYSICAL MEASURES - CARDIAC & BLOOD VESSELS	5.21 - Systolic blood pressure	PHYSICAL MEASURES - CARDIAC & BLOOD VESSELS
7.62 - Minimum carotid IMT (intima-medial thickness) at 150 degrees (2.0)	PHYSICAL MEASURES - CARDIAC & BLOOD VESSELS	5.14 - Mean carotid IMT (intima-medial thickness) at 120 degrees (2.0)	PHYSICAL MEASURES - CARDIAC & BLOOD VESSELS
7.44 - Time to complete round (0.1)	COGNITIVE PHENOTYPES	5.07 - Maximum carotid IMT (intima-medial thickness) at 150 degrees (2.0)	PHYSICAL MEASURES - CARDIAC & BLOOD VESSELS
7.41 - How are people in household related to participant (0.0)	LIFESTYLE AND ENVIRONMENT - GENERAL	5.02 - Impedance of arm (right) (0.0)	PHYSICAL MEASURES - GENERAL
7.3 - Number in household (0.0)	LIFESTYLE AND ENVIRONMENT - GENERAL	4.9 - Impedance of arm (left) (0.0)	PHYSICAL MEASURES - GENERAL
7.06 - Minimum carotid IMT (intima-medial thickness) at 240 degrees (2.0)	PHYSICAL MEASURES - CARDIAC & BLOOD VESSELS	4.83 - Hair/balding pattern (0.0)	LIFESTYLE AND ENVIRONMENT - GENERAL
6.45 - Father still alive (0.0)	LIFESTYLE AND ENVIRONMENT - GENERAL	4.7 - FI7 : synonym (0.0)	COGNITIVE PHENOTYPES
6.37 - Number of symbol digit matches made correctly (2.0)	COGNITIVE PHENOTYPES	4.32 - IGF-1 (0.0)	BLOOD ASSAYS
6.36 - Number of symbol digit matches attempted (2.0)	COGNITIVE PHENOTYPES		

Phenotype-wide association studies analysis table of the third pattern:

Supplementary Table 3. Phenotype-wide association studies analysis table of significant hits (from largest to smallest hit association magnitude) above the Bonferroni correction threshold of the third pattern. The Magnitude - Phenotype column has the magnitude of the association between the phenotype and the amygdala subregion-(sub)cortical region covariance which

are shown in units as $-\log_{10}$ of the p-values (left) and the name of the phenotype (right). The Category column shows under which of the 11 broad domains the phenotype lies.

Pattern 3			
Magnitude - Phenotype	Category	Magnitude - Phenotype	Category
69.48 - Height (2.0)	PHYSICAL MEASURES - GENERAL	9.85 - Impedance of leg (left) (0.0)	PHYSICAL MEASURES - GENERAL
65.54 - Standing height (0.0)	PHYSICAL MEASURES - GENERAL	9.66 - Total bilirubin (0.0)	BLOOD ASSAYS
60.27 - Whole body water mass (2.0)	PHYSICAL MEASURES - GENERAL	9.59 - Drive faster than motorway speed limit (0.0)	LIFESTYLE AND ENVIRONMENT - EXERCISE AND WORK
57.58 - Leg fat-free mass (right) (0.0)	PHYSICAL MEASURES - GENERAL	9.24 - HDL cholesterol (0.0)	BLOOD ASSAYS
57.06 - Hand grip strength (right) (0.0)	PHYSICAL MEASURES - GENERAL	8.86 - Platelet crit (0.0)	BLOOD ASSAYS
56.19 - Whole body fat-free mass (0.0)	PHYSICAL MEASURES - GENERAL	8.59 - Average total household income before tax (0.0)	LIFESTYLE AND ENVIRONMENT - GENERAL
55.94 - Leg fat percentage (right) (0.0)	PHYSICAL MEASURES - GENERAL	8.47 - QRS duration (2.0)	PHYSICAL MEASURES - CARDIAC & BLOOD VESSELS
55.07 - Sitting height (0.0)	PHYSICAL MEASURES - GENERAL	8.37 - Number of puzzles correctly solved (2.0)	COGNITIVE PHENOTYPES
53.6 - Hand grip strength (left) (0.0)	PHYSICAL MEASURES - GENERAL	8.36 - Own or rent accommodation lived in (0.0)	LIFESTYLE AND ENVIRONMENT - GENERAL
52.49 - Arm fat-free mass (left) (0.0)	PHYSICAL MEASURES - GENERAL	8.28 - Alanine aminotransferase (0.0)	BLOOD ASSAYS
52.05 - Arm fat-free mass (right) (0.0)	PHYSICAL MEASURES - GENERAL	8.2 - Hair/balding pattern (0.0)	LIFESTYLE AND ENVIRONMENT - GENERAL
50.5 - Arm predicted mass (left) (0.0)	PHYSICAL MEASURES - GENERAL	7.31 - Gamma glutamyltransferase (0.0)	BLOOD ASSAYS
42.49 - Forced vital capacity (FVC) (0.0)	PHYSICAL MEASURES - GENERAL	7.2 - Number in household (0.0)	LIFESTYLE AND ENVIRONMENT - GENERAL
40.53 - Body fat percentage (0.0)	PHYSICAL MEASURES - GENERAL	7.06 - IGF-1 (0.0)	BLOOD ASSAYS
40.44 - Arm fat percentage (right) (0.0)	PHYSICAL MEASURES - GENERAL	7.03 - Hair/balding pattern (0.0)	LIFESTYLE AND ENVIRONMENT - GENERAL
39.42 - Arm fat percentage (left) (0.0)	PHYSICAL MEASURES - GENERAL	6.72 - Illnesses of mother (0.0)	LIFESTYLE AND ENVIRONMENT - GENERAL
37.03 - Forced expiratory volume in 1-second (FEV1) (0.0)	PHYSICAL MEASURES - GENERAL	6.26 - Own or rent accommodation lived in (0.0)	LIFESTYLE AND ENVIRONMENT - GENERAL
33.42 - Testosterone (0.0)	BLOOD ASSAYS	6.14 - Time to complete round (0.1)	COGNITIVE PHENOTYPES
32.62 - Weight (pre-imaging) (2.0)	PHYSICAL MEASURES - GENERAL	6.09 - Whole body fat mass (0.0)	PHYSICAL MEASURES - GENERAL
31.9 - Leg fat mass (right) (0.0)	PHYSICAL MEASURES - GENERAL	6.07 - Hair/balding pattern (0.0)	LIFESTYLE AND ENVIRONMENT - GENERAL

30.49 - Impedance of arm (left) (0.0)	PHYSICAL MEASURES - GENERAL	6.06 - Weekly usage of mobile phone in last 3 months (0.0)	LIFESTYLE AND ENVIRONMENT - EXERCISE AND WORK
30.17 - Impedance of arm (right) (0.0)	PHYSICAL MEASURES - GENERAL	5.83 - Sensitivity / hurt feelings (0.0)	MENTAL HEALTH SELF-REPORT
25.8 - Weight (0.0)	PHYSICAL MEASURES - GENERAL	5.8 - Father still alive (0.0)	LIFESTYLE AND ENVIRONMENT - GENERAL
25.39 - Peak expiratory flow (PEF) (0.0)	PHYSICAL MEASURES - GENERAL	5.76 - Apolipoprotein A (0.0)	BLOOD ASSAYS
25.12 - Impedance of whole body (0.0)	PHYSICAL MEASURES - GENERAL	5.62 - Direct bilirubin (0.0)	BLOOD ASSAYS
24.85 - Haemoglobin concentration (0.0)	BLOOD ASSAYS	5.59 - Aspartate aminotransferase (0.0)	BLOOD ASSAYS
23.58 - Haematocrit percentage (0.0)	BLOOD ASSAYS	5.44 - Fluid intelligence score (0.0)	COGNITIVE PHENOTYPES
22.1 - Red blood cell (erythrocyte) count (0.0)	BLOOD ASSAYS	4.95 - Leisure/social activities (0.0)	LIFESTYLE AND ENVIRONMENT - GENERAL
19.02 - Waist circumference (0.0)	PHYSICAL MEASURES - GENERAL	4.77 - Lymphocyte count (0.0)	BLOOD ASSAYS
18.81 - Hair/balding pattern (0.0)	LIFESTYLE AND ENVIRONMENT - GENERAL	4.68 - Time spent driving (0.0)	LIFESTYLE AND ENVIRONMENT - EXERCISE AND WORK
16.52 - Trunk fat percentage (0.0)	PHYSICAL MEASURES - GENERAL	4.49 - Frequency of consuming six or more units of alcohol (0.0)	LIFESTYLE AND ENVIRONMENT - ALCOHOL
15.79 - Average weekly beer plus cider intake (0.0)	LIFESTYLE AND ENVIRONMENT - ALCOHOL	4.47 - Length of mobile phone use (0.0)	LIFESTYLE AND ENVIRONMENT - EXERCISE AND WORK
14.43 - Creatinine (0.0)	BLOOD ASSAYS	4.46 - Seen doctor (GP) for nerves	MENTAL HEALTH SELF-REPORT
13.6 - Urate (0.0)	BLOOD ASSAYS	4.4 - Qualifications (0.0)	LIFESTYLE AND ENVIRONMENT - GENERAL
12.11 - Impedance of leg (right) (0.0)	PHYSICAL MEASURES - GENERAL	4.4 - Time number displayed for (2.0)	COGNITIVE PHENOTYPES
11.71 - SHBG (0.0)	BLOOD ASSAYS	4.39 - Platelet count (0.0)	BLOOD ASSAYS
11.54 - Forced expiratory volume in 1-second (FEV1)	COGNITIVE PHENOTYPES	4.39 - Time to complete round (0.0)	COGNITIVE PHENOTYPES
10.9 - Current employment status (0.0)	LIFESTYLE AND ENVIRONMENT - GENERAL	4.3 - Number of incorrect matches in round (0.1)	COGNITIVE PHENOTYPES
10.59 - Sleeplessness / insomnia (0.0)	LIFESTYLE AND ENVIRONMENT - EXERCISE AND WORK		
10.03 - Current employment status (0.0)	LIFESTYLE AND ENVIRONMENT - GENERAL		

- The authors use two strategies of multiple comparison correction (Bonferroni and FDR) but do not explain or describe differences between the results nor specify which methods' outcome is then reported.

Those are 2 of the most frequently used multiple comparisons corrections used in biomedicine and beyond. In short, the effect sizes for each phenotype are the same; the two correction methods simply offer two different yardsticks to gauge the same modeling result.

They are based on rejecting the null hypothesis if the likelihood of the observed data under the null hypothesis is low. Bonferroni's correction happens to be more stringent than FDR (false discovery rate). The results are in the plots for appreciation by the reader. There are no inherent "differences" between the Bonferroni and FDR-thresholded version of the results, just that each phenotype has several degrees of rigor to it that are readily accessible; our reporting focused on key results with strongest statistical backing (cf. above).

The actual effect sizes of every phenotype are the same (i.e., vector entry in loading weights from fitted model) for FDR and Bonferroni's correction. The thresholding or the 'level of significance' which defines whether a phenotype is considered to be significant or not is where the thresholds diverge. In the current study, as well as in previous publications from our lab that align with this direction (e.g., those published in the Nature Publishing Group, PLOS Biology, etc.), we aim to allow readers to pick the level of rigor that they deem best - so we leave it up to the reader.

Additionally, motivated by this reviewer comment, we have gone through the results section to specify explicitly that we place a focus on the most robust, Bonferroni corrected findings. We now also clearly indicate if the phenome-wide results mentioned are FDR significant only.

- Is there a cross-validation procedure that could show that these phenome wide associations are robust results that are replicable across independent samples (e.g. a-cross validation analyses, comparing to other larger samples (such as HPC?)?

We thank the reviewer for the helpful suggestion. There is no cross-validation procedure to replicate the results onto independent samples. This is because the UK Biobank has a unique collection of phenotypes that are not available in any other population cohort. The uniqueness of the UK Biobank resource is the very reason why we set out to carry out our study based on this resource in the first place. Please also see answer to the next question.

Given the larger n (yet not in the range of 10k, see BWAS discussion –not addressed by the authors) it seems likely that even after multiple comparison corrections false positives may be present. This seems especially relevant given my methodological concerns regarding the noise in the brain measure.

We thank the reviewer for this helpful point with regards to ensuring statistical significance in big sample settings. If the reviewer was right and we are susceptible to some source of noise, we would have expected to find significant brain-behavior links in all initially derived model

components - this is however not what our results show. Out of the 6 robust population modes, 3 - that is 50% of the population covariation regimes - did not feature any significant or robust brain-behavior associations, which reinforces our point that false positives due to brain measure noise is unlikely.

We agree that statistical significance needs to be considered with a grain of salt in the big-sample setting. We took this into consideration by using both the FDR (false discovery rate) and Bonferroni's correction for multiple comparisons corrections, which enabled us to explore relationships between the subject-wise expression of a given covariation pattern and the 977 phenotypes. We chose to use Bonferroni's correction method for a more conservative correction for multiple comparisons. With the stringent threshold levels, we were even able to use our phenotype-wide association analysis to disqualify some of the association patterns from being significant enough to be considered robust.

Importantly, 3 out of the 6 original patterns did not have significant phenotypes above the more stringent Bonferroni correction threshold, hence showing us that our thresholds ensured low variance within the pattern by examining the behavior of the association between the given phenotypes and the inter-individual covariance of the plasticity coupling between the amygdala subregions and the (sub)cortical regions in each of the significant patterns. Hence, we are able to rely on our phenotype-wide association analysis to be accurate. As for the patterns that were able to survive the stringency of the Bonferroni threshold by exhibiting significant phenotypes above both thresholds, the Manhattan plots show the degree of relevance of each phenotype compared to its flanking alternative phenotypes.

6. Conceptually I do not follow the logic on page 11 about the robustness of the 6 modes. What is the standard to consider a result a “high explained variance”? Why are low/high numbers of phenotypic hits indicative of robustness? Relatedly, the authors say that they use the “phenome-wide analysis as a device to find real-world relevance of the uncovered principled signature” and argue “convincing real-world relevance” (notably, the variables with the highest significance threshold are “height” and “mother still alive” in the two modes with above threshold correction). Is the argument that if there is no significant (beyond multiple comparison correction) phenotype association, it is not considered relevant? How is noise in each of the measures (MRI, phenotypes) taken into account into a null finding?

The degree of explanatory strength of various of the modes were obtained based on three different methods:

1-The first way entails computing the amygdala-brain approximation: the Pearson's rho value for the subject-wise expression of covariation patterns of the amygdala subregions and the subject-wise expression of covariation patterns of the (sub)cortical regions for all the modes that our intrinsic plasticity coupling analysis has produced. Higher Pearson rho values entails a higher degree of explained variance at the population level. This is probably the most common way to

quantify the overall size of effect in the class of doubly-multivariate decomposition methods (Wang et al., 2020).

2-The second way of using the Pearson rho value to obtain a degree of explanatory relevance of an amygdala-brain pattern in a broader context, not just the brain itself, involves using our phenome-wide association analysis. By examining the association between the given phenotypes and the inter-individual covariance of the coupling in structural change between the amygdala subregions and the (sub)cortical regions in each of the 6 robust modes, we are able to confirm explanatory value of the plasticity covariance relationships between amygdala subregions and the (sub)cortical regions. Significant association between the phenotypes and the inter-individual covariance of the plasticity coupling between the amygdala subregions and the (sub)cortical regions indicates a high explained variance and a prevalence of the coupling of the covarying plasticity in the amygdala subregions and the (sub)cortical regions within behavioral traits with their relevance in everyday life that contribute to the significance of the phenotypes.

Outside the context of robustness testing using external non-brain data, we chose to hone in on examples of phenotypes in the text in order to illustrate wider phenomena that shows the manifestation of the prevailing covariance of plasticity in the amygdala subregions and in the (sub)cortical regions within behavioral traits related to everyday life to explain the significant phenotypes going beyond the multiple comparison correction (cf. above).

3- The third way to assess the strength of each mode entails calculating the explanatory value of every individual mode based on an empirical permutation testing frameworks. 100 permutation iterations were carried out and in each such step, a randomized permuted version of the (sub)cortical regions' plasticity covariation data is generated and fitted with the original amygdala subregional plasticity covariation. The correlation coefficients of the ensuing embedding spaces were then calculated which yielded an empirical null distribution. In each iteration, the significance of the correlation coefficients obtained from the modes builds this null distribution. The results are then compared to the original observed values from the original version of the model fitted with unpermuted data to compute p-values that validate our results against noise in the T1-derived brain features.

Given our multi-pronged mode assessment approach, we felt confident of the pertinence of the results yielded by our analyses.

References:

Wang HT, Smallwood J, Mourao-Miranda J, Xia CH, Satterthwaite TD, Bassett DS, Bzdok D. Finding the needle in a high-dimensional haystack: Canonical correlation analysis for neuroscientists, *NeuroImage*, 216, 2020.

7. The interpretation and conclusions drawn in the discussion about the functional relevance of anatomical co-variations and respective phenotypes appear post-hoc, selective and not well derived from a breadth of previous literature – instead a few single studies seem picked to fit a narrative. This concern relates back to my comment on the lack of hypotheses describing expected relationships within the context of prior literature.

We thank the reviewer for this helpful point. We have now better motivated the study in the introduction by spelling out clear a-priori hypotheses (cf. above)

Further Comments that I hope will be helpful to the authors to improve the manuscript:

a. Figures 1-5 show amygdala subregions but captions are not clearly stating the source of the anatomical images. It appears that these are taken from the Saygin et al paper, yet the authors do not state the original source and related copyright aspects. The B section of these figures also lack relevant details (left/right) and appear like a screenshot from a GUI (e.g. the colorbar selection that also misses the description of the measure).

We thank the reviewer for this important pointer. We have corrected the referenced omission in our revised version of the manuscript as follows:

Figure 1. Within-subject structural plasticity effects in the central nucleus and anterior amygdaloid area covary especially with the inferior parietal lobule. Principle patterns of structural covariation due to longitudinal plasticity between gray matter volume change over time in 18 microanatomically distinct amygdala subregions (9 per hemisphere) (Saygin ZM & Kliemann D et al., 2017).

Figure 2. Within-subject structural plasticity ties the medial, cortical, lateral and central amygdala nuclei to brain regions related to alertness and visual conscious awareness. A) shows the parameter weights tracking the volume effects of the 18 specific amygdala subregions (Saygin ZM & Kliemann D et al., 2017) with their co-occurring structural changes in (sub)cortical partner regions across the brain.

Figure 3. Within-subject structural plasticity links the basal, accessory basal, lateral and paralaminar nuclei especially with the prefrontal cortex. A) The basal, accessory basal, lateral and paralaminar subregions undergo the strongest structural covariation in population mode 3, among all amygdala subregions (Saygin ZM & Kliemann D et al., 2017), in the context of the distributed cortical volume changes.

Figure 4. Lateralization plasticity effects driven by the cortical nucleus, anterior amygdaloid area, central nucleus and lateral nucleus co-vary with awareness/alertness-related brain regions. We performed a hemispheric difference analysis in the context of the left-right divergence of the structural plasticity changes in 9 amygdala subregions with the structural plasticity of 109 brain regions by means of co-decomposition based on partial least squares canonical (PLSC). We determined how the ensuing subregion patterns lateralized in the 9 amygdala subregions. Shown here is the subregion lateralization in mode 1 of the amygdala - brain covariation. A) conveys the direction of lateralization of each of the 9 amygdala subregions in mode 1 (Saygin ZM & Kliemann D et al., 2017).

Figure 5. Lateralization plasticity effects driven by the anterior amygdaloid area, lateral nucleus, and cortical nucleus with a lateralization effect in awareness/alertness-related brain regions together with the inferior parietal lobule. Shown here is the subregion lateralization in signature 2 of the amygdala - brain covariation. A) conveys the direction of lateralization plasticity effects of each of the 9 amygdala subregions in signature 2 (Saygin ZM & Kliemann D et al., 2017).

Supplementary Figure 1: Lateralization plasticity effects driven by the cortical, central, and medial nuclei. Shown here is the subregion lateralization in signature 3 of the amygdala - brain covariation. A) conveys the direction of lateralization of each of the 9 amygdala subregions in signature 3 (Saygin ZM & Kliemann D et al., 2017).

References:

Saygin ZM & Kliemann D (joint 1st authors), Iglesias JE, van der Kouwe AJW, Boyd E, Reuter M, Stevens A, Van Leemput K, Mc Kee A, Frosch MP, Fischl B, Augustinack JC., 2017. High-resolution magnetic resonance imaging reveals nuclei of the human amygdala: manual segmentation to automatic atlas. *Neuroimage*, 155, 370-382.

We have also amended the design choices of our figures in our revised version of the manuscript as follows:

Figure 1:

Figure 2:

Figure 3:

Figure 4:

Figure 5:

A**B**
Supplementary Figure 1:

b. It seems that the study is conceptually similar to at least one other study from the same group using similar methodology (e.g. relating a brain measure with phenome-wide association analyses). To limit the degrees of freedom in analytic choices, it would have been ideal for scientific rigor and replicability to preregister the analyses plan prior to conducting the analyses. The authors may want to comment on their choice to not preregister this and may want to outline which analytic decisions were made prior to observing results thereof.

This is the first paper from our group that relates *changes* in brain region volumes to *changes* in brain volumes from other regions in two different brain compartments, thus leveraging a unique strength of the UK Biobank Imaging dataset and is therefore distinct in motivation and kind from our previous work.

Further, it is uncommon to do preregistration in our data-driven large dataset analyses in our lab and in our befriended labs in computational neuroscience.

c. It seems that the data and code for this study would not be made available to the scientific community. I understand that the input data is available. Open Sciences practices that foster reproducibility and replicability would suggest to also provide the scientific community with the extracted measures of interest (i.e. the volume measures per ROI), confound regressors and or at least the code to rerun the analyses (if in line with the data policies of the UK biobank) to reproduce said outputs.

All data has been already provided by the UK Biobank team and we did not compute the brain-derived measures ourselves (cf. above). We will publish our code onto a github repository that generated the full set of results for all intrinsic plasticity coupling patterns. Any reader interested in more carefully investigating any details of our obtained effects is welcomed to do so. We will include a link to the UKBiobank data showcase (<https://biobank.ndph.ox.ac.uk/ukb/>) for further research.

d. The structuring of the information provided in the manuscript needs to be improved and cleaned from errors.

- Figure captions are redundant with information from the text and unusually detailed.

We appreciate the reviewer's suggestion. We aim at autonomous figures in our lab which explains the unusually detailed figure captions.

- The middle paragraph on page 26 contains the same section twice (“Our findings also...”)

We appreciate the reviewer's pointer. This will be fixed in the revised version of the manuscript:

Our findings also favor a plasticity partnership between the prefrontal cortex and especially the laterobasal amygdala subregion – in line with the idea of a labor division in somatic markers processing, influenced by external stimulus input. The amygdala's laterobasal subregion is commonly believed to serve as receiving hub of stimuli information from the external environment through different sensory cortices (Bzdok et al., 2013; Janak and Tye, 2015). This may provide a driver for synaptic plasticity in this particular amygdala segment (LeDoux, 2007) (Samson et al., 2005) **and potentially the downstream processing partners of this amygdala nucleus.** Adding weight to this possibility, the central and medial amygdala nuclei have been shown receptive to information issued from the lateral subregion (LeDoux, 2007). Our results would be compatible with this interpretation, given that the vmPFC here emerged as one of the strongest covarying partners in the entire prefrontal cortex with the amygdala subregions. Taken together, plasticity events coupled between the amygdala volume and prefrontal volume may

occur due to the role of the larger laterobasal subdivision; with potential relevance for adaptations in somatic markers processing, and thus perhaps internal conscious awareness, over time. Our findings also favor a plasticity partnership between the prefrontal cortex and especially the laterobasal amygdala subregion in particular — in line with the idea of a labor division in somatic markers processing, influenced by external stimulus input. The amygdala's laterobasal subregion is commonly believed to serve as receiving hub of stimuli information from the external environment through different sensory cortices (Bzdok et al., 2013; Janak and Tye, 2015). This may provide a driver for synaptic plasticity in this particular amygdala segment (LeDoux, 2007) (Samson et al., 2005) and potentially this amygdala nucleus' downstream processing partners. Adding weight to this possibility, the central and medial amygdala nuclei have been shown receptive to information issued from the lateral subregion (LeDoux, 2007). Our results would be compatible with this interpretation, given that the vmPFC here emerged as one of the strongest covarying partners in the entire prefrontal cortex with the amygdala subregions. Taken together, plasticity events coupled between the amygdala volume and prefrontal volume may occur due to the role of the larger laterobasal subdivision; with potential relevance for adaptations in somatic markers processing, and thus perhaps internal conscious awareness, over time.

- The rationale for some analyses is stated in the results and not provided in detail for other analyses.

We appreciate the reviewer's pointer. We have added a rationale for our Phenome-wide analysis and a rationale for the Ranking of amygdala subregion changes analysis in the revised version of the manuscript as follows:

Results:

Phenome-wide analysis

We systematically explored associations between individual expressions of a covariation pattern and 977 phenotypes, applying multiple comparison corrections. Utilizing Pearson's correlation, we analyzed associations and statistical significance between phenotypes and amygdala subregion-cortical region plasticity coupling covariation patterns. Two corrections were applied to accommodate association tests for each covariation pattern: Bonferroni's correction, adjusted for the number of tested phenotypes ($0.05/977 = 5.11e-5$), and the false discovery rate (FDR) (Benjamini and Hochberg, 1995), set at 5% (Miller et al., 2016; Raizada et al., 2008; Sha et al., 2021), according to standard protocols (Genovese et al., 2002). For visualization, phenotypes in Manhattan plots were color-coded and categorized per FUNPACK-defined membership. Building upon our primary analysis, the thorough annotation of the derived AM-brain covariation patterns in the context of our phenome-wide analysis revealed distinct relationships between the AM-brain covariation patterns and 977 lifestyle factors, demographic indicators, and mental health assessments.

References:

Benjamini, Y., Hochberg, Y., 1995. Controlling the False Discovery Rate: A Practical and Powerful Approach to Multiple Testing. *Journal of the Royal Statistical Society: Series B (Methodological)* 57, 289-300.

Miller, K.L., Alfaro-Almagro, F., Bangerter, N.K., Thomas, D.L., Yacoub, E., Xu, J., Bartsch, A.J., Jbabdi, S., Sotiropoulos, S.N., Andersson, J.L.R., Griffanti, L., Douaud, G., Okell, T.W., Weale, P., Dragonu, I., Garratt, S., Hudson, S., Collins, R., Jenkinson, M., Matthews, P.M., Smith, S.M., 2016. Multimodal population brain imaging in the UK Biobank prospective epidemiological study. *Nature Neuroscience* 19, 1523-1536.

Raizada, R.D., Richards, T.L., Meltzoff, A., Kuhl, P.K., 2008. Socioeconomic status predicts hemispheric specialisation of the left inferior frontal gyrus in young children. *Neuroimage* 40, 1392-1401.

Sha, Z., Schijven, D., Carrion-Castillo, A., Joliot, M., Mazoyer, B., Fisher, S.E., Crivello, F., Francks, C., 2021. The genetic architecture of structural left-right asymmetry of the human brain. *Nat Hum Behav* 5, 1226-1239.

Genovese, C.R., Lazar, N.A., Nichols, T., 2002. Thresholding of statistical maps in functional neuroimaging using the false discovery rate. *Neuroimage* 15, 870-878.

Results:

Ranking of amygdala subregion changes:

We finally conducted a 'ranking' analysis to provide a meticulous quantification of how AM subregions undergo substantial modifications in grey matter volume across distinct age categories, thereby resonating with the earlier affirmations regarding the brain's dynamic reconfiguration. The centrality of our amygdala subregion changes ranking analysis lies in the differential trajectory of structural plasticity in the amygdala subregions across various life stages, underpinning the phenomenological variations intrinsic to individual cognitive and emotional experiences. Our motivation stems from comprehensively characterizing the most significant shifts in amygdala (AM) subregions in a longitudinal framework, providing granular insights into the temporal dynamics of structural plasticity amidst diverse age groups. Given the substantiated ties between structural plasticity and various cognitive domains, our careful quantification of the "biggest movers" in AM subregions will potentially unveil the covert neural substrates that underpin the variegated cognitive and emotional landscapes experienced by individuals as they traverse through different life stages. Moreover, it provokes additional

queries regarding how diverse environmental, genetic, and lifestyle factors might further modulate these structural transitions.

- It seems that the final conclusions of the paper are pointing to “past studies” (e.g., last sentence “Past studies further verify the roles of the central and medial nuclei in conscious awareness in all its forms, internal and external, as they have shown their roles in controlling fight or flight through noradrenaline regulation.”)

We thank the reviewer for the helpful pointer. We have edited the final conclusions as follows:

Discussion:

In line with our findings on the amygdala, the idea of conscious awareness and interoception is coherently extended from being only an internal process to regulating conscious awareness both internally and externally. External conscious awareness may seem more abstract than its internal counterpart because it encompasses one's subjective sense of relevance and presence within their environment. However, insights into the roles of the central and medial amygdala nuclei in awareness and interoception have shed light on this concept. The roles of the central and medial nuclei have been verified in conscious awareness in various forms, internal and external, as they have shown their roles in controlling fight or flight through noradrenaline regulation.

Otherwise, the reason that we point to past studies is because it is essential to explicitly state when information has been found in external sources as we link the information from past findings to our findings.

e. Figure 7: The content of the figures is very hard to decipher – have the authors considered to re-order the data according to nuclei? Their main claim seems to be that specific nuclei change most over time and not necessarily which age group shows the most changes.

The subnuclei were systematically reordered to reveal more directly what the data has to offer. This is now more clearly stated in the caption of Figure 7, thanks to the reviewer's suggestion as follows:

Figure 7. Centromedial and laterobasal nuclei groups change the most over time in UKBiobank participants. The analysis conveys the specific nuclei change in gray matter volume over time between the second and the first timepoints for the 18 amygdala subregions, arranged in ascending order, from the highest atrophy (cold = atrophy; most negative change) to largest growth (hot = growth; most positive change) in 6 different age groups (rows). It shows that the gray matter volume in the amygdala subregions is subject to different degrees of change in distinct directions at various stages of an individual's life. The medial nucleus (Me) was found at both extremities of atrophy and growth of gray matter volume over time in different age

groups and the central nucleus (Ce) and Me were found to undergo the largest growth in gray matter volume over time across all amygdala subregions, while the accessory basal nucleus (AB) atrophied the most across the subregions.

f. Comments on the supplementary data plan:

- It states that diffusion imaging was used. Respective results were not reported in the manuscript.

This has been fixed thanks to the reviewer's suggestion.

- Normalization template is labeled as "MNI152" but does not specify which one (e.g., linear, non-linear...).

This has been added thanks to the reviewer's suggestion.

- Merely relating to previous publications (e.g. Miller & Alfaro-Almagro) for relevant details on major methodological aspects is not sufficient.

We thank the reviewer for this point, and we agree. We have referenced external publications to explain that some choices in data collection and preprocessing have been made by external sources.

Minor Comments (not an exclusive list):

PHEASANT vs PHEASANT (page 9)

Page 18: "(Figure?)"

What is a "plasticity twin"? (Page 7)

Providing a table for the subregion abbreviations seems unnecessary.

We thank the reviewer for these pointers, we will fix the spotted mistakes in the revised version of the manuscript.

Reviewer #2 (Remarks to the Author):

Ghanem et al have presented an interesting paper examining structural MRI data from the UK Biobank to identify cortical/subcortical regions that have longitudinal changes (2 timepoints, adult data) that are associated with amygdala changes in grey matter. Using a partial least squares approach they identify three circuits between the amygdala and subcortical/cortical regions that have associated longitudinal changes. They examine the laterality specificity and also the association with a number of phenotypes including socioeconomic factors, and lifestyle factors. Overall the premise of the paper is interesting and creative, with what appears to be a data-driven approach to finding the relationship between two datasets (amygdala grey matter volume and cortical/subcortical grey matter volume).

We are thankful for the favorable assessment of our work.

1. The introduction makes large generalizations and, at times assumptions, that are not referenced or well-supported. As the greatest example, lines 64-71 discuss changes in gray matter volume as reflecting changes in structural plasticity. This is a large assumption, as measurements from MRI are coarse, and cannot comment on cellular-level processes, and changes in structural MRI can be due to a number of influences, making it hard to pinpoint the exact influence that led to gray matter changes, as opposed to fMRI or EEG, in which a stimulusresponse measurement can more directly be measured. Changes in gray matter can be due to a number of influences that may include, but may also not be solely due to, plasticity. Furthermore, given that the ages of the participants are middle-aged or older, there are likely to be additional influences that likely cause the observed changes. These are only a few examples. I would suggest that the authors provide a clear definition of plasticity, and thoroughly review the introduction for more thorough referencing, and specific descriptions throughout.

We thank the reviewer for the helpful suggestion. In alignment with widely acknowledged scientific practices in previous brain-imaging publications, our adoption of the notion of plasticity is consistent with methodologies and theoretical underpinnings espoused in previously published, well-cited papers that have used the same notion of plasticity from sMRI.

Eminent studies have exploited sMRI to extrapolate and interpret changes in the brain's physical and functional structure, thereby providing crucial insights into the nature and scope of neuroplasticity across diverse demographic and clinical populations (Zatorre et al., 2012;

Maguire et al., 2000; Lovden et al., 2013). Utilizing sMRI, these research endeavors have correlated volumetric and morphological changes within the brain to behavioral, cognitive, and environmental shifts, elucidating the brain's remarkable ability to remodel and reorganize itself in response to varied stimuli (Valk et al., 2017; May, 2011). The significance of utilizing diffusion MRI to interpret structural changes and elucidate neural processes has also been delineated in prior research (Assaf and Johansen-Berg, 2013). Consequently, our work is deeply rooted in these established practices, applying a similar lens of structural plasticity to delve into the complexities of brain-behavior interactions and their longitudinal implications (Taubert et al., 2010). This approach not only facilitates a nuanced understanding of neural adaptability and transformation but also ensures that our findings can be contextualized and critiqued within a broader, established scientific framework, encouraging constructive dialogue and comparative analyses with parallel studies in the realm of neuroplasticity.

In short, several previous, well-cited and widely respected brain-imaging studies have used the 'plasticity' term in the same sMRI-based setting as we do here.

References:

- Zatorre, R. J., Fields, R. D., & Johansen-Berg, H. (2012). Plasticity in gray and white: neuroimaging changes in brain structure during learning. *Nature Neuroscience*, 15(4), 528-536.
- Maguire, E. A., Gadian, D. G., Johnsrude, I. S., Good, C. D., Ashburner, J., Frackowiak, R. S., & Frith, C. D. (2000). Navigation-related structural change in the hippocampi of taxi drivers. *Proceedings of the National Academy of Sciences*, 97(8), 4398-4403.
- Lovden, M., Wenger, E., Martensson, J., Lindenberger, U., & Backman, L. (2013). Structural brain plasticity in adult learning and development. *Neuroscience & Biobehavioral Reviews*, 37(9), 2296-2310.
- Valk, S. L., Bernhardt, B. C., Trautwein, F. M., Bockler, A., Kanske, P., Guizard, N., ... & Singer, T. (2017). Structural plasticity of the social brain: Differential change after socio-affective and cognitive mental training. *Science Advances*, 3(10), e1700489.
- May, A. (2011). Experience-dependent structural plasticity in the adult human brain. *Trends in Cognitive Sciences*, 15(10), 475-482.
- Assaf, Y., & Johansen-Berg, H. (2013). The role of diffusion MRI in neuroscience. *NMR in Biomedicine*, 26(7), 849-865.
- Taubert, M., Draganski, B., Anwander, A., Müller, K., Horstmann, A., Villringer, A., & Ragert, P. (2010). Dynamic properties of human brain structure: learning-related changes in cortical areas and associated fiber connections. *Journal of Neuroscience*, 30(35), 11670-11677.

We have also defined a clear definition for plasticity in the introduction with references to past longitudinal studies by amending the introduction as follows:

Introduction: Depending on the degree of implication of a brain region in supporting a given behavioral function, it is susceptible to undergo remodeling alterations that are reflected in measures of gray matter volume. The change in gray matter volume in turn signifies not only an ability to learn, but also an enhancement of existing cognitive capabilities and probably also strengthening areas where neural processes are attenuated in other parts of the brain for the sake of compensation. Being able to examine the brain through the lens of structural plasticity provides a window into the effects of various exposures and prompted behaviors in everyday life. We are able to understand how different behavioral traits and lifestyle habits show links to the structural plasticity of a brain region through pattern learning which allows us to make steps towards more casually qualified statements about the brain (Valk et al., 2017) (Lovden et al., 2010; Mateos-Aparicio and Rodriguez-Moreno, 2019)

Structural plasticity refers to the brain's ability to undergo physical and functional reorganization in response to experiences, learning, and various environmental stimuli. This encompasses modifications in the neuronal connections, synaptic vesicle formation and uptake, neuronal remodeling, myelination, and even the observable changes in grey matter volume (Assaf, 2018). As shown by past imaging and behavioral studies, such adaptability is not limited by age or a particular cognitive domain (Zatorre et al., 2012). Whether in behavioral realms extending beyond mere motor capabilities or within more specialized cognitive capacities like language or empathy, the brain's structural constitution is known to reflect these experiences (Zatorre et al., 2012).

Central to understanding this notion of change is the comparison between cross-sectional and longitudinal studies. Past longitudinal analyses have shown their pivotality in comprehending lifelong neuroplasticity (Di Biase et al., 2023). Evidence from past brain-imaging/behavioral studies underscores the sensitivity and specificity of individual-level brain modifications: for instance, training targeted at enhancing distinct empathy systems leads to unique structural gray matter modifications (Valk et al., 2017). Similarly, language acquisition in teenagers has demonstrated distinct correlations with changes in grey matter density, emphasizing the dynamism of the brain's structural makeup in relation to evolving mental capacities (Stein et al., 2012). Furthermore, lateralization plays a role, with certain cognitive functions such as language likely favoring one hemisphere over the other, leading to discernible structural alterations in specific brain regions (Assaf, 2018; Valk et al., 2017; Taubert et al., 2010; Stein et al., 2012).

Past animal studies, conducted on rats, mice, and monkeys, have started to bridge the understanding between MRI-observable changes and the cellular architectures underlying them (Zatorre et al., 2012; Caroni et al., 2012). Notably, investigations spanning humans and rodents have identified commonalities in structural brain changes following task-based training, reinforcing the universality of structural plasticity mechanisms (Sagi et al., 2012).

In essence, structural plasticity epitomizes the brain's evolutionary advantage to adapt and evolve in response to a wide array of stimuli and experiences. This malleability, observable through advanced brain-imaging techniques and affirmed through various experimental designs, underpins the rationale for structural plasticity being the main driver for longitudinal studies that track and map the continuum of brain changes over different timescales.

References:

Valk, S. L., Bernhardt, B. C., Trautwein, F. M., Bockler, A., Kanske, P., Guizard, N., Collins, D. L., & Singer, T. (2017). Structural plasticity of the social brain: Differential change after socio-affective and cognitive mental training. *Sci Adv*, 3(10), e1700489.

<https://doi.org/10.1126/sciadv.1700489>

Assaf, Y. (2018). New dimensions for brain mapping. *Science*, 362(6418), 994-995.

<https://doi.org/10.1126/science.aav7357>

Zatorre, R. J., Fields, R. D., & Johansen-Berg, H. (2012). Plasticity in gray and white: neuroimaging changes in brain structure during learning. *Nat Neurosci*, 15(4), 528-536.

<https://doi.org/10.1038/nn.3045>

Di Biase, M. A., Tian, Y. E., Bethlehem, R. A. I., Seidlitz, J., Alexander-Bloch, A. F., Yeo, B. T. T., & Zalesky, A. (2023). Mapping human brain charts cross-sectionally and longitudinally. *Proc Natl Acad Sci U S A*, 120(20), e2216798120. <https://doi.org/10.1073/pnas.2216798120>

Stein, M., Federspiel, A., Koenig, T., Wirth, M., Strik, W., Wiest, R., Brandeis, D., & Dierks, T. (2012). Structural plasticity in the language system related to increased second language proficiency. *Cortex*, 48(4), 458-465. <https://doi.org/10.1016/j.cortex.2010.10.007>

Taubert, M., Draganski, B., Anwander, A., Müller, K., Horstmann, A., Villringer, A., & Ragert, P. (2010). Dynamic properties of human brain structure: learning-related changes in cortical areas and associated fiber connections. *J Neurosci*, 30(35), 11670-11677.

<https://doi.org/10.1523/JNEUROSCI.2567-10.2010>

Caroni, P., Donato, F., & Müller, D. (2012). Structural plasticity upon learning: regulation and functions. *Nat Rev Neurosci*, 13(7), 478-490. <https://doi.org/10.1038/nrn3258>

Sagi, Y., Tavor, I., Hofstetter, S., Tzur-Moryosef, S., Blumenfeld-Katzir, T., & Assaf, Y. (2012). Learning in the fast lane: new insights into neuroplasticity. *Neuron*, 73(6), 1195-1203.

<https://doi.org/10.1016/j.neuron.2012.01.025>

Another example is line 54, stating that the amygdala has a role in carrying out risk assessment in initiating self-preserving modes of execution. The amygdala does have a role in fear learning, as is discussed by the authors, but it is not clear (nor referenced)

what the role of the amygdala is in risk assessment—i.e, making judgements about the surroundings and decisions on how to behave.

We have also further clearly referenced and clearly defined the roles of the amygdala throughout the introduction as follows:

As a whole, the amygdala has a role in carrying out risk assessment and in initiating self-preserving modes of execution. Indeed, occupying the place as a central emotion, or even relevance, processor in the brain, the amygdala suggests itself to be an integrative hub tapping into various cortical networks (Adolphs, 2010). **A review of studies on animal amygdala lesions and axonal tracing concluded that distinct subregions within the amygdala play a crucial role in managing defensive states. These subregions amalgamate pertinent information, forming memories that aid in devising apt strategies to confront various threats, in addition to conducting threat detection risk assessments and modulating cardiovascular activity during perilous situations. The amygdala is posited to orchestrate the transition between fight and flight responses, contingent on the reaction it deems appropriate (Moscarello and Penzo, 2022).** Thus, as a central hub for emotion processing, the amygdala might well influence various cortical networks.

References:

Moscarello, J.M., Penzo, M.A., 2022. The central nucleus of the amygdala and the construction of defensive modes across the threat-imminence continuum. *Nat Neurosci* 25, 999-1008.

2. Similarly, in the introduction there are a number of confusing and oddly constructed sentences, ex: line 86, Richer datasets enable us to utilize the techniques and analytical methods that quantitatively revisit questions that we have been trying to answer since the one of the first recorded brain studies from the early periods of history (Mills and Tamnes, 2014). Or on lines 97, 98: Adding to the data wealth of the UKBiobank, we have exercised a granularity of 18 amygdala subregions (9 per hemisphere). Or lines 106-108, Instead, we here brought to bear phenome effects of various life factors on structural plasticity probed at the population level and across ~1,000 behaviour, lifestyle, and mental health indicators. These are a few examples, but numerous exist in the introduction as well as throughout the rest of the text, including the discussion. I would recommend that the authors review the full text for grammar.

We thank the reviewer for pointing out errors in our grammar. We have carefully reviewed the full manuscript text for grammar in introduction and discussion, and fixed the examples mentioned in the reviewed version of the manuscript as follows:

Introduction:

Line 86:

Richer datasets allow us to employ new analytical approaches to quantitatively revisit classical questions in neuroscience.

Lines 97,98:

To fully benefit from the data wealth of the UK Biobank, we have examined the amygdala at a granularity of 18 subregions (9 per hemisphere).

Lines 106-108:

Instead, in this study, we have examined the effects of various lifestyle factors on structural plasticity at the population level, querying approximately 1,000 indicators of behavior, everyday habits, and mental health.

3. Can the authors clarify the description of partial least squares canonical analysis? How is this different from canonical correlation analysis (CCA) or partial least squares correlation, for instance? Furthermore, can the authors provide a rationale for why this method was chosen over others?

Partial least squares canonical analysis (PLSC) aims to identify linear combinations (canonical variables) from two sets of variables such that the correlations between these combinations are maximized - it is directional (as PLS correlation) and imposes whitening decorrelation of variable sets (as CCA). However, it differs by emphasizing the variance explanation in the dependent variables, rather than just maximizing the correlation.

This method was chosen due to the following three core reasons:

1. We have quite a few variables per observation in our modeling scenario (cf. methods section); we can hence probably not afford more than a linear model, as opposed to a model class that would be capable of capturing higher-order non-linearities. It is difficult for most non-linear methods to allow for interpreting weights for understanding the profiles of neural and behavioral features carrying the brain-behavior relationship with such a high number of features and low number of observations. On the other hand, the PLSC has previously been carefully empirically evaluated to yield stable and accurate results with a large number of considered features obtained within the space of UKBiobank-level sample size (Helmer et al., 2020).

2. In the case of the brain-imaging data used, autocorrelation tends to be very prominent aspects within these datasets, which should be reflected in what analysis tools are

exactly chosen for a modeling task. Our method is not only able to handle, but actually thrives on and explicitly quantifies the autocorrelation pattern within an input data matrix (Wang et al., 2020) as part of solving an input-output modeling problem.

3. Our method is effective for reducing a large set of related variables to a typically smaller number of combinations or latent variables that capture most of the variation of the original set. Hence, given a collection of vectors, we are able to produce a derived set of uncorrelated features. Reducing high-dimensional variable sets to its essence by using the dimensionality-reduction capability of our method is an ideal solution in the case of an expected coexisting, spatially overlapping brain structure covariation pattern. The UK Biobank dataset of brain region volumes happens to be a high-dimensional dataset with a relatively low input-features-to-sample size ratio. Additionally, its features are not necessarily most informative when they are analyzed individually and so we expected the covariation in the brain regions to have an underlying topographical overlap. Hence making use of the dimensionality-reduction property of our method is suitable for the dataset we are using (Bzdok et al., 2019).

We have added these explanations in the revised version of the manuscript as follows:

Methods:

Partial least squares canonical analysis (PLSC) was a natural choice of method to evaluate a relationship between two rich variable sets. **This model class was ideally fitted to our data analysis scenario on grounds of i) feature-to-samples ratio, ii) native auto-correlation in our variable sets with brain-derived measurements, and iii) the latent-factor decomposition capability.**

References:

Helmer M, Warrington S, Mohammadi-Nejad AR, Ji JL, Howell A, Rosand B, Anticevic A, Sotiropoulos SN, Murray JD. On stability of Canonical Correlation Analysis and Partial Least Squares with application to brain-behavior associations.. doi: <https://doi.org/10.1101/2020.08.25.265546>

Wang HT, Smallwood J, Mourao-Miranda J, Xia CH, Satterthwaite TD, Bassett DS, Bzdok D. Finding the needle in a high-dimensional haystack: Canonical correlation analysis for neuroscientists, *NeuroImage*, 216, 2020.

Bzdok D, Nichols TE, Smith SM. Towards Algorithmic Analytics for Large-scale Datasets. *Nature Machine Intelligence*, 1:296-306, 2019.

4. Can the authors address whether they are powered to conduct the partial least squares canonical analysis?

PLSC provides robust and interpretable model fits when evaluating a vast array of features, as demonstrated in UKBiobank-level sample sizes in a comprehensive empirical simulation study (Helmer et al., 2020). We possess a large number of variables for each observation, which likely limits us to linear models. Many methods struggle to interpret weights when attempting to understand the profiles of neural and behavioral attributes that define the brain-behavior relationship due to the overwhelming feature count and limited number of available observations. However, PLSC is able to achieve stability and reliable results with the kind of data we are working with (Helmer et al., 2020).

Our partial least squares canonical analysis revealed 6 modes, clarifying the extent of joint structural covariation between the longitudinal changes within the 18 amygdala subregions and the 109 cortical and subcortical regions. To ascertain the robustness of these 6 modes, we employed three different strategies to ascertain sufficient power in our study:

i) First, we probed the explained variance of every mode. This was achieved by calculating the Pearson rho value between the covariation patterns of the amygdala subregions and those of the (sub)cortical regions for each mode. Through this process, we gauged the extent of correlation between the structural changes in the amygdala subregions and those in the (sub)cortical regions within our participant sample. Strong Pearson rho coefficients, pertaining to the longitudinal changes in both regions, affirmed the stability of the modes generated by our analysis.

ii) Second, we examined the behavior of the association between the inter-individual covariance of the linkage between the longitudinal changes in the amygdala subregions and in the (sub)cortical regions for each significant mode and 977 phenotypes - a large and independent array of external variables. Such a procedure shed light on specific behavioral traits underpinning the coupling of longitudinal changes in our target regions. Any phenotype associations surpassing the FDR threshold within the modes signify an interaction between the amygdala subregions and the (sub)cortical regions, which resonates with the phenotype of importance. By exploring the association between specified phenotypes and the inter-individual covariance of structural change coupling between the amygdala subregions and the (sub)cortical regions across each of the six modes, we can affirm the explanatory worth of the covariance relationships between the amygdala subregions and the (sub)cortical regions in each of the modes.

iii) Our third and final method for ascertaining robustness hinged on an empirical permutation testing framework. Here, we sought to quantify the statistical significance of every mode. This involved assessing the explained variance of all unique components derived and determining the significance of the modes based on their computed Pearson rho values. A series of 100 permutation iterations were executed. During each iteration, the significance of the correlation coefficients from the modes was gauged by producing randomized permuted versions of the

(sub)cortical regions' covariation patterns of longitudinal changes. This was followed by fitting the model using these permuted patterns alongside the amygdala subregional covariation patterns of longitudinal change. Subsequent computations yielded correlation coefficients and scores for each mode, forming an empirical null distribution. These results were juxtaposed against the original observed values from the initial model fitted with unpermuted data, allowing for the derivation of p-values. These p-values then facilitated a comprehensive statistical comparison, highlighting the relative significance of each mode in the model.

References:

Helmer M, Warrington S, Mohammadi-Nejad AR, Ji JL, Howell A, Rosand B, Anticevic A, Sotiropoulos SN, Murray JD. On stability of Canonical Correlation Analysis and Partial Least Squares with application to brain-behavior associations.. doi: <https://doi.org/10.1101/2020.08.25.265546>

5. The bootstrapping described in the methods seems to be the bare minimum to achieve a non-parametric distribution (100 iterations). More iterations (at least 1000) would lead to a distribution that is more likely to reflect the underlying distribution.

We have previously performed 1000 iterations and obtained the same results as 100 iterations. Additionally, 100 iterations is also a standard in our team (Kiesow et al., 2020; Spreng et al., 2020; Saltoun et al., 2023) as well as broadly recommended in classical authoritative sources (Efron and Tibshirani, 1994).

References:

Efron, B., & Tibshirani, R.J. (1994). *An Introduction to the Bootstrap* (1st ed.). Chapman and Hall/CRC. <https://doi.org/10.1201/9780429246593>

Zajner, C., Spreng, R.N., Bzdok, D., 2021. Loneliness is linked to specific subregional alterations in hippocampus-default network covariation. *J Neurophysiol* 126, 2138-2157.

Saltoun, K., Adolphs, R., Paul, L.K., Sharma, V., Diedrichsen, J., Yeo, B.T.T., Bzdok, D., 2022. Dissociable brain structural asymmetry patterns reveal unique phenome-wide profiles. *Nat Hum Behav*.

Kiesow H, Dunbar RIM, Kable JW, Kalenscher T, Vogeley K, Schilbach L, Marquand AF, Wiecki TV, Bzdok D. 10,000 Social Brains: Sex Differentiation in Human Brain Anatomy. *Science Advances*, 6:aaz1170, 2020.

6. The references chosen are quite odd—for example, none of the foundational scientists examining primate amygdala neuroanatomy over the past few decades have been cited (David Amaral, Helen Barbas, Julie Fudge (one paper cited), Joe Price, Lynn Selemon and others).

We thank the reviewer for their suggestions. We have already cited a paper that involves Joe Price and have also already cited papers of many other foundational amygdala scientists such as Joseph Ledoux, John Aggleton, and Ralph Adolphs. We have now expanded our references in the new version of the manuscript to add more foundational amygdala scientists that have examined primate amygdala neuroanatomy.

7. The number of phenotypes created is very large—I am glad that the authors used a Bonferroni correction.

We thank the reviewer for the positive feedback. We have also used a large number of phenotypes in our previous UK Biobank study with phenome-wide association assays (Saltoun et al., 2023) where this approach has yielded robust results.

Saltoun, K., Adolphs, R., Paul, L.K., Sharma, V., Diedrichsen, J., Yeo, B.T.T., Bzdok, D., 2022. Dissociable brain structural asymmetry patterns reveal unique phenome-wide profiles. *Nat Hum Behav.*

8. Were there only 6 modes found?

Our primary analysis produced 6 initial robust modes, from which we extracted the most relevant modes. The robust modes were extracted by evaluating the robustness of the 6 modes using a multi-step approach:

We computed the Pearson rho value to examine the correlation between covariation patterns in the amygdala subregions and the (sub)cortical regions for each mode, gauging the relationship between structural changes in both regions within our sample of participants. Following this, we analyzed the association between the inter-individual covariance of the connectivity between longitudinal alterations in the amygdala subregions and the (sub)cortical regions for each notable mode and 977 phenotypes. This examination illuminated specific behavioral traits associated with the conjoint longitudinal changes in the amygdala subregions and the brain regions. More phenotype associations exceeding the FDR threshold in the modes indicate a tighter link to the structural covariation between the amygdala subregions and (sub)cortical regions that depends on the significant phenotype.

For our final robustness verification, we employed an empirical permutation testing framework. Here, we aimed to quantify the statistical significance of each mode by evaluating the explained variance of all derived unique components and by establishing the significance of the modes through their calculated Pearson rho values. We conducted a total of 100 permutation iterations. In each, we assessed the significance of the mode's correlation coefficients by creating randomized permuted versions of the (sub)cortical regions' covariation patterns of longitudinal changes and then fitting the model with these permuted patterns and the amygdala subregion covariation patterns of longitudinal change. The subsequent calculations produced correlation coefficients and scores for each mode, constituting an empirical null distribution. Comparing these results with the original values from the initial model fitted with unpermuted data enabled us to calculate p-values, which were then utilized to convey the relative statistical significance of each mode in the model.

9. The descriptions of the results are somewhat confusing, as it seems as though the authors have jumped to interpreting, rather than simply reporting, the results. Perhaps a better way to describe the findings are to describe which amygdala nuclei had the greatest parameter weights (to draw parallels to the figures), and then the associated covariance in the cortical areas, stating that this suggests a relationship between these regions. As the results are written currently, they jump to stating there are structural plasticity effects, which is really an interpretation, rather than a description of the results of their PLSC. It also makes the methods-results relationship somewhat confusing. I would recommend that the authors stick to describing the results in the results, and bring in more interpretation in the discussion.

We thank the reviewer for the helpful suggestion. We have toned down some of the interpretation and have removed the mention of 'plasticity' in our results section in the revised version of the manuscript.

Results;

Third pattern emphasizes plasticity coupling of the laterobasal amygdala with prefrontal partners:

The covariation of the structural plasticity in the basal, lateral, accessory basal, and paralaminar subregions and the prefrontal cortex were found to strongly covary in the same direction in this pattern which indicates that the laterobasal subregions and the prefrontal cortex share coupled relationships. ~~These findings appeared to be consistent with the known synonymous roles of the laterobasal subregions and the prefrontal cortex as information decryption/distribution hubs in their respective areas of influence and verifies their known direct axonal connections. The~~

amygdala (Friedman and Robbins, 2022; LeDoux, 2007), in which the prefrontal cortex directly sends information from external stimuli to be processed by the laterobasal subregions and then distributed to the rest of the amygdala.

Lateralization plasticity effects in the anterior amygdaloid area, lateral nucleus and the cortical nucleus with hemispherically-biased brain regions in the second signature :

The unidirectionally hemisphere-biased structural plasticity in the amygdala subregions forms a coupled and cross-hemispheric relationship with the regions in the inferior parietal lobule (SMG and TPJ), while collectively undergoing a driven lateralization effect with the brain regions which have been found to co-form a constellation reminiscent of the salience network (ACC and IC) in the second primary pattern. This hence provides more in-depth more information about the hemispherically-biased tendencies of the salience network uncovered in the primary analysis (cf. Figure 2) to interact with the salient amygdala subregions. and reveals a coupled association between the covarying amygdala subregions and the inferior parietal lobule to regulate alertness and visual consciousness while also revealing a coupled association between the covarying amygdala subregions and the inferior parietal lobule.

Lateralization effects in the cortical, central and medial nuclei in the third signature:

In yet another separable lateralization plasticity signature, our hemispheric difference analysis showed the amygdala subregions to be hemispherically-biased between both hemispheres but with parameter weights that are relatively less strong in this third signature. We observed hemispherically-biased plasticity configurations occurring in the medial nucleus, and yet more strongly in the central nucleus and in the cortical nucleus. The centromedial amygdala subregions (central and medial nuclei) were found to be driven by a collective lateralization in the same hemisphere as the accessory basal nucleus and the cortical nucleus, while all the other subregions collectively lateralized to the other hemisphere. The lateralization in the amygdala subregions occurs mainly due to the collective intra-amygdala interplay of the subregions with each other, which helped convey where on the spectrum of responsibilities each amygdala from both hemispheres center via the examination of the primary domain of influence of the most hemispherically-biased subregions in each hemisphere (Pitkanen et al., 1997).

10. For Model 2, why is the Accessory Basal not discussed? It seems to have a relatively higher parameter weight.

When taking both hemispheres into consideration, the association strength of the accessory basal nucleus in the second pattern experiences relatively lower covarying longitudinal changes in the left and right hemisphere compared to the medial, cortical, lateral and central amygdala nuclei. The subregions mentioned of considerable covarying longitudinal changes showed considerable strength in both hemispheres in this particular pattern, and based on this trend, the accessory basal nucleus did not stand out when compared with the other nuclei.

On the other hand, if the accessory basal nucleus showed an exceptional association strength in one of the hemispheres, high enough to go above and beyond the lack of association strength in the other hemisphere, then it would have been essential to account for.

11. Model 3 appears to be somewhat more generalized to the entire amygdala—the authors point out the subregions with relatively greater change, but really, the map here is strikingly different from the prior two maps, with all amygdala nuclei having changes in the same direction. Perhaps relatedly, the strength of association with the cortical areas is somewhat weaker compared to the prior models. It would be helpful if the authors discussed why this may be the case in this model.

It's essential to emphasize that, by design, the modes in our analysis are sorted from strongest to weakest. Therefore, observing somewhat diminished region weights and association strengths in subsequent modes is an anticipated characteristic of the fits obtained with this model, reflecting the decreasing explanatory power of from component to component, relative to the earlier ones. The reduced association strength in mode 3, as compared to prior modes, is consistent with this methodological approach and underscores the inherent decrease of explanatory power in the successive modes. Furthermore, the universality of alterations across all amygdala nuclei in mode 3 may inherently manifest a more distributed, albeit weaker, set of associations, potentially diluting the strength of specific region-to-region correspondences observed in earlier, more localized modes.

We did point out the subregions with the relatively greater change, which are the subregions in the larger laterobasal subdivision. We also pointed out that we think the reason the subregions in the laterobasal subdivision are covarying with the brain regions in the prefrontal cortex, is because of both of their roles as information decryption/distribution hubs in their respective areas of influence. While this is the case, we also think the nature of the third pattern in terms of direction and magnitude of all the amygdala subregions, is because of the information distribution occurring from the laterobasal subregions to the rest of the amygdala subregions. The covarying amygdala subregions and brain regions in this pattern, other than the laterobasal amygdala subregions and the prefrontal cortex regions, are also a part of this circuit of information distribution as receivers of information distributed from the amygdala subregions in the laterobasal amygdala subdivision and the prefrontal cortex.

Addressing your insightful comment, we would add that regardless of the precise impetus for the change in the laterobasal amygdala subregions, it appears to be tracking change variation distributed across the entirety of the amygdala as a whole. This suggests a driving effect situated in the laterobasal amygdala subregions, which is intrinsically linked to changes in the rest of the amygdala subregions.

Further, it is crucial to underline once again that the essence of our analysis hinges on the relative differences in the coefficients between each other. The precise numerical values themselves do not have importance but are only valuable in their relational context to one another.

12. In Figures 1-5, in the cortex, the heat scales that are set have a color for zero (blue or red, for instance), yet not every region of the cortex is colored. Can the authors provide an explanation for why not every region of the cortex is colored?

In Figures 1-3 all the cortical and subcortical regions are coloured because all coefficients are relevant and the mode as a whole is significant (not single subregions). On the other hand, given the nature of the analysis in Figures 4 and 5, the results happen to show which cortical/subcortical regions are experiencing lateralization in the covariance of their longitudinal changes with the amygdala subregions. The results also show the magnitude of the lateralized covariance in the regions showing lateralization among the 109 cortical/subcortical brain regions.

In response to this reviewer comment, we have added an explanation for all relevant figures that unpack why certain regions may appear to have no color weight.

Figure 4. Lateralization plasticity effects driven by the cortical nucleus, anterior amygdaloid area, central nucleus and lateral nucleus co-vary with awareness/alertness-related brain regions. We performed a hemispheric difference analysis in the context of the left-right divergence of the structural plasticity changes in 9 amygdala subregions with the structural plasticity of 109 brain regions by means of co-decomposition based on partial least squares canonical (PLSC). We determined how the ensuing subregion patterns lateralized in the 9 amygdala subregions and which cortical/subcortical regions are experiencing lateralization in the covariance of their longitudinal changes with the lateralized amygdala subregions. The results also show the magnitude of the lateralized covariance in the regions experiencing lateralization among the 109 cortical/subcortical brain regions and among the 18 amygdala subregions. Shown here is the subregion lateralization in mode 1 of the amygdala - brain covariation. A) conveys the direction of lateralization of each of the 9 amygdala subregions in mode 1 (Saygin ZM & Kliemann D et al., 2017). The parameter weights of the subregions that diverge between both hemispheres are depicted on 2 columns of 4 coronal slices of the amygdala parcellated into 9 subregions with each column portraying a different direction of lateralization occurring in each hemisphere.

Figure 5. Lateralization plasticity effects driven by the anterior amygdaloid area, lateral nucleus, and cortical nucleus with a lateralization effect in awareness/alertness-related brain regions together with the inferior parietal lobule. We determined how the ensuing subregion patterns lateralized in the 9 amygdala subregions and which cortical/subcortical regions are experiencing lateralization in the covariance of their longitudinal changes with the lateralized amygdala subregions. The results also show the magnitude of the lateralized covariance in the regions experiencing lateralization among the 109 cortical/subcortical brain regions and among the 18 amygdala subregions. Shown here is the subregion lateralization in

signature 2 of the amygdala - brain covariation. A) conveys the direction of lateralization plasticity effects of each of the 9 amygdala subregions in signature 2 (Saygin ZM & Kliemann D et al., 2017). The parameter weights of the subregions that diverge between both hemispheres are depicted on 2 columns of 4 coronal slices of the amygdala parcellated into 9 subregions with each column portraying a different direction of lateralization occurring in each hemisphere.

Supplementary Figure 1: Lateralization plasticity effects driven by the cortical, central, and medial nuclei. We determined how the ensuing subregion patterns lateralized in the 9 amygdala subregions and which cortical/subcortical regions are experiencing lateralization in the covariance of their longitudinal changes with the lateralized amygdala subregions. The results also show the magnitude of the lateralized covariance in the regions experiencing lateralization among the 109 cortical/subcortical brain regions and among the 18 amygdala subregions. Shown here is the subregion lateralization in signature 3 of the amygdala - brain covariation. A) conveys the direction of lateralization of each of the 9 amygdala subregions in signature 3 (Saygin ZM & Kliemann D et al., 2017). The parameter weights of the subregions that diverge between both hemispheres are depicted on 2 columns of 4 coronal slices of the amygdala parcellated into 9 subregions with each column portraying a different direction of lateralization occurring in each hemisphere.

13. Can the authors provide an explanation for why they did two separate analyses for combined R/L amygdala and R and L amygdala separately? What is the advantage of doing these both? Why wouldn't the combined R/L amygdala analysis be the most comprehensive analysis? If the separate R/L amygdala analyses do not add much, the authors could consider whether to place these analyses in a supplement.

In short, the answer is that we wanted to directly contrast the left-amygdala-brain and right-amygdala brain with each other, which we have achieved by direct model comparison to tease apart their relevant differences.

In particular, we hypothesized that certain amygdala subregions, which we would identify as being associated with (sub)cortical regions, would exhibit lateralization in their longitudinal changes. This expectation was based on previous imaging studies revealing unique lateralization patterns within the larger amygdala subdivisions (Ball et al., 2007). We anticipated observing lateralization in specific subregions of each amygdala subdivision. Our hypothesis extended to the likelihood that the (sub)cortical regions experiencing covarying longitudinal changes with amygdala subregions would also demonstrate lateralization in their structural couplings.

In the goal to peel apart the nuanced hemispheric asymmetries between amygdala-brain couplings, we aspire to orchestrate a formal comparison between the left and right cortex and explore how each hemisphere individually evolves in conjunction with the amygdala subregions. Considering that our current model employs a double-multivariate approach, specifically tailored for amygdala-cortex covariation analysis, to facilitate a robust contrast analysis for the left versus right hemisphere presents a compelling challenge. Thus, we have conceived a 'Hemispheric Difference Analysis' to reveal lateralized effects in the covariance of the longitudinal changes in the (sub)cortical regions with the longitudinal changes in the amygdala subregion. This innovative approach enables us to dive deeper into the differences and commonalities between the two hemispheres, identifying not only their individual changes but also how these modifications covary with alterations in the amygdala subregions.

References:

Ball T, Rahm B, Eickhoff SB, Schulze-Bonhage A, Speck O, Mutschler I. Response properties of human amygdala subregions: evidence based on functional MRI combined with probabilistic anatomical maps. PLoS One. 2007 Mar 21;2(3):e307. doi: 10.1371/journal.pone.0000307. PMID: 17375193; PMCID: PMC1819558.

14. Please fix the figure callout on line 615.

We thank the reviewer for the pointers, we have fixed the spotted mistake in the revised version of the manuscript:

In our phenome-wide assays, the first pattern of the primary analysis showed 34 (Bonferroni's correction for multiple comparisons) and 65 (above the FDR threshold) significant associations with target phenotypes **Supplementary Figure 2**.

15. As the authors have defined the circuits that have similar GM changes, it might be helpful to have a summary figure that depicts the three types of circuits/relationships from their analysis.

We thank the reviewer for the insightful suggestion. We believe that our present figures already encapsulate a division that echoes the three types of circuits/relationships derived from our analysis, with a distinct emphasis placed on circuit relations in each title. While we value the proposition of introducing a summarizing figure for clarity and conciseness, we also conscientiously seek to balance comprehensiveness with simplicity in our visual representations.

We have created a new figure, showing the different types of circuits/relationships between the amygdala subregions and the covarying brain networks:

Supplementary Figure 7. Different types of circuits/relationships between the amygdala subregions (Elivera et al., 2022) and the covarying brain networks A) Plasticity coupling in the first pattern of the central nucleus and the anterior amygdaloid area with right parietal cortex B) Plasticity coupling of the medial, cortical and central amygdala with brain regions related to alertness and visual conscious awareness C) Plasticity coupling of the laterobasal amygdala subregions with prefrontal cortex

References:

Elvira UKA, Seoane S, Janssen J, Janssen N (2022) Contributions of human amygdala nuclei to resting-state networks. PLoS ONE 17(12): e0278962. <https://doi.org/10.1371/journal.pone.0278962>

16. I am not sure what to make of the phenotypic findings—could the authors provide some interpretation of the findings? What is the specificity of these findings to the different amygdala circuits? These are all associative findings, and it is hard to describe these as causative.

Manhattan plots offer a comprehensive framework for interpreting phenome-wide association studies, thereby aiding in evaluating brain-behavior connections across extensive phenotypic domains. We have created a Manhattan plot for every association pattern of distinct amygdala subregions-brain region covariations produced in our primary analysis, with every pattern having its own unique set of significant phenotypes that go above the correction thresholds within the Manhattan plot. We chose to hone on specific significant phenotypes above the correction threshold of the phenome-wide analysis of every association pattern in the text, to depict broader phenomena. For instance, the third pattern, which showed longitudinal changes in the basal, lateral, accessory basal and paralaminar nuclei, identified by the primary analysis

revealed phenotype associations connected to income, work status, sleep, risk-taking, and regular leisure activities—all surpassing the Bonferroni correction threshold. Thus, we summarized the disclosed amygdala subregion longitudinal changes -(sub)cortical region longitudinal changes associations in the third pattern as being related to phenotypes contributing to both mental health and socioeconomic status.

We have edited our manuscript to clarify the non-causal nature of our findings:

Figure 6. Phenome-wide assay spotlights phenotypes related to socioeconomic status, work status, sleep, risk taking and leisure regular activities. A) Manhattan plot shows phenotype associations with individual expressions in the third plasticity pattern (cf. Figure 3) in the UKBiobank population which charts 977 lifestyle indicators related variables divided across 11 domains. For each phenotype, the plasticity-behavior links are shown in units as p-values (-log. scale). Horizontal lines indicate the significance thresholds at Bonferroni correction and at FDR correction for phenotypes (0.05/977). 146 phenotypes exceeded the FDR threshold and 79 exceeded the Bonferroni threshold in the third pattern. **These significant phenotypes do not endorse or imply causality, but rather afford a valuable lens through which the amygdala-brain covariations can be contextualized** B) shows the median expression of the covariation pattern in the cortical and subcortical regions across age and sex which classify the differences in sex and age pattern strength. Error bars illustrate the lower 5th percentile and upper 95th percentile thresholds obtained by bootstrapping the median of the population, and two lines of best fit to the data are shown: the purple line corresponds to the female data while the blue line corresponds to the male data. The phenotype analysis showed the most significant associations with physical characteristics such as body fat percentage, phenotypes related to social activities, physical health of parents, household income, employment status, alcohol consumption related phenotypes while other significant phenotypes found in this analysis are haemoglobin concentration, ventricular depolarization (QRS Duration), professional help for nerves, anxiety tension or depression, sleeplessness and insomnia, and balding pattern.

Supplementary Figure 2. Bonferroni threshold passed by body constitution, liver health marker (glutamyl transferase), and blood work indicator phenotypes linked to dominant AM-brain plasticity pattern.A) Manhattan plot shows phenotype associations with individual (sub)cortical regions expressions in the first plasticity pattern (cf. Figure 1) in the UK Biobank population which charts 977 lifestyle indicators related variables divided across 11 domains. For each phenotype, the plasticity behaviour-links are shown as p-values (log. scale). Horizontal lines indicate the significance thresholds at Bonferroni correction (0.05/977) and at FDR correction for phenotypes. 65 phenotypes exceeded the FDR threshold and 34 exceeded the Bonferroni threshold in the first pattern. **These significant phenotypes do not endorse or imply causality, but rather afford a valuable lens through which the amygdala-brain covariations can be contextualized.** B) shows the median expression of the covariation pattern in the cortical and subcortical regions across age and sex of the first pattern which classify the differences in sex and age pattern strength. Error bars illustrate the lower 5th percentile and upper 95th percentile thresholds obtained by bootstrapping the median of the population, and two lines of best fit to the data are shown: the purple line corresponds to the female data while the blue line corresponds to the male data. The phenome-wide analysis showed significant associations with phenotypes related to body constitution phenotypes under the physical-general domain, liver health marker (glutamyl transferase) and blood work indicator phenotypes under the blood-assays domain which in turn helps translate the nature of the association that the primary analysis (cf. Figure 1) revealed between the central nucleus/anterior amygdaloid area and the inferior parietal lobule.

Supplementary Figure 3. Bonferroni threshold passed by household size, educational attainment, phone use frequency, smoking habits and cardiovascular risk factor phenotypes. A) Manhattan plot shows phenotype associations with individual (sub)cortical regions expressions in the second plasticity pattern (cf. Figure 2) in the UK Biobank population which charts 977 lifestyle indicators related variables divided across 11 domains. For each phenotype, the plasticity behaviour-links are shown as

p-values (log. scale). Horizontal lines indicate the significance thresholds at Bonferroni correction (0.05/977). and at FDR correction for phenotypes. 70 phenotypes exceeded the FDR threshold and 31 exceeded the Bonferroni threshold in the second pattern. **These significant phenotypes do not endorse or imply causality, but rather afford a valuable lens through which the amygdala-brain covariations can be contextualized.** B) shows the median expression of the covariation pattern in the cortical and subcortical regions across age and sex of the second pattern which classify the differences in sex and age pattern strength. Error bars illustrate the lower 5th percentile and upper 95th percentile thresholds obtained by bootstrapping the median of the population, and two lines of best fit to the data are shown: the purple line corresponds to the female data while the blue line corresponds to the male data. The phenotype analysis showed significant associations with phenotypes related to financial well-being, number of people living in the household and if the people in the household are sons/daughters, physical health of the mother while other significant phenotypes found in this analysis are related to carotid IMT and phenotypes which are observed in cognitive tests.

17. I found the discussion to be a bit unfocused. I would suggest tightening the sections on the salience network, as these are more established findings within the literature, and focusing the discussion on the circuits identified in the results.

We thank the reviewer for the pointer. We have incorporated additional signposts to enhance navigational ease for the reader in our reviewed version of the manuscript.

18. The discussion continues to focus on plasticity, without clear evidence that the findings represent plastic changes. I would suggest that the authors focus on the circuits that they identify as having similar longitudinal changes, as these findings may not necessarily represent 'plastic' changes.

We thank the reviewer for their helpful suggestion. We have defined a clear definition for plasticity in the introduction (cf. above), Previous brain-imaging papers defined 'plasticity' similarly, which helps give a basis for our discussion.

Reviewer #3 (Remarks to the Author):

The authors study the amygdala's temporal dynamics and its connections with cortical and subcortical areas. The authors contend that much of the research has been focused on the amygdala's role in fear recognition, often overlooking its contributions to memory, attention, decision-making, and other processes that are connected to cortico-subcortical circuits.

The study aims to map the changes that show co-variation between amygdala subregions and cortical partners, shedding light on the structural plasticity (longitudinal changes over years) of these interconnected brain areas. They perform an additional

analysis linking these amygdala-cortical changes with lifestyle measures that reflect behavior and cognition, providing a broader understanding of the amygdala's function.

The study utilizes the UKBiobank initiative which was able to scan the brains of over 1,400 healthy participants at two different timepoints, with a delay of 2-3 years between each, providing more authentic insights into amygdala-brain-behavior changes.

The study uses a detailed approach, analyzing 9 amygdala subregions and various behavioral, lifestyle, and mental health indicators. This represents a departure from past research, which often examined the amygdala as a single region or under three broader subdivisions. It is hoped that this more granular approach will allow for a better understanding of the amygdala's function and plasticity in response to various external stimuli and life factors.

Advantages:

Longitudinal Analysis: Longitudinal studies provide a significant advantage over cross-sectional studies in studying brain development, as they can track changes within the same individuals over time. This approach provides more reliable data about individual changes and could have greater potential for understanding causal relationships.

Large Sample Size: The study uses a large sample size of over 1,400 participants from the UKBiobank, which increases the reliability of the results and allows for the detection of smaller effects that may be missed in studies with fewer participants.

Detailed Analysis: The detailed analysis of 9 amygdala subregions is a significant step forward from the common approach of studying the amygdala as a whole or in a few broad subdivisions. This could provide new insights into the specific functions and connections of these subregions.

Connection to Lifestyle Factors: By connecting the analysis to around 1,000 behavioral, lifestyle, and mental health indicators, the study aims to provide a more holistic understanding of the brain's operations and adaptations in real-world contexts.

We are thankful for the favorable assessment of our work.

Potential Limitations:

Generalizability: While the large sample size is a strength, the participants are all from the UKBiobank. This could limit the generalizability of the findings to other populations, particularly those with different demographic characteristics or lifestyle factors.

We thank the reviewer for the helpful pointer. While the UK Biobank aims to approximate the representativeness and diversity of the UK population, it is nearly representative, although there is some skewing from an overall lower rate of all-cause mortality and total cancer incidence compared to the general population. Despite this, the assessment of exposure-disease relationships conducted using the UK Biobank may still be widely generalizable and does not necessitate the participants to be completely representative of the larger population (Batty et al., 2019; Fry et al., 2017). Consequently, this suggests that the findings derived from this data set may have broader implications to the UK and other geographies with similar sociodemographics and backgrounds, such as the US and Canada. The similarity in lifestyle, healthcare systems, and demographic structure between the UK and other Western countries in Europe and North America can potentially make the results relevant and applicable for these regions. Nevertheless, we agree with the reviewer that caution should be exercised when generalizing the findings to other cohorts, considering that certain socio-cultural and environmental factors might vary across different countries.

References:

G. David Batty, Catharine R. Gale, Mika Kivimäki, Ian J. Deary, Steven Bell
.Generalisability of Results from UK Biobank: Comparison With a Pooling of 18 Cohort Studies.
medRxiv 19004705; doi: <https://doi.org/10.1101/19004705>

Fry A, Littlejohns TJ, Sudlow C, Doherty N, Adamska L, Sprosen T, Collins R, Allen NE.
Comparison of Sociodemographic and Health-Related Characteristics of UK Biobank
Participants With Those of the General Population. *Am J Epidemiol.* 2017 Nov 1;186(9):1026-
1034. doi: 10.1093/aje/kwx246. PMID: 28641372; PMCID: PMC5860371.

Time Delay: The 2-3 year delay between brain scans could be both a strength and a limitation. While it allows for the detection of longitudinal changes, it may also miss more rapid changes that occur on a shorter timescale.

Investigating longitudinal changes is particularly advantageous when exploring the impact of behavioral traits and lifestyle choices which affect the structural plasticity of involved regions in the brain and subregions in the amygdala. Past literature on longitudinal studies have shown how training in emotional empathy and language proficiency leads to structural changes in specific brain regions, emphasizing the adaptability and specificity of brain plasticity in response to different learning experiences during a span of multiple months or several years (Valk et al., 2017; Stein et al, 2012).

In this context, a 2-3 year delay between brain scans presents a clear strength. This rarely available time interval allows researchers to observe the cumulative and sustained effects of life factors and behavioral traits on brain structure, capturing the longitudinal changes in response

to shifts or adaptations of diverse mental capacities. By focusing on these long-term alterations, the study can uncover the persistent and potentially more impactful adaptations in the brain that correlate with the enhancement of distinct mental systems.

However, we agree with the reviewer that it is also crucial to acknowledge the limitations of time delay between brain scan acquisitions. The longitudinal perspective is invaluable in tracking sustained changes, which are stable over extended time periods. Yet, as the reviewer said, it might miss transient, rapid modifications that arise and potentially dissipate between the scans. These more immediate brain responses to training could provide additional insights into the adaptability of the brain and the initial stages of structural modifications, thereby contributing to a more nuanced understanding of how distinct mental systems influence gray matter.

The best way rapid changes could be taken into consideration within the context of our analyses is if there exists a dataset that specifically measures transient and rapid modifications in the brain structure, which we could use to run another set of analyses with that dataset. In doing so, we could take into consideration the results of such an analysis in addition to our already available results to paint a more detailed picture about the long-term changes and the short-term changes in a same study cohort - importantly, we are not aware of a UKBiobank-like dataset with similar deep phenotyping and short as well as long-term time delays between brain-imaging acquisitions. The additional issue with such an analysis is that rapid changes would be more likely due to transient life experiences unique to every single individual within the sample. This would affect the reliability of the results in terms of generalizability. The effects of unique individual experiences which cause rapid changes in the brain are harder to generalize from a large sample set like ours to the rest of the population.

The benefits of having a large dataset with precise measurements of longitudinal changes is the ability to extrapolate a more significant and generalized consensus on more trait-like, rather than transient, brain phenotypes; with their Phewas links across 11 rich domains.

References:

Valk, S. L., Bernhardt, B. C., Trautwein, F. M., Bockler, A., Kanske, P., Guizard, N., Collins, D. L., & Singer, T. (2017). Structural plasticity of the social brain: Differential change after socio-affective and cognitive mental training. *Sci Adv*, 3(10), e1700489. <https://doi.org/10.1126/sciadv.1700489>

Stein, M., Federspiel, A., Koenig, T., Wirth, M., Strik, W., Wiest, R., Brandeis, D., & Dierks, T. (2012). Structural plasticity in the language system related to increased second language proficiency. *Cortex*, 48(4), 458-465. <https://doi.org/10.1016/j.cortex.2010.10.007>

Structural Plasticity Measures: The use of gray matter volume as a measure of structural plasticity may not reflect the complexity of neural changes. It can capture large-scale

alterations, but smaller scale, yet significant, changes might not be adequately represented.

We thank the reviewer for the helpful suggestion. In alignment with widely acknowledged scientific norms, our adoption of the notion of plasticity is consistent with methodologies and theoretical underpinnings espoused in previously published, well-cited papers - these previously published papers have used the same notion of plasticity from sMRI measurements. Eminent studies have exploited sMRI to extrapolate and interpret changes in the brain's physical and functional structure, thereby providing crucial insights into the nature and scope of neuroplasticity (Zatorre et al., 2012; Maguire et al., 2000; Lovden et al., 2013).

Utilizing sMRI, these research endeavors have correlated volumetric and morphological changes within the brain to behavioral, cognitive, and environmental shifts, elucidating the brain's remarkable ability to remodel and reorganize itself in response to varied stimuli (Valk et al., 2017; May, 2011). The significance of utilizing diffusion MRI to interpret structural changes and elucidate neural processes has also been delineated in prior research (Assaf and Johansen-Berg, 2013).

Consequently, our work is a natural continuation of these established practices, applying a similar lens of structural plasticity to delve into the complexities of brain-behavior interactions and their longitudinal implications (Taubert et al., 2010). This approach not only facilitates a nuanced understanding of neural adaptability and transformation but also ensures that our findings can be contextualized and critiqued within a broader, established scientific framework, encouraging accumulation of scientific knowledge and interpretations seamlessly dovetail with earlier studies in the realm of neuroplasticity.

References:

- Zatorre, R. J., Fields, R. D., & Johansen-Berg, H. (2012). Plasticity in gray and white: neuroimaging changes in brain structure during learning. *Nature Neuroscience*, 15(4), 528-536.
- Maguire, E. A., Gadian, D. G., Johnsrude, I. S., Good, C. D., Ashburner, J., Frackowiak, R. S., & Frith, C. D. (2000). Navigation-related structural change in the hippocampi of taxi drivers. *Proceedings of the National Academy of Sciences*, 97(8), 4398-4403.
- Lovden, M., Wenger, E., Martensson, J., Lindenberger, U., & Backman, L. (2013). Structural brain plasticity in adult learning and development. *Neuroscience & Biobehavioral Reviews*, 37(9), 2296-2310.
- Valk, S. L., Bernhardt, B. C., Trautwein, F. M., Bockler, A., Kanske, P., Guizard, N., ... & Singer, T. (2017). Structural plasticity of the social brain: Differential change after socio-affective and cognitive mental training. *Science Advances*, 3(10), e1700489.

May, A. (2011). Experience-dependent structural plasticity in the adult human brain. *Trends in Cognitive Sciences*, 15(10), 475-482.

Assaf, Y., & Johansen-Berg, H. (2013). The role of diffusion MRI in neuroscience. *NMR in Biomedicine*, 26(7), 849-865.

Taubert, M., Draganski, B., Anwander, A., Müller, K., Horstmann, A., Villringer, A., & Ragert, P. (2010). Dynamic properties of human brain structure: learning-related changes in cortical areas and associated fiber connections. *Journal of Neuroscience*, 30(35), 11670-11677.

We have amended the introduction to take the reviewer's note into account as follows:

Introduction: Depending on the degree of implication of a brain region in supporting a given behavioral function, it is susceptible to undergo remodeling alterations that are reflected in measures of gray matter volume. The change in gray matter volume in turn signifies not only an ability to learn, but also an enhancement of existing cognitive capabilities and probably also strengthening areas where neural processes are attenuated in other parts of the brain for the sake of compensation. Being able to examine the brain through the lens of structural plasticity provides a window into the effects of various exposures and prompted behaviors in everyday life. We are able to understand how different behavioral traits and lifestyle habits show links to the structural plasticity of a brain region through pattern learning which allows us to make steps towards more causally qualified statements about the brain (Valk et al., 2017) (Lovden et al., 2010; Mateos-Aparicio and Rodriguez-Moreno, 2019)

Structural plasticity refers to the brain's ability to undergo physical and functional reorganization in response to experiences, learning, and various environmental stimuli. This encompasses modifications in the neuronal connections, synaptic vesicle formation and uptake, neuronal remodeling, myelination, and even the observable changes in grey matter volume (Assaf 2018). As shown by past imaging and behavioral studies, such adaptability is not limited by age or a particular cognitive domain (Zatorre et al., 2012). Whether in behavioral realms extending beyond mere motor capabilities or within more specialized cognitive capacities like language or empathy, the brain's structural constitution is known to reflect these experiences (Zatorre et al., 2012).

Central to understanding this notion of change is the comparison between cross-sectional and longitudinal studies. Past longitudinal analyses have shown their pivotality in comprehending lifelong neuroplasticity (Di Biase et al., 2023). Evidence from past brain-imaging/behavioral studies underscores the sensitivity and specificity of individual-level brain modifications: for instance, training targeted at enhancing distinct empathy systems leads to unique structural gray matter modifications (Valk et al., 2017). Similarly, language acquisition in teenagers has demonstrated distinct correlations with changes in grey matter density, emphasizing the dynamism of the brain's structural makeup in relation to evolving mental capacities (Stein et al.,

2012). Furthermore, lateralization plays a role, with certain cognitive functions such as language likely favoring one hemisphere over the other, leading to discernible structural alterations in specific brain regions (Assaf, 2018; Valk et al., 2017; Taubert et al., 2010; Stein et al., 2012).

Past animal studies, conducted on rats, mice, and monkeys, have started to bridge the understanding between MRI-observable changes and the cellular architectures underlying them (Zatorre et al., 2012; Caroni et al., 2012). Notably, investigations spanning humans and rodents have identified commonalities in structural brain changes following task-based training, reinforcing the universality of structural plasticity mechanisms (Sagi et al., 2012).

In essence, structural plasticity epitomizes the brain's evolutionary advantage to adapt and evolve in response to a wide array of stimuli and experiences. This malleability, observable through advanced brain-imaging techniques and affirmed through various experimental designs, underpins the rationale for structural plasticity being the main driver for longitudinal studies that track and map the continuum of brain changes over different timescales.

References:

- Valk, S. L., Bernhardt, B. C., Trautwein, F. M., Bockler, A., Kanske, P., Guizard, N., Collins, D. L., & Singer, T. (2017). Structural plasticity of the social brain: Differential change after socio-affective and cognitive mental training. *Sci Adv*, 3(10), e1700489. <https://doi.org/10.1126/sciadv.1700489>
- Assaf, Y. (2018). New dimensions for brain mapping. *Science*, 362(6418), 994-995. <https://doi.org/10.1126/science.aav7357>
- Zatorre, R. J., Fields, R. D., & Johansen-Berg, H. (2012). Plasticity in gray and white: neuroimaging changes in brain structure during learning. *Nat Neurosci*, 15(4), 528-536. <https://doi.org/10.1038/nn.3045>
- Di Biase, M. A., Tian, Y. E., Bethlehem, R. A. I., Seidlitz, J., Alexander-Bloch, A. F., Yeo, B. T. T., & Zalesky, A. (2023). Mapping human brain charts cross-sectionally and longitudinally. *Proc Natl Acad Sci U S A*, 120(20), e2216798120. <https://doi.org/10.1073/pnas.2216798120>
- Stein, M., Federspiel, A., Koenig, T., Wirth, M., Strik, W., Wiest, R., Brandeis, D., & Dierks, T. (2012). Structural plasticity in the language system related to increased second language proficiency. *Cortex*, 48(4), 458-465. <https://doi.org/10.1016/j.cortex.2010.10.007>
- Taubert, M., Draganski, B., Anwander, A., Müller, K., Horstmann, A., Villringer, A., & Ragert, P. (2010). Dynamic properties of human brain structure: learning-related changes in cortical areas and associated fiber connections. *J Neurosci*, 30(35), 11670-11677. <https://doi.org/10.1523/JNEUROSCI.2567-10.2010>

Caroni, P., Donato, F., & Muller, D. (2012). Structural plasticity upon learning: regulation and functions. *Nat Rev Neurosci*, 13(7), 478-490. <https://doi.org/10.1038/nrn3258>

Sagi, Y., Tavor, I., Hofstetter, S., Tzur-Moryosef, S., Blumenfeld-Katzir, T., & Assaf, Y. (2012). Learning in the fast lane: new insights into neuroplasticity. *Neuron*, 73(6), 1195-1203. <https://doi.org/10.1016/j.neuron.2012.01.025>

Lack of Functional Data: While the focus on structural changes is important, it could be complemented by functional data to provide a more comprehensive picture of how these changes influence brain function and behavior.

We thank the reviewer for their constructive suggestion. Freesurfer is predominantly used for analyzing and visualizing structural MRI data and Freesurfer granularity is not available in fMRI. By focusing on structural alterations, our study lays a solid foundation for future research that can incorporate functional data.

Reliance on Self-Reported Measures: This study likely relies on self-reported measures for lifestyle and mental health indicators. These measures can be subject to bias and inaccuracies, which could impact the results.

We thank the reviewer for bringing up a great pointer. While it is true that reliance on self-reported measures can introduce elements of bias and inaccuracies, it is essential to consider the broader context in which these measures are used. A significant portion of psychological research, particularly in the 20th century, has been underpinned by self-reported measures. Despite their inherent limitations, such measures have played a pivotal role in advancing our understanding of human behavior and mental processes, influencing both academic research and notions widely entertained in society (e.g., IQ, personality traits, growth mindset etc. - all widely known concepts).

Moreover, it's noteworthy that this study utilizes data from the UK Biobank, which is renowned for being one of the largest, most comprehensive, and most expensive biomedical datasets, not only in the Western world but worldwide. Esteemed for its sheer scale and diversity, the dataset is uniquely poised to help mitigate some of the potential inaccuracies through coordinated expert-guided planning of the battery of self-reported measures and thereby facilitating the drawing of more reliable conclusions. It's important to highlight that the questionnaires utilized in this study were meticulously designed by experts, further bolstering the reliability and validity of the self-reported data within this expansive and unparalleled research endeavor.

That being said, the acknowledgment of the limitations of self-reported measures is crucial, and future studies might benefit from incorporating additional objective measures where possible, to

validate and complement the findings derived from self-reported data. This multi-method approach would serve to bolster the reliability and validity of the research outcomes.

Specifically, the individual figures need to be updated to reflect the statistical measures directly on the brain images to include the strength of associations across each brain region.

The colors depicted across the brain regions in our images signify the robust associations unearthed through our Partial Least Squares Canonical (PLSC) analysis, which meticulously evaluated the interconnected structural plasticity within amygdala subregions and 109 (sub)cortical territories. PLSC does not only decipher these intricate neural correlations but also visualizes the magnitude and direction of these associations, thereby offering profound insights into the brain's structural covariance. The color gradients employed in the brain mappings are a direct translation of these complex relationships, reflecting the relative strength of associations and changes in gray matter volume — a testament to the nuanced interplay of brain regions identified by our rigorous PLSC model. We have shown color scales that reflect the statistical measure of the magnitude of association in the amygdala subregions and the brain regions. We have revised the color bars in our figures to explicitly state this as such:

There are grammatical issues throughout the manuscript that need to be corrected. As a specific example: "This hence provides more in-depth more information about the hemispherically-biased tendencies of the salience network uncovered in the primary analysis (cf. Figure 2) to interact with the salient amygdala subregions to regulate alertness and visual consciousness while also revealing a coupled association between the covarying amygdala subregions and the inferior parietal lobule."

Thanks, we have carefully assessed language by an English native speaker in the entire manuscript, as a consequence to this reviewer feedback. We have corrected the example mentioned by the reviewer as follows:

Results:

This, therefore, offers more in-depth information about the hemispherically-biased tendencies of the salience network uncovered in the primary analysis (cf. Figure 2) and reveals a coupled association between the covarying amygdala subregions and the inferior parietal lobule.

The strength of associations to the behavioral/social measures is not indicated. This is a major problem and needs to be corrected/included.

In our Manhattan plots, the y-axis represents the statistical significance of the association between the amygdala-brain effects and 977 phenotypes that represent the behavioral/social measures. We have now added a table for each pattern that lists all significant effects, with their precise numerical values of effect sizes for future readers who are interested in looking up this information (Supplementary Table 1, Supplementary Table 2, Supplementary Table 3, Supplementary Table 4).

In particular, this transformation is applied to the p-values to both magnify the differences and make these values more manageable for visualization. A p-value is a measure of the evidence against a null hypothesis, and the smaller the p-value, the stronger the evidence for the null hypothesis of absent association between our brain structure patterns and a target phenotype of interest. By transforming the p-values using $-\log_{10}(p)$, smaller p-values (indicating stronger associations) will be represented by higher values on the y-axis. In this way, the higher the value on the y-axis, the stronger the association between the genetic variant and the phenotype in question. Hence, the current y axis label represents the strength of associations to the behavioral/social measures.

Overall, while this study provides a detailed structural analysis of the amygdala as it correlates with (sub)cortical structures, I found the behavioral/social measures somewhat arbitrarily chosen. However, this is my personal bias and cannot be used against the authors.

We appreciate the helpful suggestion from the reviewer. The 977 phenotypes analyzed which represent the behavioral/social traits were directly obtained from the UKBiobank, with automated pre-processing steps such as FUNPACK (created by the FMRIB team responsible for the brain-imaging arm of the UKBiobank initiative) and PHESANT.

The Manhattan plots serve to summarize the results of phenome-wide association studies, offering a framework that aids in interpreting the associations between brain and behaviour across 11 broad extensive phenotypic domains that classify 977 Biobank phenotype variables (Miller et al., 2016). Consequently, we opted to focus on specific instances in the text to depict broader phenomena. For instance, the third pattern observed in the intrinsic plasticity coupling analysis displayed associations with phenotypes such as income, work status, sleep, risk-taking, and regular leisure activities, all surpassing the Bonferroni correction threshold. Therefore, we summarized the uncovered associations as being related to contributors to both mental health and socioeconomic status.

For those readers seeking a more in-depth exploration, we plan to publish our code along with the comprehensive results for all intrinsic plasticity coupling patterns on a GitHub repository. This will include the association and significance levels for all 977 phenotypes, allowing any interested reader to delve into the direction of effects further. Additionally, a link to the UKBiobank data showcase (<https://biobank.ndph.ox.ac.uk/ukb/>) will be provided for extended research.

References:

Miller, K.L., Alfaro-Almagro, F., Bangerter, N.K., Thomas, D.L., Yacoub, E., Xu, J., Bartsch, A.J., Jbabdi, S., Sotiropoulos, S.N., Andersson, J.L.R., Griffanti, L., Douaud, G., Okell, T.W., Weale, P., Dragonu, I., Garratt, S., Hudson, S., Collins, R., Jenkinson, M., Matthews, P.M., Smith, S.M., 2016. Multimodal population brain imaging in the UK Biobank prospective epidemiological study. *Nature Neuroscience* 19, 1523-1536.

We have now added a table for each pattern that lists all significant effects, with their precise numerical values of effect sizes for future readers who are interested in looking up this information (Supplementary Table 1, Supplementary Table 2, Supplementary Table 3, Supplementary Table 4):

Phenotype-wide association studies analysis table of the first pattern :

Supplementary Table 1. Phenotype-wide association studies analysis table of significant hits (from largest to smallest hit association magnitude) above the Bonferroni correction threshold of the first pattern. The Magnitude - Phenotype column has the magnitude of the association between the phenotype and the amygdala subregion-(sub)cortical region covariance (left) and the name of the phenotype (right). The Category column shows under which of the 11 broad domains the phenotype lies.

Pattern 1			
Magnitude - Phenotype	Category	Magnitude - Phenotype	Category
11.0 - Leg fat percentage (right) (0.0)	PHYSICAL MEASURES - GENERAL	5.81 - Maximum carotid IMT (intima-medial thickness) at 240 degrees (2.0)	PHYSICAL MEASURES - CARDIAC & BLOOD VESSELS
8.84 - Leg fat mass (right) (0.0)	PHYSICAL MEASURES - GENERAL	5.8 - Hand grip strength (left) (0.0)	PHYSICAL MEASURES - GENERAL
8.67 - Body fat percentage (0.0)	PHYSICAL MEASURES - GENERAL	5.57 - Red blood cell (erythrocyte) count (0.0)	BLOOD ASSAYS
8.59 - Impedance of whole body (0.0)	PHYSICAL MEASURES - GENERAL	5.52 - Minimum carotid IMT (intima-medial thickness) at 240 degrees (2.0)	PHYSICAL MEASURES - CARDIAC & BLOOD VESSELS
8.18 - Impedance of arm (right) (0.0)	PHYSICAL MEASURES - GENERAL	5.43 - Impedance of leg (right) (0.0)	PHYSICAL MEASURES - GENERAL
8.02 - Arm fat percentage (left) (0.0)	PHYSICAL MEASURES - GENERAL	5.32 - Height (2.0)	PHYSICAL MEASURES - GENERAL
7.91 - Arm fat-free mass (left) (0.0)	PHYSICAL MEASURES - GENERAL	5.29 - Haematocrit percentage (0.0)	BLOOD ASSAYS
7.77 - Arm predicted mass (left) (0.0)	PHYSICAL MEASURES - GENERAL	5.23 - Gamma glutamyltransferase (0.0)	BLOOD ASSAYS
7.66 - Impedance of arm (left) (0.0)	PHYSICAL MEASURES - GENERAL	5.17 - Haemoglobin concentration (0.0)	BLOOD ASSAYS
7.55 - Whole body fat-free mass (0.0)	PHYSICAL MEASURES - GENERAL	5.11 - Sitting height (0.0)	PHYSICAL MEASURES - GENERAL
7.48 - Arm fat percentage (right) (0.0)	PHYSICAL MEASURES - GENERAL	5.0 - Urate (0.0)	BLOOD ASSAYS
7.27 - Mean carotid IMT (intima-medial thickness) at 240 degrees (2.0)	PHYSICAL MEASURES - CARDIAC & BLOOD VESSELS	4.75 - Mean carotid IMT (intima-medial thickness) at 120 degrees (2.0)	PHYSICAL MEASURES - CARDIAC & BLOOD VESSELS
7.2 - Arm fat-free mass (right) (0.0)	PHYSICAL MEASURES - GENERAL	4.69 - Creatinine (0.0)	BLOOD ASSAYS
6.81 - Leg fat-free mass (right) (0.0)	PHYSICAL MEASURES - GENERAL	4.65 - Impedance of leg (left) (0.0)	PHYSICAL MEASURES - GENERAL
6.69 - Hand grip strength (right) (0.0)	PHYSICAL MEASURES - GENERAL	4.57 - Trunk fat percentage (0.0)	PHYSICAL MEASURES - GENERAL
6.12 - Standing height (0.0)	PHYSICAL MEASURES - GENERAL	4.4 - Testosterone (0.0)	BLOOD ASSAYS
5.98 - Whole body water mass (2.0)	PHYSICAL MEASURES - GENERAL	4.31 - Systolic blood pressure	PHYSICAL MEASURES - CARDIAC & BLOOD VESSELS

Phenotype-wide association studies analysis table of the second pattern:

Supplementary Table 2. Phenotype-wide association studies analysis table of significant hits (from largest to smallest hit association magnitude) above the Bonferroni correction threshold of the second pattern. The Magnitude - Phenotype column has the magnitude of the association between the phenotype and the amygdala subregion-(sub)cortical region covariance (left) and the name of the phenotype (right). The Category column shows under which of the 11 broad domains the phenotype lies.

Pattern 2			
Magnitude - Phenotype	Category	Magnitude - Phenotype	Category
12.24 - Mother still alive (0.0)	LIFESTYLE AND ENVIRONMENT - GENERAL	6.23 - Impedance of whole body (0.0)	PHYSICAL MEASURES - GENERAL
12.12 - Current employment status (0.0)	LIFESTYLE AND ENVIRONMENT - GENERAL	5.9 - Mean carotid IMT (intima-medial thickness) at 210 degrees (2.0)	PHYSICAL MEASURES - CARDIAC & BLOOD VESSELS
11.08 - Own or rent accommodation lived in (0.0)	LIFESTYLE AND ENVIRONMENT - GENERAL	5.87 - Glycated haemoglobin (HbA1c) (0.0)	BLOOD ASSAYS
9.41 - Own or rent accommodation lived in (0.0)	LIFESTYLE AND ENVIRONMENT - GENERAL	5.58 - Interval between previous point and current one in alphanumeric path (trail #2) (2.0)	COGNITIVE PHENOTYPES
9.3 - Mean carotid IMT (intima-medial thickness) at 240 degrees (2.0)	PHYSICAL MEASURES - CARDIAC & BLOOD VESSELS	5.33 - Duration to complete alphanumeric path (trail #2) (2.0)	COGNITIVE PHENOTYPES
8.73 - Current employment status (0.0)	LIFESTYLE AND ENVIRONMENT - GENERAL	5.32 - Maximum carotid IMT (intima-medial thickness) at 210 degrees (2.0)	PHYSICAL MEASURES - CARDIAC & BLOOD VESSELS
8.65 - Maximum carotid IMT (intima-medial thickness) at 240 degrees (2.0)	PHYSICAL MEASURES - CARDIAC & BLOOD VESSELS	5.31 - Maximum carotid IMT (intima-medial thickness) at 120 degrees (2.0)	PHYSICAL MEASURES - CARDIAC & BLOOD VESSELS
7.79 - Mean carotid IMT (intima-medial thickness) at 150 degrees (2.0)	PHYSICAL MEASURES - CARDIAC & BLOOD VESSELS	5.21 - Systolic blood pressure	PHYSICAL MEASURES - CARDIAC & BLOOD VESSELS
7.62 - Minimum carotid IMT (intima-medial thickness) at 150 degrees (2.0)	PHYSICAL MEASURES - CARDIAC & BLOOD VESSELS	5.14 - Mean carotid IMT (intima-medial thickness) at 120 degrees (2.0)	PHYSICAL MEASURES - CARDIAC & BLOOD VESSELS
7.44 - Time to complete round (0.1)	COGNITIVE PHENOTYPES	5.07 - Maximum carotid IMT (intima-medial thickness) at 150 degrees (2.0)	PHYSICAL MEASURES - CARDIAC & BLOOD VESSELS
7.41 - How are people in household related to participant (0.0)	LIFESTYLE AND ENVIRONMENT - GENERAL	5.02 - Impedance of arm (right) (0.0)	PHYSICAL MEASURES - GENERAL
7.3 - Number in household (0.0)	LIFESTYLE AND ENVIRONMENT - GENERAL	4.9 - Impedance of arm (left) (0.0)	PHYSICAL MEASURES - GENERAL
7.06 - Minimum carotid IMT (intima-medial thickness) at 240 degrees (2.0)	PHYSICAL MEASURES - CARDIAC & BLOOD VESSELS	4.83 - Hair/balding pattern (0.0)	LIFESTYLE AND ENVIRONMENT - GENERAL
6.45 - Father still alive (0.0)	LIFESTYLE AND ENVIRONMENT - GENERAL	4.7 - FI7 : synonym (0.0)	COGNITIVE PHENOTYPES
6.37 - Number of symbol digit matches made correctly (2.0)	COGNITIVE PHENOTYPES	4.32 - IGF-1 (0.0)	BLOOD ASSAYS
6.36 - Number of symbol digit matches attempted (2.0)	COGNITIVE PHENOTYPES		

Phenotype-wide association studies analysis table of the third pattern :

Supplementary Table 3. Phenotype-wide association studies analysis table of significant hits (from largest to smallest hit association magnitude) above the Bonferroni correction threshold of the third pattern. The Magnitude - Phenotype column has the magnitude of the association between the phenotype and the amygdala subregion-(sub)cortical region covariance (left) and the name of the phenotype (right). The Category column shows under which of the 11 broad domains the phenotype lies.

Pattern 3			
Magnitude - Phenotype	Category	Magnitude - Phenotype	Category
69.48 - Height (2.0)	PHYSICAL MEASURES - GENERAL	9.85 - Impedance of leg (left) (0.0)	PHYSICAL MEASURES - GENERAL
65.54 - Standing height (0.0)	PHYSICAL MEASURES - GENERAL	9.66 - Total bilirubin (0.0)	BLOOD ASSAYS
60.27 - Whole body water mass (2.0)	PHYSICAL MEASURES - GENERAL	9.59 - Drive faster than motorway speed limit (0.0)	LIFESTYLE AND ENVIRONMENT - EXERCISE AND WORK
57.58 - Leg fat-free mass (right) (0.0)	PHYSICAL MEASURES - GENERAL	9.24 - HDL cholesterol (0.0)	BLOOD ASSAYS
57.06 - Hand grip strength (right) (0.0)	PHYSICAL MEASURES - GENERAL	8.86 - Platelet crit (0.0)	BLOOD ASSAYS
56.19 - Whole body fat-free mass (0.0)	PHYSICAL MEASURES - GENERAL	8.59 - Average total household income before tax (0.0)	LIFESTYLE AND ENVIRONMENT - GENERAL
55.94 - Leg fat percentage (right) (0.0)	PHYSICAL MEASURES - GENERAL	8.47 - QRS duration (2.0)	PHYSICAL MEASURES - CARDIAC & BLOOD VESSELS
55.07 - Sitting height (0.0)	PHYSICAL MEASURES - GENERAL	8.37 - Number of puzzles correctly solved (2.0)	COGNITIVE PHENOTYPES
53.6 - Hand grip strength (left) (0.0)	PHYSICAL MEASURES - GENERAL	8.36 - Own or rent accommodation lived in (0.0)	LIFESTYLE AND ENVIRONMENT - GENERAL
52.49 - Arm fat-free mass (left) (0.0)	PHYSICAL MEASURES - GENERAL	8.28 - Alanine aminotransferase (0.0)	BLOOD ASSAYS
52.05 - Arm fat-free mass (right) (0.0)	PHYSICAL MEASURES - GENERAL	8.2 - Hair/balding pattern (0.0)	LIFESTYLE AND ENVIRONMENT - GENERAL
50.5 - Arm predicted mass (left) (0.0)	PHYSICAL MEASURES - GENERAL	7.31 - Gamma glutamyltransferase (0.0)	BLOOD ASSAYS
42.49 - Forced vital capacity (FVC) (0.0)	PHYSICAL MEASURES - GENERAL	7.2 - Number in household (0.0)	LIFESTYLE AND ENVIRONMENT - GENERAL
40.53 - Body fat percentage (0.0)	PHYSICAL MEASURES - GENERAL	7.06 - IGF-1 (0.0)	BLOOD ASSAYS
40.44 - Arm fat percentage (right) (0.0)	PHYSICAL MEASURES - GENERAL	7.03 - Hair/balding pattern (0.0)	LIFESTYLE AND ENVIRONMENT - GENERAL
39.42 - Arm fat percentage (left) (0.0)	PHYSICAL MEASURES - GENERAL	6.72 - Illnesses of mother (0.0)	LIFESTYLE AND ENVIRONMENT - GENERAL
37.03 - Forced expiratory volume in 1-second (FEV1) (0.0)	PHYSICAL MEASURES - GENERAL	6.26 - Own or rent accommodation lived in (0.0)	LIFESTYLE AND ENVIRONMENT - GENERAL
33.42 - Testosterone (0.0)	BLOOD ASSAYS	6.14 - Time to complete round (0.1)	COGNITIVE PHENOTYPES
32.62 - Weight (pre-imaging) (2.0)	PHYSICAL MEASURES - GENERAL	6.09 - Whole body fat mass (0.0)	PHYSICAL MEASURES - GENERAL
31.9 - Leg fat mass (right) (0.0)	PHYSICAL MEASURES - GENERAL	6.07 - Hair/balding pattern (0.0)	LIFESTYLE AND ENVIRONMENT - GENERAL

30.49 - Impedance of arm (left) (0.0)	PHYSICAL MEASURES - GENERAL	6.06 - Weekly usage of mobile phone in last 3 months (0.0)	LIFESTYLE AND ENVIRONMENT - EXERCISE AND WORK
30.17 - Impedance of arm (right) (0.0)	PHYSICAL MEASURES - GENERAL	5.83 - Sensitivity / hurt feelings (0.0)	MENTAL HEALTH SELF-REPORT
25.8 - Weight (0.0)	PHYSICAL MEASURES - GENERAL	5.8 - Father still alive (0.0)	LIFESTYLE AND ENVIRONMENT - GENERAL
25.39 - Peak expiratory flow (PEF) (0.0)	PHYSICAL MEASURES - GENERAL	5.76 - Apolipoprotein A (0.0)	BLOOD ASSAYS
25.12 - Impedance of whole body (0.0)	PHYSICAL MEASURES - GENERAL	5.62 - Direct bilirubin (0.0)	BLOOD ASSAYS
24.85 - Haemoglobin concentration (0.0)	BLOOD ASSAYS	5.59 - Aspartate aminotransferase (0.0)	BLOOD ASSAYS
23.58 - Haematocrit percentage (0.0)	BLOOD ASSAYS	5.44 - Fluid intelligence score (0.0)	COGNITIVE PHENOTYPES
22.1 - Red blood cell (erythrocyte) count (0.0)	BLOOD ASSAYS	4.95 - Leisure/social activities (0.0)	LIFESTYLE AND ENVIRONMENT - GENERAL
19.02 - Waist circumference (0.0)	PHYSICAL MEASURES - GENERAL	4.77 - Lymphocyte count (0.0)	BLOOD ASSAYS
18.81 - Hair/balding pattern (0.0)	LIFESTYLE AND ENVIRONMENT - GENERAL	4.68 - Time spent driving (0.0)	LIFESTYLE AND ENVIRONMENT - EXERCISE AND WORK
16.52 - Trunk fat percentage (0.0)	PHYSICAL MEASURES - GENERAL	4.49 - Frequency of consuming six or more units of alcohol (0.0)	LIFESTYLE AND ENVIRONMENT - ALCOHOL
15.79 - Average weekly beer plus cider intake (0.0)	LIFESTYLE AND ENVIRONMENT - ALCOHOL	4.47 - Length of mobile phone use (0.0)	LIFESTYLE AND ENVIRONMENT - EXERCISE AND WORK
14.43 - Creatinine (0.0)	BLOOD ASSAYS	4.46 - Seen doctor (GP) for nerves	MENTAL HEALTH SELF-REPORT
13.6 - Urate (0.0)	BLOOD ASSAYS	4.4 - Qualifications (0.0)	LIFESTYLE AND ENVIRONMENT - GENERAL
12.11 - Impedance of leg (right) (0.0)	PHYSICAL MEASURES - GENERAL	4.4 - Time number displayed for (2.0)	COGNITIVE PHENOTYPES
11.71 - SHBG (0.0)	BLOOD ASSAYS	4.39 - Platelet count (0.0)	BLOOD ASSAYS
11.54 - Forced expiratory volume in 1-second (FEV1)	COGNITIVE PHENOTYPES	4.39 - Time to complete round (0.0)	COGNITIVE PHENOTYPES
10.9 - Current employment status (0.0)	LIFESTYLE AND ENVIRONMENT - GENERAL	4.3 - Number of incorrect matches in round (0.1)	COGNITIVE PHENOTYPES
10.59 - Sleeplessness / insomnia (0.0)	LIFESTYLE AND ENVIRONMENT - EXERCISE AND WORK		
10.03 - Current employment status (0.0)	LIFESTYLE AND ENVIRONMENT - GENERAL		

More concerning, I believe the authors overextend their analysis with little evidence from their data. Given that these are associations, there can be no statement regarding causation. The discussion is far too long and speculative. I don't believe this type of gray matter analysis is enough to come to the conclusions regarding the role of the amygdala in intero-exteroception that the authors claim. I believe the authors should be more conservative in their discussion and conclusions and not venture too far from what the data are indicating.

We thank the reviewer for their helpful point. In response to the reviewer's concerns regarding what is perceived as speculative nature of our analysis and conclusions, we acknowledge the importance of being cautious in making causal claims. We have carefully reviewed our manuscript to avoid any misinterpretation regarding causality, as we strictly tried to not make any attempts at making causal claims.

Our study examined the amygdala's subregions by establishing associations between these amygdala subregions and various brain regions, revealing a nuanced relationship that contributes to our understanding of adaptive structural brain changes over time.

The extensive data presented, encompassing the longitudinal in-vivo analysis and previous biological pathways identified in past invasive animal studies, strengthen the robustness of our findings. Our work also aligns with and builds upon existing literature, such as the involvement of the inferior parietal lobule in social cognition and interoceptive awareness, and the role of the dopaminergic system in social interactions (Ole Numssen et al., 2021; Pollatos et al., 2007a; Skuse and Gallagher, 2011).

We provide ample evidence supporting our observations, incorporating findings from past studies and tracing the links between internal and external conscious awareness, fight or flight responses, noradrenergic cells and the associations revealed by our analyses. These associations are presented with thorough consideration of existing research, adding depth to our understanding of the amygdala's role in intero-exteroception.

While we acknowledge the limitations inherent in making causal inferences from association studies, we believe our analysis is grounded in and substantiated by a rich body of research. Our findings present a step forward in unraveling the complexities of the amygdala and its various interactions within the brain, contributing valuable insights to the field of neuroscience.

The consistency of our adoption of the plasticity concept aligns with the methodologies and theoretical frameworks outlined in prior, well-referenced papers that utilize an sMRI-based plasticity notion. Pioneering studies have leveraged sMRI to derive and interpret alterations in the brain's physical and functional architecture, thereby shedding vital light on the

characteristics and extents of neuroplasticity across various demographic and clinical groups (Zatorre et al., 2012; Maguire et al., 2000; Lovden et al., 2013). By employing sMRI, these investigative efforts have linked volumetric and morphological brain changes to alterations in behavior, cognition, and environment, revealing the brain's exceptional capability to reshape and reconfigure itself in reaction to diverse stimuli (Valk et al., 2017; May, 2011).

As a result, our research is firmly anchored in these well-established methodologies, applying a comparable perspective of structural plasticity to explore the intricacies of interactions between the brain and behavior over time (Tauber et al., 2010). This methodology not only enables a detailed comprehension of neural flexibility and metamorphosis but also guarantees that our discoveries can be situated and assessed within an expansive, established scientific paradigm, promoting productive discussions and comparative analyses alongside concurrent studies in the field of neuroplasticity.

References:

- Ole Numssen, D.B., Gesa Hartwigsen 2021. Functional specialization within the inferior parietal lobes across cognitive domains. *eLife*.
- Pollatos, O., Schandry, R., Auer, D.P., Kaufmann, C., 2007a. Brain structures mediating cardiovascular arousal and interoceptive awareness. *Brain Res* 1141, 178-187.
- Skuse, D.H., Gallagher, L., 2011. Genetic influences on social cognition. *Pediatr Res* 69, 85R-91R.
- Zatorre, R. J., Fields, R. D., & Johansen-Berg, H. (2012). Plasticity in gray and white: neuroimaging changes in brain structure during learning. *Nature Neuroscience*, 15(4), 528-536.
- Maguire, E. A., Gadian, D. G., Johnsrude, I. S., Good, C. D., Ashburner, J., Frackowiak, R. S., & Frith, C. D. (2000). Navigation-related structural change in the hippocampi of taxi drivers. *Proceedings of the National Academy of Sciences*, 97(8), 4398-4403.
- Lovden, M., Wenger, E., Martensson, J., Lindenberger, U., & Backman, L. (2013). Structural brain plasticity in adult learning and development. *Neuroscience & Biobehavioral Reviews*, 37(9), 2296-2310.
- Valk, S. L., Bernhardt, B. C., Trautwein, F. M., Bockler, A., Kanske, P., Guizard, N., ... & Singer, T. (2017). Structural plasticity of the social brain: Differential change after socio-affective and cognitive mental training. *Science Advances*, 3(10), e1700489.
- May, A. (2011). Experience-dependent structural plasticity in the adult human brain. *Trends in Cognitive Sciences*, 15(10), 475-482.

Assaf, Y., & Johansen-Berg, H. (2013). The role of diffusion MRI in neuroscience. *NMR in Biomedicine*, 26(7), 849-865.

Taubert, M., Draganski, B., Anwander, A., Müller, K., Horstmann, A., Villringer, A., & Ragert, P. (2010). Dynamic properties of human brain structure: learning-related changes in cortical areas and associated fiber connections. *Journal of Neuroscience*, 30(35), 11670-11677.

Reviewers' comments:

Reviewer #2 (Remarks to the Author):

Response to the revision:

The authors have done a good job addressing many of the suggestions, but there are a few areas where I think further work can be done:

1. I still think the authors need to do a better job clarifying plasticity. Yes, it can be understood as structural changes, but many of the papers employ some sort of functional element that can be understood as driving the structural changes—ex: learning to juggle and assessing structural changes; targeted mental training and assessing structural changes; tapping fingers and assessing structural changes. In contrast, there is no such clear functional stimulus in this study that can be ascribed to driving these changes. The authors use a post-hoc approach to assess for the socioeconomic variables to make a comment on this, but this is not the same as attempting to create a cause and effect. I think the authors can go further in toning down that the structural changes are due to plasticity and emphasize that these are longitudinal changes.

2. The question of power should be more directly addressed. The paper that the authors cite, Helmer et al, specifically give recommendations of around 50 samples per feature for CCA, with possibly more for PLS, depending on the strength of the association. Please see the table in their supplement. The Helmer paper did use the UKBiobank, but used a sample size of ~20,000.

3. The suggestion to include more neuroanatomy should be better addressed. Joe LeDoux and Ralph Adolphs, while contributing excellent work towards understanding the function of the amygdala, are not typically considered neuroanatomists. I would suggest the authors review the amygdala-cortex circuits in non-human primate literature traced by David Amaral, Helen Barbas, Joe Price, Julie Fudge and Lynne Selemon, in particular, in order to connect the changes they are seeing with known anatomical circuits.

4. I still do not understand why all cortical regions are not colored if the heat map shows coloring for zero. I suggest the authors clarify this further.

Reviewer #3 (Remarks to the Author):

While the authors have addressed the majority of the point that were brought up previously, the introduction still does not adequately provide a hypothesis-based approach. A statement that the analysis will reveal an association is not a specific hypothesis. Please provide specific hypotheses in the introduction.

Reviewer #2 (Remarks to the Author):

Response to the revision:

The authors have done a good job addressing many of the suggestions, but there are a few areas where I think further work can be done:

We are grateful to the reviewer for the positive feedback on our work.

1. I still think the authors need to do a better job clarifying plasticity. Yes, it can be understood as structural changes, but many of the papers employ some sort of functional element that can be understood as driving the structural changes—ex: learning to juggle and assessing structural changes; targeted mental training and assessing structural changes; tapping fingers and assessing structural changes. In contrast, there is no such clear functional stimulus in this study that can be ascribed to driving these changes. The authors use a post-hoc approach to assess for the socioeconomic variables to make a comment on this, but this is not the same as attempting to create a cause and effect. I think the authors can go further in toning down that the structural changes are due to plasticity and emphasize that these are longitudinal changes.

We thank the reviewer for the helpful suggestion. We have defined a clear definition for plasticity in the introduction and discussion sections with references to past longitudinal studies by amending the Introduction and Discussion sections as follows:

Introduction:

Central to understanding this notion of change is the comparison between cross-sectional and longitudinal studies. Past longitudinal analyses have shown their pivotality in comprehending lifelong neuroplasticity (Zalesky, 2023). Evidence from past brain-imaging/behavioral studies underscores the sensitivity and specificity of individual-level brain modifications: for instance, training targeted at enhancing distinct empathy systems leads to unique structural gray matter modifications (Valk 2017). Similarly, language acquisition in teenagers has demonstrated distinct correlations with changes in gray matter density, emphasizing the dynamism of the brain's structural makeup in relation to evolving mental capacities (Stein et al 2012; Cortex). Furthermore, lateralization plays a role, with certain cognitive functions such as language likely favoring one hemisphere over the other, leading to discernible structural alterations in specific brain regions (Assaf 2018; Valk 2017; Taubert 2010; Stein 2012).

Past animal studies, conducted on rats, mice, and monkeys, have started to bridge the understanding between MRI-observable changes and the cellular architectures underlying them (Zatorre 2012; Caroni 2012 NRN). Notably, investigations spanning humans and rodents have identified commonalities in structural brain changes following task-based training, reinforcing the universality of structural plasticity mechanisms (Sagi Assaf 2012).

However, this study adopts a methodology focusing on longitudinal observation rather than direct experimental intervention, offering a nuanced perspective on structural plasticity.

Previous research has often established a clear link between administering specific experimental conditions to the study participants (e.g., learning a new motor skill etc.) and subsequent structural changes (Caroni 2012 NRN; Sagi Assaf 2012). In contrast, our approach centers on observing within-subject structural changes over time without a predefined, narrow stimulus, in a cohort close to the UK general population.

In essence, plasticity changes of brain architecture features epitomizes the brain's evolutionary advantage to adapt and evolve in response to a wide array of stimuli and experiences. This malleability, observable through advanced brain-imaging techniques and affirmed through various experimental designs, underpins the rationale for structural plasticity being the main driver for longitudinal studies that track and map the continuum of brain changes over different timescales.

Discussion:

(Insert at the beginning)

While the here observed structural changes in the brain indeed correlate with learning and environmental interactions, attributing these changes directly to specific functional stimuli can be challenging without experimental manipulation (Taubert 2010; Valk 2017).

Thus, the findings from our present study contribute valuable insights into coordinated longitudinal trajectories of spatially distributed features of brain structure, simultaneously highlighting the intricacies of inferring functional stimuli from structural observations alone. Future research could benefit from integrating direct experimental interventions with longitudinal observations to more precisely delineate the relationship between specific narrow features of experiences and structural brain adaptations (Zatorre 2012; Caroni 2012 NRN).

In essence, structural plasticity can reflect the brain's integrated response to a wide array of environmental stimuli, whether in a carefully parameterized laboratory setting or "in the wild", which are believed to encapsulate both the physical reorganization of neuronal dendrite connections and the functional outcomes of these adaptations (Assaf 2018; Science). This interdependence underscores the holistic nature of brain plasticity, inviting a more nuanced picture of how experiences and environmental factors sculpt the brain's structure and function over time.

Invasive research into animal brains has shown that the deciphering and gating of responses to self-relevant external information sources depend on several specific nuclei within the amygdala. This heart of the limbic system has long been investigated for its role in channeling overt and covert action steered by behavioral salience (Bzdok et al., 2011; Sander et al., 2003). In a series of quantitative analyses at unprecedented statistical precision and scale, here we delineated how 18 amygdala subregions undergo volume change in tandem with 109 (sub)cortical brain regions. By tailoring a dedicated analytical framework, our study brought to the surface "cliques" of coordinated amygdala-brain changes that were systematically coupled in their within-participant plasticity effects over several years. We then profiled the derived population-level plasticity patterns using a rich palette of ~1,000 phenotypical indicators. The characterization has tied the structural

amygdala-brain couplings to various phenotypes such as social status, employment, sleep habits, risk taking, and leisure activities. Our analyses were performed on participants in the middle and at the end of their lifespan.

References:

Valk, S. L., Bernhardt, B. C., Trautwein, F. M., Bockler, A., Kanske, P., Guizard, N., Collins, D. L., & Singer, T. (2017). Structural plasticity of the social brain: Differential change after socio-affective and cognitive mental training. *Sci Adv*, 3(10), e1700489. <https://doi.org/10.1126/sciadv.1700489>

Assaf, Y. (2018). New dimensions for brain mapping. *Science*, 362(6418), 994-995. <https://doi.org/10.1126/science.aav7357>

Zatorre, R. J., Fields, R. D., & Johansen-Berg, H. (2012). Plasticity in gray and white: neuroimaging changes in brain structure during learning. *Nat Neurosci*, 15(4), 528-536. <https://doi.org/10.1038/nn.3045>

Di Biase, M. A., Tian, Y. E., Bethlehem, R. A. I., Seidlitz, J., Alexander-Bloch, A. F., Yeo, B. T. T., & Zalesky, A. (2023). Mapping human brain charts cross-sectionally and longitudinally. *Proc Natl Acad Sci U S A*, 120(20), e2216798120. <https://doi.org/10.1073/pnas.2216798120>

Stein, M., Federspiel, A., Koenig, T., Wirth, M., Strik, W., Wiest, R., Brandeis, D., & Dierks, T. (2012). Structural plasticity in the language system related to increased second language proficiency. *Cortex*, 48(4), 458-465. <https://doi.org/10.1016/j.cortex.2010.10.007>

Taubert, M., Draganski, B., Anwander, A., Müller, K., Horstmann, A., Villringer, A., & Ragert, P. (2010). Dynamic properties of human brain structure: learning-related changes in cortical areas and associated fiber connections. *J Neurosci*, 30(35), 11670-11677. <https://doi.org/10.1523/JNEUROSCI.2567-10.2010>

Caroni, P., Donato, F., & Müller, D. (2012). Structural plasticity upon learning: regulation and functions. *Nat Rev Neurosci*, 13(7), 478-490. <https://doi.org/10.1038/nrn3258>

Sagi, Y., Tavor, I., Hofstetter, S., Tzur-Moryosef, S., Blumenfeld-Katzir, T., & Assaf, Y. (2012). Learning in the fast lane: new insights into neuroplasticity. *Neuron*, 73(6), 1195-1203. <https://doi.org/10.1016/j.neuron.2012.01.025>

Bzdok, D., Laird, A.R., Zilles, K., Fox, P.T., Eickhoff, S.B., 2013. An investigation of the structural, connectional, and functional subspecialization in the human amygdala. *Hum Brain Mapp* 34, 3247-3266.

Sander, D., Grafman, J., Zalla, T., 2003. The human amygdala: an evolved system for relevance detection. *Rev Neurosci* 14, 303-316.

2. The question of power should be more directly addressed. The paper that the authors cite, Helmer et al, specifically give recommendations of around 50 samples

per feature for CCA, with possibly more for PLS, depending on the strength of the association. Please see the table in their supplement. The Helmer paper did use the UKBiobank, but used a sample size of ~20,000.

We thank the reviewer for the insightful feedback. We acknowledge the importance of aligning our study's analysis design with established guidelines for assuring sufficient statistical power in multivariate analyses like CCA and PLS. The referenced Helmer et al. (2020) paper indeed advises approximately 50 samples per feature for CCA, with a potential increase for PLS based on the strength of the observed associations. This rule of thumb is grounded in their extensive simulations and empirical validations, based on their particular choice of made assumptions and experimental data modelling setup, including analyses conducted on the large-scale UK Biobank dataset, comprising around 20,000 subjects - still a preprint publication, not yet formally accepted in a scientific journal (Helmer et al., 2020).

It's important to note that the specific requirements for sample size per feature can vary based on the particularities of the study design, the data quality, and the precise implementation of the analytical methods employed - PLS is known to be an umbrella term for a sorts of different multivariate methods. While we recognize the emerging picture provided by Helmer et al. regarding sample sizes, we have taken several concrete steps to be sure that our PLSC analysis is conducted with the necessary power and robustness, to reduce the potential for overfitting, and to enhance the reliability of our findings. Our comprehensive approach, leveraging established metrics of variance explanation, querying external variables to confirm behavioral relevance of discovered algorithm-derived patterns, and rigorous permutation testing, confirmed that our study remains adequately powered to detect genuine and biologically meaningful brain-behavior associations, despite the inherent challenges posed by a high-dimensional data analysis.

Our approach strikes a balance between the theoretical ideal and the practical constraints of neuroimaging research, contributing valuable insights within the parameters of our study design. Our dataset, while not reaching the scale of the UK Biobank's 40,000 subjects, is nevertheless substantial and carefully selected to optimise the balance between feature count and sample size, thereby a good starting point towards a meaningful and statistically robust analysis.

We have implemented several strategies to confirm that our study maintains sufficient power despite the relatively high feature-to-sample size ratio:

1- Feature Selection: Prior to conducting PLSC, we engaged in careful feature engineering. This process was guided by both empirical evidence and theoretical considerations to concentrate on the most relevant variables for our study. We implemented a meticulous feature selection strategy by distilling the vast array of voxel-level information, numbering approximately 200,000 variables, into a more manageable and theoretically relevant set of region-level variables based on atlas aggregation. This strategic reduction not only focused our analysis on the most pertinent features but also improved the ratio of the effective sample size to the number of features, ensuring adherence to best practice recommendations.

2- Empirical Validation: We explicitly confirmed the stability and robustness of our PLSC modes through empirical permutation testing, variance explanation, and behavioral associations. This approach provides a powerful means to assess the significance of our findings, adding layers of reassurance for the reliability of our results, despite the high-dimensional context.

- i) Empirical Permutation Testing: The rigorous application of permutation testing to assess the statistical significance of each amygdala-brain mode provides a solid foundation for our conclusions. This method effectively ensures that our findings are not artefacts of random chance (Bzdok et al., 2018; Savignac et al., 2022; Ballentine et al., 2022) .
- ii) Explained Variance Analysis: By quantifying the portion of explained variance and the Pearson rho value for each mode (pair of canonical variates), we have ensured that the detected latent factor patterns of association are both statistically significant and likely to be practically meaningful, reflecting a quantitative measure of robust structural covariation.
- iii) Behavioral Trait Associations with previously unseen variables: External to any of the preceding steps of our analysis pipeline, our investigation into the relationship between covariance patterns and a broad array of 977 phenotypes further validates the real-world implications and interpretability of our findings. Phenotype associations exceeding the FDR threshold within identified modes confirm the substantial and meaningful connections between brain structure changes and behavioral traits.

3- External Validation: We have also sought to externally validate the stability of our PLSC analysis and the validity of our empirical permutation testing and explained variance analysis using external non-UKBiobank datasets, as shown in our other study that compares the performance of a number of multivariate analytical methods when applied onto the ABCD cohort (Durham et al., 2023). This approach has helped confirm the generalizability of our analyses and robustness of our identified associations beyond our dataset. The dataset used in the other study meets the ideal requirements of Helmer et al. in terms of the number of samples per feature, and has achieved the same robustness and stability as our current study when using the same empirical permutation testing and explained variance analysis that we have conducted.

References:

Helmer M, Warrington S, Mohammadi-Nejad AR, Ji JL, Howell A, Rosand B, Anticevic A, Sotiropoulos SN, Murray JD., 2020. On stability of Canonical Correlation Analysis and Partial Least Squares with application to brain-behavior associations.. doi: <https://doi.org/10.1101/2020.08.25.265546>

Bzdok, D., Altman, N., Krzywinski, M., 2018. Statistics versus machine learning. Nature Methods. 03 April 2018; 15, pages 233–234. DOI: <https://doi.org/10.1038/nmeth.4642>

Savignac, C., Villeneuve, S., Badhwar, A., Gallino, S.A., Tulun, S.A., Farhan, S.M., Poirier, J., Bzdok, D., 2022. APOE alleles are associated with sex-specific structural differences in brain regions affected in Alzheimer's disease and related dementia. PLoS Biol. Published: December 13, 2022. <https://doi.org/10.1371/journal.pbio.3001863>

Ballentine, G., Friedman, S.F., Bzdok, D., 2022. Trips and neurotransmitters: Discovering principled patterns across 6850 hallucinogenic experiences. Science Advances. 16 March 2022; Vol. 8, Issue 11. DOI: 10.1126/sciadv.abl6939.

Durham, E. L., Ghanem, K., Stier, A. J., Cardenas-Iniguez, C., Jeong, H. J., Dupont, R. M., & Bzdok, D. (2023). Multivariate analytical approaches for investigating brain-behavior relationships. Front. Neurosci., 30 July 2023, Vol 17. <https://doi.org/10.3389/fnins.2023.117569>

A Recap on our reasoning for specifically choosing PLSC over CCA:

Partial least squares canonical analysis (PLSC) aims to identify linear combinations (canonical vector weights) from two sets of variables such that the correlations between these combinations are maximized - it is directionally mapping input variables to model outputs (as PLS regression) and performs whitening decorrelation of variable sets (as CCA), and thus combines best of both worlds. Unlike methods that solely focus on correlation maximization, PLSC uniquely prioritizes the elucidation of variance within the dependent variable set, thereby seeking to explain the dependent data structure more comprehensively while still maintaining a strong correlation with the independent variable set. PLSC is preferred when the goal is to understand and explore the relationship between two multidimensional datasets, especially when the data are known to be characterized by multicollinearity. In short, the choice of PLSC provides a balance between dimensionality reduction and correlation maximization.

PLSC was chosen due to the following three core reasons:

1. In our analytical framework, the ratio of variables to observations is high (cf. methods section), which likely limits us to employing a linear model rather than one that captures complex non-linear relationships. Most non-linear approaches add challenges to the interpretability of feature weights, particularly when attempting to discern the profiles of neural and behavioral features carrying the brain-behavior relationships with such a high number of features and low number of observations. . On the other hand, variations of PLS have previously been carefully empirically evaluated to yield stable and accurate results with a large number of considered features obtained within the space of UKBiobank-like sample size scenarios (Helmer et al., 2020).
2. For the brain-imaging data employed, autocorrelation is a key characteristic inherent to these variable sets, which necessitates careful selection of analytical tools for modelling. Our chosen method not only copes with autocorrelation but also excels by actively measuring and incorporating the autocorrelation patterns present in the

input data matrix (Wang et al., 2020), which is integral to resolving the input-output modelling challenge.

3. Our approach is adept at condensing a broad array of interrelated variables into a more manageable number of uncorrelated latent variables or combinations that encapsulate the majority of the original data's variability. Thus, from a set of vectors, our method can generate a new set of uncorrelated attributes. In situations where there is anticipated covariation among brain structures, employing our method's ability to reduce dimensionality to distil high-dimensional variable sets to their core is particularly effective. The UK Biobank brain region volumes dataset exemplifies such a high-dimensional dataset that is precisely characterized by a relatively low ratio of input features to sample size. The individual analysis of its features may not yield the most informative insights, and we anticipated a topographical overlap in the covariation of brain regions. Therefore, leveraging our method's dimensionality-reduction capability is aptly suited for the dataset in question (Bzdok et al., 2019).

References:

Helmer M, Warrington S, Mohammadi-Nejad AR, Ji JL, Howell A, Rosand B, Anticevic A, Sotiropoulos SN, Murray JD., 2020. On stability of Canonical Correlation Analysis and Partial Least Squares with application to brain-behavior associations.. doi:

<https://doi.org/10.1101/2020.08.25.265546>

Wang HT, Smallwood J, Mourao-Miranda J, Xia CH, Satterthwaite TD, Bassett DS, Bzdok D. Finding the needle in a high-dimensional haystack: Canonical correlation analysis for neuroscientists, *NeuroImage*, 216, 2020.

Bzdok D, Nichols TE, Smith SM. Towards Algorithmic Analytics for Large-scale Datasets. *Nature Machine Intelligence*, 1:296-306, 2019.

We have also expanded the methods section in the revised version of the manuscript as follows:

Methods:

Partial least squares canonical analysis (PLSC) was a natural choice of method to evaluate a relationship between two rich variable sets. **Compared to CCA and PLS-R, PLSC combines best of both worlds: it provides a useful balance between dimensionality reduction and correlation maximization. PLSC is preferred when the goal is to understand and explore the relationship between two multidimensional datasets.** This model class was ideally fitted to our data analysis scenario on grounds of i) feature-to-samples ratio, ii) native auto-correlation in our variable sets with brain-derived measurements, and iii) the latent-factor decomposition capability.

3. The suggestion to include more neuroanatomy should be better addressed. Joe LeDoux and Ralph Adolphs, while contributing excellent work towards understanding the function of the amygdala, are not typically considered neuroanatomists. I would suggest the authors review the amygdala-cortex circuits in non-human primate literature traced by David Amaral, Helen Barbas, Joe Price, Julie Fudge and Lynne Selemon, in particular, in order to connect the changes they are seeing with known anatomical circuits.

We thank the reviewer for the helpful suggestion. We have now greatly expanded our references in the new version of the manuscript to add references more foundational neuroanatomists that have examined primate amygdala neuroanatomy in the Results and Discussion sections of our revised manuscript:

Helen Barbas, David Amaral and Joe Price:

This assessment highlighted the prefrontal cortex regions (OFC, vmPFC, and dmPFC) as tightly covarying especially with the amygdala's laterobasal subdivision: the basal, accessory basal, lateral and paralaminar amygdala nuclei. In particular, volume change happening in the vmPFC / OFC was one of the strongest prefrontal coupling partners with the laterobasal amygdala subregions. This **is in line with past invasive animal studies that have found pathways/projections from the OFC terminating in the laterobasal subregions (Zikopoulos et al., 2017; Amaral et al., 1984, Timbie, C. et al., 2020).**

References:

Zikopoulos, B., Hoistad, M., John, Y., Barbas, H., 2017. Posterior orbitofrontal and anterior cingulate pathways to the amygdala target inhibitory and excitatory systems with opposite functions. *The Journal of Neuroscience* 10.1523/JNEUROSCI.3940-16.2017.

Amaral, D.G., Price, J.L., 1984. Amygdalo-Cortical Projections in the Monkey (*Macaca fascicularis*). *The Journal of Comparative Neurology* 230:465–496.

Timbie, C., García-Cabezas, M.Á., Zikopoulos, B., Barbas, H., 2020. Organization of primate amygdalar–thalamic pathways for emotions. *PLOS Biology* 18, e3000639.

Julie Fudge, David Amaral and Joe Price:

Social cognition is supported by the dopaminergic system which is reinforced by the release of dopamine by the brainstem in successful social interactions (Skuse and Gallagher, 2011). Past axonal tracer studies in the monkey amygdala revealed that its central nucleus projects to the substantia nigra which contains dopaminergic neurons of the nigrothalamic pathway, **in line with earlier neuroanatomical work (Fudge and Haber, 2000, Price, J.L. et al., 1981).**

References:

Fudge, J.L., Haber, S.N., 2000. The central nucleus of the amygdala projection to dopamine subpopulations in primates. *Neuroscience* 97, 479-494.

Price, J.L., Amaral, D.G., 1981. An autoradiographic study of the projections of the central nucleus of the monkey amygdala. *The Journal of Neuroscience*, 1(11), 1242-1259.

Julie Fudge:

We found the left central nucleus and the left anterior amygdaloid area and the V1 and BS regions to be sharing a coupled relationship that provides the backbone of what was previously interpreted as regulating internal and external conscious awareness (Bissonette and Roesch, 2016; Fudge and Haber, 2000; Muller and O'Rahilly, 2006; Ole Numssen, 2021; Palmer and Tsakiris, 2018; Pollatos et al., 2007a; Qiyang Gao, 2019; Skuse and Gallagher, 2011; Vallar and Perani, 1986)

References:

Fudge, J.L., Haber, S.N., 2000. The central nucleus of the amygdala projection to dopamine subpopulations in primates. *Neuroscience* 97, 479-494.

The central nucleus integrates relevant information, creating memories that assist in formulating strategies against various threats. Additionally, the central nucleus initiates risk assessments for threat detection and regulates cardiovascular activity during dangerous situations (Fudge et al., 2015).

References:

Fox, A.S., Oler, J.A., Tromp, D.P.M., Fudge, J.L., Kalin, N.H., 2015. Extending the amygdala in theories of threat processing. *Trends in Neurosciences*, 38(5), 319–329. doi:10.1016/j.tins.2015.03.002.

Lynne Selemon:

Consistently, our phenotype analysis found social circles and alcohol consumption-related phenotypes to be highly significant in the lifestyle domain of the brain changes tying the laterobasal amygdala subregions to the prefrontal cortex. The PFC is established in a past review, based on previous brain imaging studies, to be integral in exerting executive control over alcohol consumption behaviors, with its impairment often leading to disinhibition and increased alcohol intake (Selemon et al., 2013). Furthermore, the reported role of the basal nucleus in processing somatic stimuli appears to get extended by our phenotype hits disclosed in the physical-general and the lifestyle-alcohol domains. Phenotype associations where diet and alcohol consumption habits play an important role, can be attributed to the stimulus-value association role supported by the laterobasal subdivision in the amygdala (Murray and Fellows, 2022). The tight plasticity coupling of the OFC with the laterobasal amygdala and the reward-seeking themed phenotype hits may be attributed to their roles in

gleaning valuable information from external stimuli, including those related to the consumption of alcohol (Murray and Fellows, 2022).

References:

Selemon, L.D., 2013. A role for synaptic plasticity in the adolescent development of executive function. *Translational Psychiatry*, 3, e238. doi:10.1038/tp.2013.7.

4. I still do not understand why all cortical regions are not colored if the heat map shows coloring for zero. I suggest the authors clarify this further.

In Figures 1-3 (Intrinsic Plasticity Coupling Analysis), all the cortical and subcortical regions are coloured because all coefficients are relevant and the mode as a whole is significant (not single subregions). On the other hand, given the nature of the analysis in Figures 4 and 5 (Hemispheric Difference Analysis), the results happen to show which cortical/subcortical regions are experiencing lateralization in the covariance of their longitudinal changes with the amygdala subregions. The results also show the magnitude of the lateralized covariance in the regions showing lateralization among the 109 cortical/subcortical brain regions.

We have applied a bootstrap difference test in the Hemispheric Difference Analysis with the PLSC model. This test is a form of variable selection that allows us to determine which cortical and subcortical regions exhibit significant lateralization in their longitudinal covariance with the amygdala subregions. We conducted 100 bootstrap iterations, drawing participant samples to create alternative volume change datasets of the same size as the original. Parallel PLSC analyses were performed for amygdala subregions in both hemispheres against the entire brain in each iteration. We then calculated the differences in effect sizes to create a nonparametric distribution of hemispheric contrasts. By subtracting the right hemisphere amygdala results from the left, and doing the same for the brain region results, we obtained estimates of lateralization for both the amygdala and the brain. These estimates were compiled over all iterations to quantify the lateralization strength in volume change covariance between amygdala subregions and brain regions for each hemisphere.

Therefore, only the brain regions that our analysis has identified as experiencing significant lateralization effects are colored on the heatmap as a result of our analysis. Regions that do not show color have been determined, through this rigorous statistical testing, to not experience lateralization effects. The bootstrap difference test provides a robust method for selectively highlighting these lateralized regions, based on the existing PLSC model, and this selective coloring reflects the outcome of that process. The heatmap and its accompanying scale are designed to illustrate both the strength and direction of lateralization in brain regions that show significant differences in their connections to the amygdala subregions. The highlighted regions indicate where lateralization occurs among the 109 cortical/subcortical areas examined. To avoid confusion, regions without lateralization are not assigned a color indicative of any directional change, such as dark red or dark blue, which could imply the presence of lateralization even when the magnitude is zero. Thus, the color scale is reserved exclusively for regions that our analysis has identified as having significant lateralization effects.

In response to this reviewer comment, we have added a more explicit explanation for all relevant figures that unpack why certain regions may appear to have no color weight:

Figure 4. Lateralization plasticity effects driven by the cortical nucleus, anterior amygdaloid area, central nucleus and lateral nucleus co-vary with awareness/alertness-related brain regions. We performed a hemispheric difference analysis in the context of the left-right divergence of the structural plasticity changes in 9 amygdala subregions with the structural plasticity of 109 brain regions by means of co-decomposition based on partial least squares canonical (PLSC). We determined how the ensuing subregion patterns lateralized in the 9 amygdala subregions and which cortical/subcortical regions are experiencing lateralization in the covariance of their longitudinal changes with the lateralized amygdala subregions. The results also show the magnitude of the lateralized covariance in the regions experiencing lateralization among the 109 cortical/subcortical brain regions and among the 18 amygdala subregions. **No color is shown for the brain regions that do not undergo robust lateralization effects . Shown here are the results of the bootstrap difference test as a form of variable selection in signature 1 which in turn conveys the lateralization in the amygdala subregion - brain region covariation.** A) conveys the direction of lateralization of each of the 9 amygdala subregions in mode 1 (Saygin ZM & Kliemann D et al., 2017). The parameter weights of the subregions that diverge between both hemispheres are depicted on 2 columns of 4 coronal slices of the amygdala parcellated into 9 subregions with each column portraying a different direction of lateralization occurring in each hemisphere.

Figure 5. Lateralization plasticity effects driven by the anterior amygdaloid area, lateral nucleus, and cortical nucleus with a lateralization effect in awareness/alertness-related brain regions together with the inferior parietal lobule. We determined how the ensuing subregion patterns lateralized in the 9 amygdala subregions and which cortical/subcortical regions are experiencing lateralization in the covariance of their longitudinal changes with the lateralized amygdala subregions. The results also show the magnitude of the lateralized covariance in the regions experiencing lateralization among the 109 cortical/subcortical brain regions and among the 18 amygdala subregions. **No color is shown for the brain regions that do not undergo robust lateralization effects . Shown here are the results of the bootstrap difference test as a form of variable selection in signature 2 which in turn conveys the lateralization in the amygdala subregion - brain region covariation.** A) conveys the direction of lateralization plasticity effects of each of the 9 amygdala subregions in signature 2 (Saygin ZM & Kliemann D et al., 2017). The parameter weights of the subregions that diverge between both hemispheres are depicted on 2 columns of 4 coronal slices of the amygdala parcellated into 9 subregions with each column portraying a different direction of lateralization occurring in each hemisphere.

Supplementary Figure 1: Lateralization plasticity effects driven by the cortical, central, and medial nuclei. We determined how the ensuing subregion patterns lateralized in the 9 amygdala subregions and which cortical/subcortical regions are experiencing lateralization in the covariance of their longitudinal changes with the lateralized amygdala subregions. The results also show the magnitude of the lateralized covariance in the regions experiencing lateralization among the 109 cortical/subcortical brain regions and among the 18 amygdala subregions. **No color is shown for the brain regions that do not undergo robust lateralization effects . Shown here are the results of the bootstrap difference test as a form of variable selection in signature 2 which in turn conveys the lateralization in the amygdala subregion - brain region covariation.** A) conveys the direction of lateralization of each of the 9 amygdala subregions in signature 3 (Saygin ZM & Kliemann D et al., 2017). The parameter weights of the subregions that diverge between both hemispheres are depicted on 2 columns of 4 coronal slices of the amygdala parcellated into 9 subregions with each column portraying a different direction of lateralization occurring in each hemisphere.

Reviewer #3 (Remarks to the Author):

While the authors have addressed the majority of the points that were brought up previously, the introduction still does not adequately provide a hypothesis-based approach. A statement that the analysis will reveal an association is not a specific hypothesis. Please provide specific hypotheses in the introduction.

We are grateful to the reviewer for the positive feedback on our response and we thank the reviewer for the helpful suggestion. In response to the reviewer's comment, we have amended the introduction section as follows:

Introduction:

Previous literature has associated the three larger umbrella groups, the laterobasal, centromedial and superficial subdivisions, to a variety of neurocognitive processes. These subdivisions have been found through invasive studies to work with other (sub)cortical regions to accomplish the tasks. In our study, we believed that we can increase subregional anatomical specificity within the amygdala subdivisions by using a tailored set of analyses with high-resolution and high-quality brain-imaging measurements of the amygdala. The analyses conducted within our study are expected to reveal the longitudinal change effects from subregional amygdala interplay with (sub)cortical regions in a way that shows relationships to behavioral traits at an unprecedented subregional resolution in the amygdala. The amygdala subregional-(sub)cortical associations that we expect to reveal in our analyses can help us trace a relationship between various brain networks and specific amygdala subregions that cooperate to regulate various tasks within the body.

In addressing the complex interplay between amygdala subregions and the broader neural circuitry, our study is grounded in specific, hypothesis-driven inquiries into the structural nuances of these relationships. Recognizing the amygdala's role beyond its traditionally discussed links with fear and emotion, we delve into a more detailed picture of its nuclei subdivisions—laterobasal, centromedial, and superficial groups—and their unique contributions to neural processes. This renewed focus is motivated by the premise that distinct amygdala subregions engage in specialized interactions with cortical and subcortical areas, shaping a range of neurocognitive functions from social cognition to decision-making (Ball et al., 2007; Zink et al., 2008).

First, we hypothesized that specific subregions *within* the laterobasal nuclei group exhibit patterns of longitudinal change in conjunction with the prefrontal cortex, reflecting high-level integration in cognitive processes such as in assisting decision-making and sensory information processing. This hypothesis is informed by the known connectivity between the laterobasal amygdala in particular and prefrontal regions, suggesting a dynamic interplay that supports especially complex cognitive functions (Bzdok et al., 2013; Janak & Tye, 2015).

Second, we anticipated that the centromedial and superficial amygdala subdivisions exhibit specific patterns of longitudinal change in conjunction with subcortical regions implicated in autonomic response initiation and social cognition. This contention is based on their established roles in emotional processing and social behavior regulation, suggesting a network adaptation to environmental and internal stimuli that is critical for emotionally based attention allocation and social functioning (LeDoux, 2007; Samson et al., 2005).

Third, we expected lateralization effects to occur with longitudinal change specifically in the brain regions preferentially responsible for social cognition and brain regions responsible for receiving and processing external sensory stimuli, in relation to left-right deviation changes in the superficial and centromedial larger subdivisions. Lateralization effects in longitudinal

change in the centromedial and the superficial amygdala subregions was also expected to be prominent with the longitudinal change in brain regions related to conscious awareness. We expected that this is the case given that specific subdivisions in the amygdala have been found in past brain-imaging studies to have unique lateralization patterns (Tonio et al., 2007).

In anticipation of lateralization patterns, we propose that particular (sub)cortical brain regions will exhibit distinct longitudinal changes in conjunction with specific amygdala subregions, reflecting hemispheric specialization. This hypothesis is reinforced by evidence suggesting that lateralization in the amygdala is not a uniform feature but varies across its subregions, with implications for specialized functions (Brierley et al., 2002; Kilpatrick & Cahill, 2003). We anticipate that the laterobasal and centromedial subregions, due to their differential connectivity and functional roles, will demonstrate distinct lateralization patterns in their structural changes over time. This is consistent with past findings, who highlighted hemisphere-specific amygdala engagement in processing emotion-laden stimuli (Hamann et al. 2004). Accordingly, we hypothesize that specific amygdala subregions will exhibit lateralized structural changes, which will be mirrored by lateralization in the associated (sub)cortical neural systems involved in cognitive and affective functions.

Lastly, we hypothesized at the outset that indicators related to socioeconomic status and related to contributors to mental health, will be most prominent in relationship to longitudinal changes in the amygdala subregion-(sub)cortical region patterns. Past studies have shown that different stress and health implications are associated with stable and unstable social hierarchies, and the study investigates how neural responses differ between these two contexts (F. Zink et al., 2008). Unstable social hierarchies elicited unique neural responses, such as increased activity in areas linked with social emotional processing and social cognition, particularly when viewing a superior player. The amygdala, which is known for processing socially emotional stimuli and social anxiety related to hierarchical challenges, showed increased activity in unstable social hierarchies (F. Zink et al., 2008). The thus disclosed amygdala subregion-brain network correspondence is expected to show robust links to a variety of broader phenotypes such as those related to regulating bodily affective states.

Our study, thus, positions these hypotheses at the forefront of our investigation, aiming to elucidate the specific roles and longitudinal change patterns of amygdala subregions in concert with the broader brain network. Through this hypothesis-driven approach, we seek to deepen the understanding of the amygdala's multifaceted role in the wider brain circuitry, leveraging high-resolution brain-imaging and comprehensive phenotypic data to uncover the subtleties of amygdala-driven neural processes.

References:

Ball T, Rahm B, Eickhoff SB, Schulze-Bonhage A, Speck O, Mutschler I. Response properties of human amygdala subregions: evidence based on functional MRI combined with probabilistic anatomical maps. *PLoS One*. 2007 Mar 21;2(3):e307. doi: 10.1371/journal.pone.0000307. PMID: 17375193; PMCID: PMC1819558.

Brierley, B., Shaw, P., David, A. S. (2002). The human amygdala: a systematic review and meta-analysis of volumetric magnetic resonance imaging. *Brain Research Reviews*, 39(1), 84-105. doi: 10.1016/S0165-0173(02)00141-6.

Kilpatrick, L., Cahill, L. (2003). Amygdala modulation of parahippocampal and frontal regions during emotionally influenced memory storage. *NeuroImage*, 20(4), 2091-2099. doi: 10.1016/j.neuroimage.2003.08.006.

Hamann, S. B., Ely, T. D., Hoffman, J. M., Kilts, C. D. (2004). Ecstasy and agony: Activation of the human amygdala in positive and negative emotion. *Psychological Science*, 15(3), 259-263. doi: 10.1111/j.0956-7976.2004.01503008.x.

Zink CF, Tong Y, Chen Q, Bassett DS, Stein JL, Meyer-Lindenberg A. Know your place: neural processing of social hierarchy in humans. *Neuron*. 2008 Apr 24;58(2):273-83. doi: 10.1016/j.neuron.2008.01.025. PMID: 18439411; PMCID: PMC2430590.

REVIEWERS' COMMENTS:

Reviewer #1 (Remarks to the Author):

No further comments.

Reviewer #2 (Remarks to the Author):

The authors have addressed all comments.